# Flatland: The Adventures of Gradient Descent with Large Step Sizes

**Leonardo Galli** [* 1 2] **Curtis Fox** [* 3] **Wiebke Bartolomaeus** [1 2] **Mark Schmidt** [3 4] **Holger Rauhut** [1 2]

## Abstract

The training of neural networks often entails objective functions that are not globally $L$-smooth. For these functions, it is both theoretically and practically difficult to answer the question: *what is the largest possible step size that ensures the convergence of gradient descent (GD)?* We address this longstanding open question in deep learning by providing a unifying definition of "large" step sizes that requires only local Lipschitz (or even Hölder) continuity of the gradient. We design first-order adaptive methods that provably yield large step sizes and show that they operate at the edge of stability (EoS) right from the start of the training. In particular, the loss decreases nonmonotonically and the product between the step size and sharpness, i.e., the largest eigenvalue of the Hessian, stays above the EoS threshold of 2 throughout training. Using our method, we are able to decrease the sharpness down to its global minimum. Contrary to expectation, we find that encountering globally-flat regions too early in training may slow down convergence and jeopardize the generalization ability of the network. Exploiting a self-stabilization argument, we allow GD to enter slightly sharper valleys and turn unsuccessful training runs into successful ones.

## 1. Introduction

In the recent literature on Deep Learning (DL), the term sharpness commonly refers to the largest eigenvalue of the Hessian $H$ of the training loss. A longstanding hypothesis (Hochreiter & Schmidhuber, 1997) suggests that solutions with low sharpness (flat minima) may lead to better generalization error when training neural networks. Despite the ongoing debate concerning the correlation between

low sharpness and good generalization (Keskar et al., 2017; Dinh et al., 2017; Petzka et al., 2021; Andriushchenko et al., 2023), the success of Sharpness Aware Minimization (SAM) (Foret et al., 2020) in improving the generalization properties of neural networks across datasets and tasks suggests that flatter minima may be (at least on average) preferable over sharper ones.

In a large variety of architectures and datasets, the connection between sharpness and step sizes has been extensively explored in Cohen et al. (2021). The paper numerically observes that gradient descent (GD) with a fixed step size $\eta$ goes through two distinct phases. In the first phase (progressive sharpening), the loss function decreases monotonically, while the sharpness increases. In the second phase (Edge of Stability, EoS), the loss decreases nonmonotonically, while the sharpness stabilizes around the value $2/\eta$. By testing various fixed step sizes, the paper also shows that the step sizes that hit the EoS very early in the training yield the most spiking oscillations in function values while also achieving the fastest convergence. Beyond these step sizes, the training usually diverges because of large, uncontrolled oscillations. Given these observations, large step sizes seem preferable over small ones because they both speed up the training and yield flatter solutions.

The interest in large step sizes has rapidly grown as a consequence of the results in Cohen et al. (2021). In the recent literature, various effects of large step sizes have been proven under specific assumptions, e.g., speeding up convergence (Wu et al., 2024), escaping sharp minima (Mohtashami et al., 2023), and improving generalization (Ren et al., 2024). In applications, large learning rates have been shown to enhance model compressibility and robustness to duplicated features (Barsbey et al., 2025), and to speed up the overwriting of previous learned features in continual learning (Lyle et al., 2025). However, the definition of "large" and "small" step sizes is incoherent between papers (Mohtashami et al., 2023; Wang et al., 2025). Furthermore, as already observed in Cohen et al. (2021), it is unclear whether GD with a certain step size $\eta$ will ever hit the edge of stability, and if so, how early this will occur in the training process. In this paper, we address these questions by providing a unifying definition of "large" step sizes which still allows GD to converge. Moreover, we propose an algorithm that finds these step sizes and consequently show that GD operates at

---

[*]Equal contribution [1]LMU Munich [2]Munich Center for Machine Learning [3]University of British Columbia [4]Canada CIFAR AI Chair (Amii). Correspondence to: Leonardo Galli <galli@math.lmu.de>, Curtis Fox <curtfox@cs.ubc.ca>.

*Proceedings of the 43rd International Conference on Machine Learning*, Seoul, South Korea. PMLR 306, 2026. Copyright 2026 by the author(s).

the EoS right from the start of the training without the use of second-order information (Roulet et al., 2024). With this method, we provide a concrete answer to a longstanding open question in DL (Goodfellow et al., 2016):

*Q: How to select the largest possible learning rates while still ensuring the convergence of GD?*

In answering this question, we find that GD with the largest possible converging step sizes often finds regions of the loss landscape where the sharpness achieves its global minimum. Interestingly enough, these globally-flat regions are found by a first-order method that may be able to pick up second-order information only through its oscillatory behavior (Cohen et al., 2025; Shugart & Altschuler, 2025). Contrary to expectations, our results suggest that finding points where the sharpness is *globally* minimized is not beneficial for the speed of convergence nor for the generalization ability of the resulting network. As we discuss further in Section 3, these are saddle-points that when encountered too early in training, do not allow the network to learn meaningful features. On the other hand, we show that if GD avoids the globally-flat regions and takes steps through slightly sharper valleys, it very efficiently minimizes the training loss while achieving improved test accuracy (see Figure 3).

Our contributions can be summarized as follows:

- We propose a new definition of large step sizes and an algorithm that practically finds them. Given the similar oscillating behavior achieved by the step sizes operating at the EoS (Cohen et al., 2021) and that of nonmonotone line searches (Grippo et al., 1986; Galli et al., 2023), our method exploits the latter to answer the question $Q$. We analyze the properties of these large step sizes both in terms of convergence and relationship to sharpness. We show that they are not only Lipschitz-aware, but also Hölder-aware for any $k \geq 0$ (see Propositions 2.5 and 2.8). Without ever employing second-order information, we further show that these properties translate into an approximate sharpness-awareness in proximity of a stationary point (see Corollary 2.10).
- Beyond verifying numerically that the above properties hold, we show that these large step sizes operate at the EoS right from the start of the training. On various architectures trained on different $K$-class classification datasets, we showcase that GD with large step sizes may quickly run into regions with very low and unexpectedly constant sharpness values. We further characterize these points and find that they are almost stationary and that the $K$-largest eigenvalues of the Hessian all achieve the exact value of $\frac{2}{K}$. We prove that $\frac{2}{K}$ is the global minimum value of the sharpness, as the sub-Hessian of a square error loss calculated w.r.t. the bias term of the last layer is indeed diagonal with $K$ entries at $\frac{2}{K}$. Moreover, we observe that in some cases the eigenvalues just below the first $K$

pair-wise sum to 0. This implies that the above-mentioned stationary points are actually saddle.

- Among the conducted experiments, we identify a subset of those that "get stuck" in the globally-flat regions for many iterations, while not managing to reduce the training loss. We label these trainings as "unsuccessfully-flat" and notice that all other experiments in our benchmark never bring the sharpness to its global minimum. We thus identify another subset of runs that get very close to the globally-flat regions, but with sharpness values that are slightly higher than $\frac{2}{K}$. We label these runs "successfully-flat", as they often achieve very low training loss and high test accuracy. By exploiting a self-stabilization argument (Damian et al., 2022), we propose to limit our large step sizes to stay below the value of $K$ and we turn "unsuccessfully-flat" runs into "successfully-flat" ones.

## 2. Large Step Sizes and How to Find Them

The training of modern neural networks is known to involve objective functions that may not be globally smooth (Li et al., 2023). We overcome this limitation by assuming instead that $f$ is only locally $l(w, \eta_w)$-smooth and that this property holds only along the anti-gradient direction. We call *segment smoothness* the resulting intersection between local and directional smoothness (Mishkin et al., 2024).

**Definition 2.1.** Let $f$ be continuously differentiable ($f \in C^1(\mathbb{R}^n)$) and let $w_\eta := w - \eta \nabla f(w)$, then we define the local segment smoothness function $l : \mathbb{R}^n \times \mathbb{R}_+ \to \mathbb{R}_+$ as

$$l(w, \eta) := \sup_{\hat{\eta} \in (0, \eta]} \frac{\| \nabla f(w) - \nabla f(w_{\hat{\eta}}) \|}{\| w - w_{\hat{\eta}} \|}, \qquad (1)$$

with the convention that $\frac{0}{0} = 0$.

**Assumption 2.2** (Segment Smoothness). Let $f \in C^1(\mathbb{R}^n)$, we say that $f$ is segment smooth if for every $w \in \mathbb{R}^n$ there exists an $\eta_w > 0$ such that $l(w, \eta_w)$ is bounded. Without loss of generality[1], we can assume that $\eta_w \geq \frac{2}{l(w, \eta_w)}$.

Given the assumption above, one can derive a local descent lemma (see Lemma B.5) which guarantees monotone decrease of the function value at iteration $k$ as long as $\eta < \frac{2}{l(w_k, \eta_{w_k})}$. On the contrary, if $\eta$ is always above $\frac{2}{l(w_k, \eta_{w_k})}$, it is easy to find examples for which GD diverges (e.g., on quadratic functions). Our definition of large step sizes requires that $\eta > \frac{2}{l(w_k, \eta_{w_k})}$ holds at least every $P$ iterations, but not for any $k$ (Wang et al., 2025). In fact, the goal is to enforce an oscillatory behavior in $f$, as happens at the EoS and as required in Wu et al. (2024) to prove GD's improved convergence speed.

---

[1]The key point is that $l(w, \eta)$ is monotone $\eta$, which allows us to increase $\eta_w$ until it satisfies $\eta_w \geq \frac{2}{l(w, \eta_w)}$. More details are given in Lemma B.4

**Definition 2.3** (Large Step Sizes)**.** Let $\{\eta_k\}$ be a sequence of step sizes, $\{w_k\}$ the corresponding sequence of iterates, and $P \in \mathbb{N}$. Then for any iteration $k$, the step size $\eta_k$ is said to be large if $(i)$ $\eta_k \geq \frac{1}{l(w_k, \eta_{w_k})}$ and $(ii)$ there exists a integer $p \in [0, P]$ such that $\eta_{k+p} > \frac{2}{l(w_{k+p}, \eta_{w_{k+p}})}$.

Given the above definition of large step sizes, we now propose a practical algorithm to find them. Although the nonmonotone convergence of GD at the EoS may be at first surprising, a deeper study reveals that the same behavior was already observed when employing nonmonotone line searches. These methods were first designed in Grippo et al. (1986) to accept large step sizes that may not lead to a monotonic decrease of the objective function $f$, as required instead by their monotone counter-part (Armijo, 1966). Additionally, the convergence of GD with a nonmonotone line search can be ensured despite the nonmonotonic decrease of $f$ (see Theorem 2.9).

**Definition 2.4.** We call $R_k$ a *reference value* for a function $f$ and iterates $\{w_k\}_{k \in \mathbb{N}}$ if $R_k \geq f(w_k)$ for all $k \in \mathbb{N}$.

For example, $R_k = f(w_k)$ recovers the monotone line search, and $R_k = \max_i f(w_{k-i})$ is the one suggested in (Grippo et al., 1986). For $c \in (0, 1)$, the step size $\eta_k > 0$ in (non-)monotone line searches has to satisfy

$$f(w_k - \eta_k \nabla f(w_k)) \leq R_k - c\eta_k \|\nabla f(w_k)\|^2. \quad (2)$$

Setting $\epsilon_k := R_k - f(w_k) \geq 0$, the above inequality can be reformulated as

$$f(w_k - \eta_k \nabla f(w_k)) \leq f(w_k) - c\eta_k \|\nabla f(w_k)\|^2 + \epsilon_k. \quad (3)$$

In order to find the step sizes that are "large" according to Definition 2.3, we design a new line search algorithm with both backtracking and extrapolation steps. More specifically, if (3) is initially false, the step size is halved until the inequality is true; if (3) is initially true, the step size is instead doubled (see Algorithm 2 for more details). We call line searches following this procedure *equality line searches* as one can show that $\eta_k$ approximates $\eta_k^*$, i.e., the step size for which (3) is satisfied as an equality. In particular, from Lemma B.3 we have $\eta_k^* \in (\eta_k, \frac{\eta_k}{2})$. Notice that instead of doubling or halving the step size at each internal iteration of the line search, one could more generally multiply or divide the step size by a constant $\delta \in (0, 1)$. For the step sizes yielded by the equality line searches, we have the following properties.

**Proposition 2.5** (Lipschitz-Awareness)**.** *Let $f$ be segment smooth and let $\eta_k$ be the result of an equality line search (Algorithm 2) employed to satisfy (3) with $\epsilon_k \geq 0$, then*

$$\eta_k \geq \frac{2\delta(1-c)}{l(w_k, \eta_{w_k})}. \quad (4)$$

Thus, if we choose $c$ and $\delta$ such that $2\delta(1 - c) \geq 1$, the property $(i)$ of Definition 2.3 is guaranteed. Moreover, by setting $\delta \approx 1$ and $c \approx 0$, then $(ii)$ also holds with $P \approx 0$ (see Lemma B.9 for more details).

While the previous bound is a property of both monotone and nonmonotone equality line searches, the next property only holds for the latter. That is, we can obtain a similar result for a more general class of functions, i.e., functions with Hölder continuous gradients along the segment $[w, w_{\eta_w}]$.

**Definition 2.6.** Let $f \in C^1(\mathbb{R}^n)$ and $\nu \in [0, 1]$, we define the local Hölder segment smoothness function $M_\nu : \mathbb{R}^n \times \mathbb{R}_+ \to \mathbb{R}$ as

$$M_\nu(w, \eta) = \sup_{\hat{\eta} \in (0, \eta]} \frac{\|\nabla f(w) - \nabla f(w_{\hat{\eta}})\|}{\|w - w_{\hat{\eta}}\|^\nu},$$

again with the convention that $\frac{0}{0} = 0$.

**Assumption 2.7** (Hölder segment smoothness)**.** Let $f \in C^1(\mathbb{R}^n)$ and $\nu \in [0, 1]$, we say that $f$ is Hölder segment smooth if $\forall w \in \mathbb{R}^n$, $\exists \eta_w > 0 : M_\nu(w, \eta_w)$ is finite. Without loss of generality we can assume for any fixed $d > 0$, that

$$\eta_w \geq \frac{2}{d M_\nu(w, \eta_w)^{\frac{2}{1+\nu}}}.$$

We then say $f$ is Hölder segment smooth with constant $d > 0$.

**Proposition 2.8.** *Let $f$ be Hölder segment smooth with $d = \min\{1, e^{-2\frac{f(x_0) - f^*}{e}}\}$ and let $\eta_k$ be the result of an equality line search employed to satisfy (3) with $\epsilon_k > 0$, then*

$$\eta_k \geq \frac{2\delta(1-c)}{\alpha(M_\nu(w_k, \eta_{w_k}), 2\epsilon_k)} \quad (5)$$

*with*

$$\alpha(M_\nu(w_k, \eta_{w_k}), \epsilon_k) := M_\nu(w_k, \eta_{w_k})^{\frac{2}{1+\nu}} \cdot \left(\frac{1-\nu}{1+\nu} \cdot \frac{1}{\epsilon_k}\right)^{\frac{1-\nu}{1+\nu}}.$$

Notice that for the reference value (Definition 2.4) from Zhang & Hager (2004) we have $\epsilon_k > 0$ for any finite $k$, but $\epsilon_k$ is sometimes 0 in the case of the original nonmonotone line search (Grippo et al., 1986). For the former, one can also prove the following convergence result without global Lipschitz continuity of $f$ as required instead by the latter. From now on, we will only consider the line search by Zhang & Hager (2004).

**Theorem 2.9.** *Let $f \in C^1(\mathbb{R}^n)$ be lower bounded with compact level set $\mathcal{L}_0$. Let $\{w_k\}$ be generated by GD and $\{\eta_k\}$ be generated by a nonmonotone equality line search of the type Zhang & Hager (2004) (Algorithm 4), then*

$$\lim_{k \to \infty} \|\nabla f(w_k)\| = 0.$$

To conclude on the properties, we have that when the gradient vanishes and $w_k \to w^*$, $l(w_k, \eta)$ converges to the directional sharpness at $w^*$ which implies that for large enough $k$, the step size $\eta_k$ yielded by the new equality line searches is lower bounded by the reciprocal of the sharpness.

**Corollary 2.10** (Sharpness-Awareness). *Let $f \in C^2(\mathbb{R}^n)$. Let $\{w_k\}$ be generated by GD where $\eta_k$ is generated by Algorithm 2. Let $f$ satisfy Assumption 2.2 on $w_k \; \forall k > \bar{k}$, and let $\lambda_i$ be the eigenvalues of the hessian ordered by their absolute value. Assume that $\{w_k\}$ converges to $w^*$. Then $\forall \varepsilon > 0, \exists \hat{k} > \bar{k}$:*

$$\eta_k \geq \frac{2(1-c)\delta}{|\lambda_1(H(w^*))| + \varepsilon} \quad \forall k \geq \hat{k}.$$

If we have that only a subsequence of $w_k$ converges to $w^*$, then the statement still holds by restricting to that subsequence. The proofs of all the results in the current and following sections are given in Appendix B.

### 2.1. Other Large Step Sizes

Beyond equality line searches, we show that there are other ways to achieve large step sizes and approximate sharpness-awareness. A step size with very similar properties is the one obtained via classical backtracking line searches initialized with a Polyak step size (Polyak, 1987; Galli et al., 2023). In this case, the initial step size on each iteration is set such that $\eta_{k,0} = \frac{f(w_k) - f^*}{c_p || \nabla f(w_k)||^2}$, where $f^*$ denotes the minimum value of $f$. For these step sizes (see Algorithm 3 in the Appendix), we can also show Lipschitz-awareness and sharpness-awareness, but not Hölder-awareness.

**Proposition 2.11.** *Let $f \in C^2(\mathbb{R}^n)$ and satisfy Assumption 2.2. Let $\eta_k$ be generated by Algorithm 3 and $H_\eta(w_k) := \nabla^2 f(w_k - \eta \nabla f(w_k))$, then*

$$\eta_k \geq \frac{\min\{2(1-c)\delta, 1/(2c_p)\}}{\max_{\eta \in [0, \eta_{w_k}]} |\lambda_1(H_\eta(w_k))|}. \tag{6}$$

In contraposition to line search methods, Malitsky & Mishchenko (2020) proposed an adaptive step size for which one can show that $\eta_k \geq \min_{i=1,\dots,k-1} \frac{1}{2l(w_i, \eta_{w_i})}$, but not Proposition 2.5. Another more recent method suggested by Roulet et al. (2024) employs second-order information to set the step size as

$$\eta_k := \sigma \frac{\|\nabla f(w_k)\|^2}{|\nabla f(w_k)^T H(w_k) \nabla f(w_k)| + \epsilon}, \tag{7}$$

for some small $\epsilon > 0$ and $\sigma > 0$ the "stepping-on-the-edge" parameter. Note that for this step size one can derive that

$$\eta_k \approx \sigma \frac{\|\nabla f(w_k)\|^2}{|\nabla f(w_k)^T H(w_k) \nabla f(w_k)|} \geq \sigma \frac{1}{|\lambda_1(H(w_k))|}.$$

However, for values of $\sigma$ that allow the method to operate at the EoS (i.e., $\sigma \geq 2$), the corresponding step size does not necessarily lead to the convergence of the method.

## 3. Numerical Results

We train various neural networks for image classification to verify whether the bounds derived in the previous section are satisfied and whether GD with these large step sizes operates at the EoS. As in Cohen et al. (2021), we run full-batch GD on a subsample of 5000 images from the original datasets and compute the sharpness every 100 iterations. Our experiments are conducted on the cartesian product of the datasets CIFAR10, CIFAR100, SVHN, and EMNIST with the following networks: a Multi-Layer Perceptron (MLP), a Convolutional Neural Network (CNN), resnet34, wide_resnet50_2, densenet121, and vgg11. In the main paper, we focus on 3 datasets, each with a different number of classes $K$, and train each model on a different dataset. In particular, a CNN on EMNIST letters with 26 classes, resnet34 on SVHN with 10 classes, and vgg11 on CIFAR100 with 100 classes. The rest of the experiments can be found in the appendix. We use the Mean Square Error (MSE) loss for the experiments in the main paper, while the cross-entropy experiments can be found in Appendix H. For more experimental details, see Appendix C.

In Figure 1 we compare a monotone equality line search (LS) and a nonmonotone equality line search (NLS) with PoNLS (the deterministic version of the nonmonotone line search PoNoS (Galli et al., 2023)). We also compare these methods with SAM (Foret et al., 2020), the quintessential sharpness-aware method, and Curvature Dynamics Aware Tuning (CDAT) (Roulet et al., 2024). In order from the top row to the bottom, Figure 1 presents the product between the sharpness $\lambda_1(H(w_k))$ and the step size $\eta_k$, the step size, the sharpness, and the training loss. Notice that to verify (4), we employ $\lambda_1(H(w_k))$ as an approximation for $l(w_k, \eta_{w_k})$, both because of our interest in EoS and because of the difficulties in computing $l(w_k, \eta_{w_k})$. We compare $\lambda_1(H(w_k))$ and a discretized version of $l(w_k, \eta_{w_k})$ in Appendix I.4 and in Figures 40-41.

When plotting $\lambda_1(H(w_k)) \cdot \eta_k$, we also report two more lines, the EoS threshold of 2 and the predicted lower bound of $2\delta(1-c)$. In Figure 1, we fix the line search parameters $c$ to $10^{-4}$ and $\delta$ to 0.5, as these are very commonly used values for these hyper-parameters when line searches are combined with full-batch GD (Nocedal & Wright, 2006). When $\delta \not\approx 1$, $(ii)$ of Definition 2.3 is not ensured by (4). Strictly speaking, with $c = 10^{-4}$ and $\delta = 0.5$, neither is $(i)$, which is why we check both numerically. Despite leaving to future works the runtime comparison between the methods, Appendix G presents some preliminary results showcasing that NLS, CDAT and SAM have very similar per-iteration

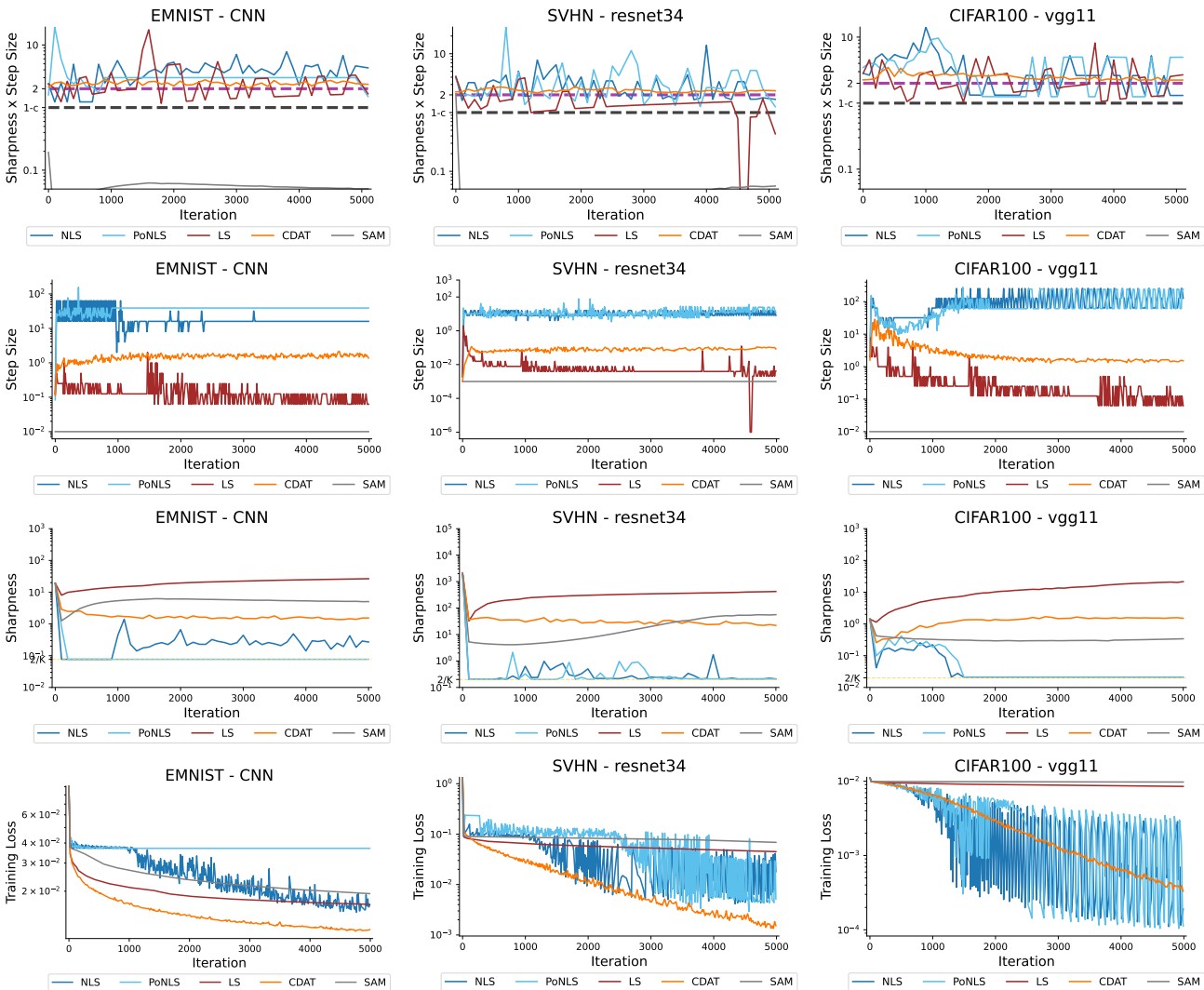

*Figure 1.* We plot the sharpness * step size (1st Row), step size (2nd Row), sharpness (3rd Row), and training loss (4th Row) for five different methods. We compare gradient descent with the LS, NLS, and PoNLS line searches as well as gradient descent with the CDAT step size selection and SAM. This is repeated for 3 different models and 3 different datasets.

cost when $\delta = 0.5$, while this is not the case when $\delta = 0.9$.

We first focus on NLS and PoNLS. We observe that $\eta_k \cdot \lambda_1(H(w_k))$ is always greater than $1 - c$, as proved in (4) and (6), and very often greater than 2. Together with the frequent oscillations in the loss seen in the last row of Figure 1, this shows that the step sizes yielded by NLS are large in the sense of Definition 2.3, sharpness-aware, and allow GD to operate at the EoS right from the beginning of training.

Moving onto LS, we see that the sharpness at each iteration continues increasing during training, while the corresponding step sizes are decreasing. We note that this is in agreement with the previous work by Roulet et al. (2024). A detailed explanation of the behavior of LS and the reason why $\eta_k \cdot \lambda_1(H(w_k))$ is not always above $1 - c$ despite Proposition 2.5 can be found in Appendix I.1.

For CDAT, the stepping-on-the-edge parameter $\sigma$ is fixed to 2.06, as suggested in Roulet et al. (2024), while the step size for SAM is selected via a grid search (see Appendix C). We notice that the product of sharpness and step size for CDAT is always above the EoS threshold of 2, while this product is not particularly meaningful for SAM as its EoS is evaluated differently (Long & Bartlett, 2024). While these methods are sometimes more and sometimes less successful than NLS and PoNLS in minimizing the training loss, NLS and PoNLS outperform SAM and CDAT in minimizing the sharpness. In most cases both NLS and PoNLS bring the sharpness to the surprisingly small and constant value of $2/K$, where $K$ is the number of classes in the dataset. In the next section, we will further study this phenomenon and show that the attained value is indeed the global minimum of the sharpness. For now, notice that when the globally-flat

region is encountered early in the training, the corresponding loss does not decrease below the value expected for a model that is randomly guessing an output (e.g., 0.09 with $K = 10$, see Appendix C.5 for the calculation). In the extreme case of PoNLS on EMNIST×CNN, the method is stuck at the same training loss throughout all of training.

Similar observations made for Figure 1 also hold in Figures 19-26. Beyond the setting proposed in the main paper, early globally-flat regions lead to poor training convergence in the case of stochastic GD (SGD) (Appendix E), training vision transformers (Appendix F), for warmed up step sizes (Appendix D), and for the cross-entropy loss (Appendix H).

### 3.1. Globally-Flat Points and their Characteristics

In this section, we provide a finer analysis of the points which attain a sharpness of $2/K$. Given a $K$-class classification problem minimized using the MSE loss (Cohen et al., 2021), the next lemma clarifies that the sharpness is indeed lower bounded by $2/K$, meaning that the points found by NLS and PoNLS are global minimizers of the sharpness. This result holds for most neural networks as the only requirement is that the last layer has a bias term (e.g., MLPs, CNNs, residual networks, transformers).

**Lemma 3.1** (Informal, see Lemma B.23). *Let $f$ be the MSE-loss. We consider networks $n : \mathbb{R}^n \to \mathbb{R}^K$ with linear last layer, i.e. $n(W, b) = Wx' + b$, where $x'$ is the output of the previous layer. Then the Hessian of $f$ w.r.t. $W$ and $b$ are*

$$\nabla_W^2 f(w) = \frac{2}{K} 1_k \otimes x' x'^T, \; \nabla_b^2 f(w) = \frac{2}{K} I_K.$$

In Figure 2, we re-run NLS exactly as in Figure 1, but focus on the first window of 100 iterations in which the sharpness hits $2/K$. The first row of Figure 2 shows the trace of the Hessian and the top-20 (30 for EMNIST) eigenvalues, where the positive eigenvalues are blue and negative ones are red. The second row shows the average of the gradient norms extracted per layer in blue and the norm of the gradient w.r.t. the last bias in cyan. Finally, for the last row we compute the percentage of the neurons with approximately zero output for each network layer separately, and plot the maximum percentage across the layers (see Appendix I.2 for the details on these measures).

We first observe that when the sharpness hits $2/K$, the following $K-1$-largest eigenvalues also achieve exactly the same value. Although this occurs in the EMNIST×CNN and SVHN×resnet34 experiments, it does not in the CIFAR100×vgg11 case where instead the largest eigenvalues hover slightly above $2/K$. Note that in our experiments (see also Figures 27-31), the model training accuracies correspond to random guessing for iterations where the sharpness is stuck at $2/K$. For this reason, we label these training runs "unsuccessfully-flat". Instead, we call training runs

"successfully-flat" when the sharpness hovers above $2/K$, rather than precisely hitting $2/K$. Another common characteristic shared between unsuccessfully flat experiments is that they all reach the value $2/K$ very early in the training (e.g., within 100 iterations), while successfully-flat ones only later (e.g., after 1200 iterations). Moreover, the second row of Figure 2 suggests that the points encountered in an unsuccessful training are approximately stationary. In fact, the norms of the gradients w.r.t. the inner layers suddenly drop to very small values (e.g., around $10^{-4}$), while those w.r.t. the bias of the last layer keep the training "barely alive". That is, without a meaningful gradient updating the parameters of the hidden layers, the feature extraction process cannot take place, and the corresponding accuracy remains low. Moreover, preliminary results show that for unsuccessfully-flat runs if one removes the bias of the last layer, GD collapses to a 0-network, a trivial stationary point that outputs 0 regardless of the input (see Section J). In the case of successfully-flat trainings, the norms of all the gradients are closer together and not as small as in the previous case, while the corresponding losses and accuracies are also better. Finally, all the globally-flat points have Hessians with both positive and negative eigenvalues. Consequently, the stationary points encountered by the unsuccessfully-flat trainings are saddle points.

Beyond verifying that $2/K$ is the global minimum of the sharpness, Lemma 3.1 states that the sub-Hessian w.r.t. the bias of the last layer $\nabla_b^2 f$ is diagonal with $K$ values at $2/K$. This guarantees that the largest $K$ eigenvalues *can* all originate from $\nabla_b^2 f$, as observed in Figure 2, but it does not clarify *how*. Given $\nabla_W^2 f$ from Lemma 3.1, we expect that most of the entries of the Hessian are close to 0. In fact, from the third row of Figure 2 we observe that the percentage of 0-outputs for at least one of the layers in the network is beyond 95% (sometimes close to 99%). Interestingly, even for the successfully-flat training runs this percentage is very high. While we leave to future work the investigation of what causes GD to enter globally-flat regions, some preliminary results suggest that this behavior may be connected to the neural collapse phenomenon (Papyan et al., 2020).

We conclude this section with one further observation whose origin will be also addressed in future work. For globally-flat points, the eigenvalues just below the top $K$ are remarkably smaller than $2/K$, but not negligible. In particular, they pair-wise sum to 0 and the resulting trace is $\approx 2 = K \cdot 2/K$.

### 3.2. Avoiding the Globally-Flat Points

In order to avoid the globally-flat points, we propose a new variation of NLS called NLS-ub that maintains the step size below the number of classes $K$. This choice is suggested by the semi-formal argument of Damian et al. (2022), which shows that the sharpness would be upper bounded by $2/\eta_k$.

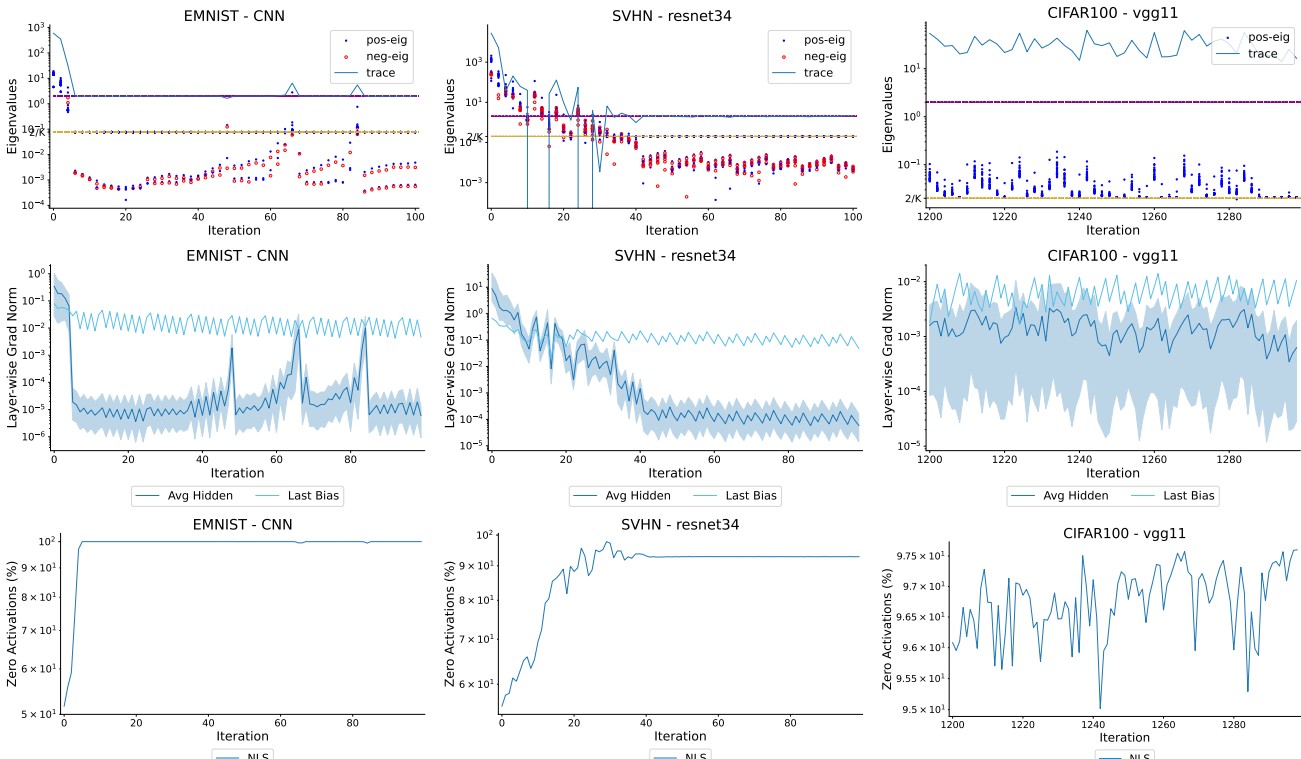

*Figure 2.* We plot the top 20 (30 for EMNIST) eigenvalues (Top Row), layer-wise gradient norm for the hidden layers (which we average across all hidden layers) and gradient norm of the bias parameters in the last layer (Middle Row), and the maximum per layer percentage of neural activations approximately equal to zero (Bottom Row) for the NLS method. This is repeated for 3 different models and 3 different datasets.

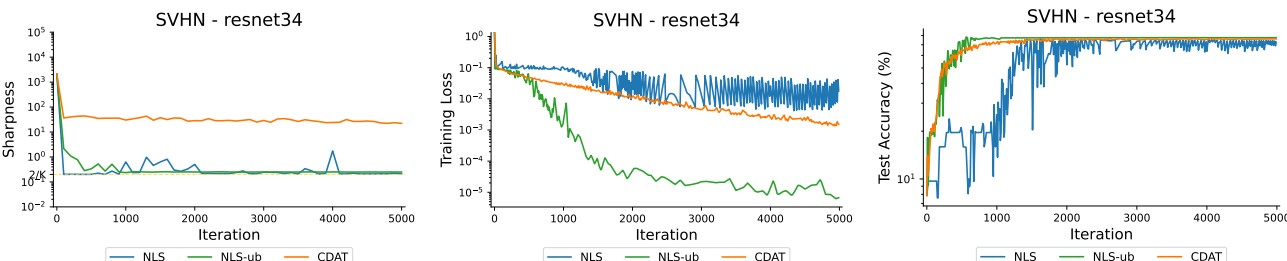

*Figure 3.* We plot the sharpness (Left), training loss (Middle), and test accuracy (Right) for three different methods. We compare gradient descent with the NLS and NLS-ub line searches as well as gradient descent with the CDAT step size selection. This experiment is performed for the resnet34 model on the SVHN dataset.

By this argument, if $\eta_k = K$, the sharpness is forced to hit its minimum of $2/K$, while if we maintain $\eta_k < K$, we allow GD to enter slightly-sharper valleys. We showcase the results of NLS-ub in Figure 3 (and Figures 32-39 in the Appendix). Similar to NLS, we see from the first plot of Figure 3 that NLS-ub also achieves low sharpness. However, NLS-ub never hits precisely $2/K$. Although NLS performs worse than CDAT, the training performance and the test accuracy of NLS-ub are often better than that of CDAT (see also Figures 32-39). Notice that given the small number of training examples (5000) in our experiments, larger experiments would be needed to confirm these con-

clusions. Nonetheless, we report the test accuracy of the methods to advocate for the following thesis: "flatter" is not always "better", especially if one compares points that do not achieve the same training loss (e.g., NLS vs NLS-ub).

### 3.3. On the Validity of Local Segment Smoothness

Assumption 2.2 holds for all twice continuously differentiable functions, but it may not hold everywhere for neural networks with ReLU activation functions. We assume that $f$ is differentiable almost everywhere, and check numerically whether the points encountered by GD satisfy local segment smoothness, and if so, for which val-

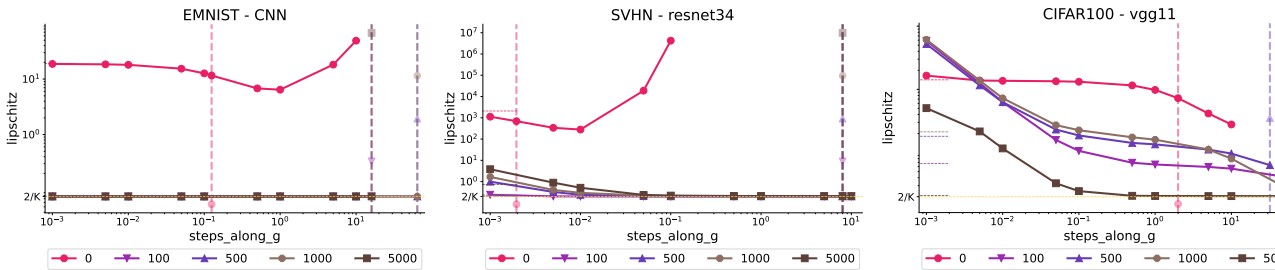

*Figure 4.* We plot the segment smoothness (2.1) for different steps along the gradient direction at varying training iterations for the NLS line search method. In addition, the vertical dashed lines correspond to the selected step size at each iteration. Finally, the horizontal dashed lines correspond to the sharpness at each iteration. This is repeated for 3 different models and 3 different datasets.

ues of $\eta_w$. In Figure 4, we plot the Lipschitz constant $l'(w_k, \eta) := \frac{\|\nabla f(w_k) - \nabla f(w_\eta)\|}{\|\eta \nabla f(w_k)\|}$ along the gradient line at various iterations (0, 100, 500, 1000, and 5000) achieved by NLS. These lower estimates of $l(w_k, \eta_{w_k})$ are calculated at various values of $\eta$ (from $10^{-3}$ to $10^1$) and at $\eta_k$, which is the step size yielded by NLS at the corresponding iteration $k$. The vertical dashed lines represent $\eta_k$ and are of the same color as the iteration $k$ at which they are selected. While the full analysis of Figures 4 is provided in Appendix I.4, the main conclusion is that Assumption 2.2 holds on the points we tested it for values of $\eta_w$ up to 0.1. For values beyond that, $l'(w_k, \eta)$ may be numerically unbounded. In other words, $f$ is not globally directionally smooth (Mishkin et al., 2024) for these neural networks. We provide additional experiments in Figures 40 and 41 in Appendix I.4.

## 4. Related Works

For locally Lipschitz non-convex functions outside of deep learning applications, nonmonotone line searches have been used to solve the step size selection problem in Kanzow & Mehlitz (2022) and De Marchi (2023). To the best of our knowledge, our work is the first aimed at finding the largest converging first-order steps through the use of extrapolation passes. In particular, this allows us to push to the limit the concept of *large* gradient steps and consequently find the maximum step size for gradient descent that still leads to convergence on classification problems.

While the idea of a smoothness assumption along the segment between $w_k$ and $w_{k+1}$ is not new to optimization, we are only aware of the work by Scheinberg et al. (2014) providing a similar point-wise definition of segment smoothness as in Assumption 2.2. However, their work only looks at the case of convex functions. To the best of our knowledge, local segment smoothness was never used in the context of deep learning before our work.

There are some lower bounds for the step size in the literature that are conceptually similar to Proposition 2.5. However, they are derived for convex (Scheinberg et al., 2014)

or strongly convex functions (Vaswani et al., 2022), and only for monotone line searches. Moreover, these papers do not specify how to select the largest step size such that (2) holds. From our analysis, Proposition 2.5 (or Proposition 2.11) seems achievable only if one makes use of extrapolation passes (or a Polyak initial step size). Proposition 2.8 is built upon the convex universal gradient method (UGM) by Nesterov (2015) and the previously unnoticed connection to nonmonotone line searches. This association allows the removal of the $\epsilon$ hyper-parameter in UGM, which is difficult to tune (Orabona, 2023). In fact, with nonmonotone line searches this hyperparameter is automatically selected via the adaptive $\epsilon_k = R_k - f(w_k)$.

## 5. Conclusion

In this work, we propose a formal definition of "large" step sizes and two first-order algorithms to find them. We show both in theory and in practice that these step sizes operate at the edge of stability. Contrary to expectations, we find that when employing step sizes that are too large, GD may encounter globally flat regions that can severely degrade performance. We further characterize these points as being saddle, which notably slow down convergence. We thus allow GD to take smaller step sizes, enter slightly sharper valleys, and successfully avoid globally-flat saddle points.

To conclude, we discuss some of the limitations of this work and the consequent research directions we plan to explore in future work. Despite the fact that Propositions 2.5 and 2.8 as well as Corollary 2.11 hold when replacing $f$ with a stochastic approximation in Assumption 2.2, it is unclear whether a stochastic version of Assumption 2.2 is sufficient to prove the convergence of SGD. Although we conducted preliminary experiments with other activation functions (e.g., leakyReLU, GeLU), we are not yet able to determine what causes GD with large step sizes to find globally-flat saddle points so efficiently. Starting with the work by Grippo et al. (1986), the evidence collected in the nonmonotone line search literature shows a clear numerical advantage of using these methods over their monotone

counterparts. However, an improved rigorous theoretical bound showing that nonmonotone line searches are faster is still missing. We aim to close this gap by exploiting our oscillating equality line searches starting with the simplified setting of logistic regression problems with separable data (Wu et al., 2024). In Figures 19-26, there are some cases in which the step sizes selected by NLS or PoNLS are significantly smaller than $2/K$. To achieve larger step sizes in these settings, one would need to increase the amount of nonmonotonicity beyond that of the nonmonotone line search proposed in Zhang & Hager (2004), but not as much as the nonmonotone term proposed in Grippo et al. (1986), as this method requires the global Lipschitz continuity of $f$.

## Acknowledgements

The work was partially supported by the Canada CIFAR AI Chair Program and NSERC Discovery Grant RGPIN-2022-036669. The first author is grateful to his soon-to-be mother of his first son and wife for suggesting the use of "Flatland" in the title. In truth, the decision was only final after the fourth author independently suggested the same title, proving once more that she is always right.

## Impact Statement

This paper presents work whose goal is to advance the field of Machine Learning. There are many potential consequences of our work, none which we feel must be specifically highlighted here.

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

## A. Algorithms

---

**Algorithm 1** Backtracking Line Search

---

**Input:** $w_k, d_k \in \mathbb{R}^n, c \in (0, 1), \delta \in (0, 1), \eta_{k,0} > 0, R_k$ a monotone or nonmonotone reference value
$l_k = 0$
$\eta_k = \eta_{k,0}$
**while** $f(w_k + \eta_k d_k) \leq R_k + c\eta_k \nabla f(w_k)^T d_k$ is False **do**
   $l_k = l_k + 1$
   $\eta_k = \delta^{l_k} \eta_{k,0}$
**end while**
**Return:** $\eta_k$

---

**Algorithm 2** Equality Line Search

---

**Input:** $w_k, d_k \in \mathbb{R}^n, c \in (0, 1), \delta \in (0, 1), \eta_{k,0} > 0, R_k$ a monotone or nonmonotone reference value
$l_k = 1$
$is\_equality = False$
**while** $is\_equality == False$ **do**
   $l_k = l_k - 1$
   $\eta_k = \delta^{l_k} \eta_{k,0}$
   **while** $f(w_k + \eta_k d_k) \leq R_k + c\eta_k \nabla f(w_k)^T d_k$ is False **do**
      $l_k = l_k + 1$
      $\eta_k = \delta^{l_k} \eta_{k,0}$
      $is\_equality = True$
   **end while**
**end while**
**Return:** $\eta_k$

---

**Algorithm 3** Line Search with a Polyak initial step size

---

**Input:** $w_k, d_k \in \mathbb{R}^n, c \in (0, 1), \delta \in (0, 1), c_p > 0, R_k$ a monotone or nonmonotone reference value and $f^*$
$l_k = 0$
$\eta_{k,0} = \frac{f(w_k) - f^*}{c_p || \nabla f(w_k)||^2}$
$\eta_k = \eta_{k,0}$
**while** $f(w_k + \eta_k d_k) \leq R_k + c\eta_k \nabla f(w_k)^T d_k$ is False **do**
   $l_k = l_k + 1$
   $\eta_k = \delta^{l_k} \eta_{k,0}$
**end while**
**Return:** $\eta_k$

---

**Algorithm 4** GD with a Equality Line Search

---

**Input:** $w_0 \in \mathbb{R}^n, \epsilon > 0, c \in (0, 1), \delta \in (0, 1), \eta_{0,0} > 0, \eta_{\max} > 0$
$k = 0$
**while** $\| \nabla f(w_k)\| > \epsilon$ **do**
   $\eta_k \leftarrow$ Algorithm 2 with initial step size $\eta_{k,0}$
   $w_{k+1} = w_k - \eta_k \nabla f(w_k)$
   $\eta_{k+1,0} = \eta_k$
   $k = k + 1$
**end while**

---

**Algorithm 5** GD with a Line Search with a Polyak initial step size

**Input:** $w_0 \in \mathbb{R}^n, \epsilon > 0, c \in (0,1), \delta \in (0,1), f^*, c_p > 0, \eta_{\max} > 0$
$k = 0$
**while** $\| \nabla f(w_k) \| > \epsilon$ **do**
    $\eta_k \leftarrow$ Algorithm 3
    $w_{k+1} = w_k - \eta_k \nabla f(w_k)$
    $k = k + 1$
**end while**

## B. Mathematical Details and Proofs

In this chapter, we give the proof of the results in the main paper. The appendix is organized as follows:

- B.1 gives a proof of the termination of the equality line search Algorithm 2. The convergence of Algorithm 4 is then given in B.5.
- B.2 and B.3 show the Lipschitz and Hölder Awareness of the Algorithms 1 and 2. This is followed by some discussion on the benefits of Algorithm 2.
- B.4 draws connections to sharpness, i.e. proving Corollary 2.10. This is extended in B.6 for Algorithm 3.
- B.7 then gives a formal version of Lemma 3.1 along with some extra results and discussion.

### B.1. Equality Line Searches and Their Finite Convergence

We first remind the reader of the original definition of the Zhang & Hager (2004) line search:

$$f(w_k - \eta_k \nabla f(w_k)) \leq R_k - c\eta_k \|\nabla f(w_k)\|^2, \quad \text{with } c \in (0,1),$$
$$R_k = \frac{\xi Q_k R_{k-1} + f(w_k)}{Q_{k+1}}, \quad Q_{k+1} = \xi Q_k + 1, \quad \xi \in (0,1). \tag{8}$$

We start with two useful bounds on $Q_{k+1}, R_k$ from the above definition observed by (Zhang & Hager, 2004), which will enable us to prove the convergence of the equality line search.

**Lemma B.1.** *From the definition of $Q_k$ in (8), it follows*

$$1 \leq \xi Q_k + 1 \leq \frac{1}{1 - \xi},$$

$$Q_{k+1} \leq k + 2, \tag{9}$$

*and*

$$\frac{\xi Q_k}{\xi Q_k + 1} \leq \xi.$$

*Proof.* From the definition of $Q_{k+1}$ we have

$$1 \leq \xi Q_k + 1 =: Q_{k+1} = 1 + \sum_{j=0}^{k} \xi^{j+1} \leq \sum_{j=0}^{\infty} \xi^j = \frac{1}{1-\xi},$$

but from $\xi \in (0,1)$ also

$$Q_{k+1} = 1 + \sum_{j=0}^{k} \xi^{j+1} \leq k + 2.$$

Thus we have

$$\frac{\xi Q_k}{\xi Q_k + 1} = \frac{\xi Q_k + 1 - 1}{\xi Q_k + 1} = 1 - \frac{1}{\xi Q_k + 1} \leq 1 - \frac{1}{\frac{1}{1-\xi}} = \xi.$$

which implies (B.1) and concludes the proof. $\qquad\square$

**Lemma B.2.** *Let $A_k := \frac{1}{k+1} \sum_{i=0}^{k} f(w_i)$. From the definition of $R_k$ and $Q_k$ in (8), it follows that $f(w_k) \leq R_k \leq A_k$.*

*Proof.* Defining $D_k : \mathbb{R} \to \mathbb{R}$ by

$$D_k(t) = \frac{tR_{k-1} + f(w_k)}{t+1},$$

we have

$$D'_k(t) = \frac{R_{k-1} - f(w_k)}{(t+1)^2}.$$

It follows from (8) that $f(w_k) \leq R_{k-1}$, which implies that $D'_k(t) \geq 0$ for all $t \geq 0$. Hence, $D_k$ is nondecreasing, and $f(w_k) = D_k(0) \leq D_k(t)$ for all $t \geq 0$. In particular, taking $t = \xi Q_{k-1}$ gives

$$f(w_k) = D_k(0) \leq D_k(\xi Q_{k-1}) = R_k.$$

The upper bound $R_k \leq A_k$ is proved by induction. For $k = 0$, this holds by the initialization $R_0 = f(w_0)$. Now assume that $R_j \leq A_j$ for all $0 \leq j < k$.
Since $D_k$ is monotone nondecreasing, (9) implies that

$$R_k = D_k(\xi Q_{k-1}) = D_k(Q_k - 1) \leq D_k(k).$$

By the induction step,

$$D_k(k) = \frac{kR_{k-1} + f(w_k)}{k+1} \leq \frac{kA_{k-1} + f(w_k)}{k+1} = A_k,$$

which concludes the induction argument. $\qquad\square$

**Lemma B.3.** *Let $f \in C^1(\mathbb{R}^n)$ be lower bounded by $f^*$. Let $w_k \in \mathbb{R}^n$ and $d_k \in \mathbb{R}^n$ a corresponding descent direction. Then the equality line search Algorithm 2 with reference value $R_k = f(w_k)$ or $R_k$ as defined in (8) terminates in a finite number of steps with step size $\eta_k > 0$ satisfying both*

*(a)* $f(w_k + \eta_k d_k) \leq R_k + c\eta_k \nabla f(w_k)^T d_k$;
*(b)* $f(w_k + \frac{\eta_k}{\delta} d_k) > R_k + c\frac{\eta_k}{\delta} \nabla f(w_k)^T d_k$.

*Proof.* There are two reasons the Algorithm could not converge, that is either the internal while loop does not terminate, or the external while loop does not terminate.

Let us first consider the case of the internal while loop not terminating, this implies $l \to \infty$. And for every $l > 0$ the stopping criterion (3) is met, i.e.

$$\frac{f(w_+ + \eta_{k,0}\delta^l d_k) - f(w_k)}{\eta_{k,0}\delta^l} \geq c\nabla f(w_k)^T d_k + \frac{\varepsilon_k}{\eta_{k,0}\delta^l} \geq c\nabla f(w_k)^T d_k.$$

As $\delta < 0$, with $l \to \infty$ the left hand side converges to the directional derivative of f along $d_k$. In particular, we get

$$\nabla f(w_k)^T d_k \geq c \nabla f(w_k)^T d_k,$$

which is a contradiction, as $\nabla f(w_k)^T d_k < 0$ and $c < 1$.

Let us now consider the case where the external while of Algorithm 2 does not converge. This implies the stopping criterion (2) is always satisfied for $l \to -\infty$, that is

$$f(w_k + \eta_{k,0}\delta^l d_k) \leq R_k + c\eta_{k,0}\delta^l \nabla f(w_k)^T d_k \leq f(w_0) + c\eta_{k,0}\delta^l \nabla f(w_k)^T d_k,$$

where the second inequality follows from the fact that in both cases (i.e., $R_k = f(w_k)$ and $R_k$ defined as in (8)) $R_k$ is upper bounded by $f(w_0)$ (by Lemma B.2 in the case of $R_k$.) Given $\delta < 1$ we have that $\lim_{l \to -\infty} \delta^l = +\infty$ and $\lim_{l \to -\infty} \eta_{k,0}\delta^l \nabla f(w_k)^T d_k = -\infty$, which contradicts the inequality above as $f$ is lower bounded by $f^*$.
Finally, because of the steps of Algorithm 2, the stopping criterion (2) has to be false at least once before it terminates, which implies that both $(a)$ and $(b)$ are true for $\eta_k$. $\qquad\square$

## B.2. Proof of Proposition 2.5

This subsection is dedicated to the proof of Proposition 2.5. To this end, we first derive a version of the classical Descent Lemma and use this to derive a version of Proposition 2.5 for the Backtracking Line Search, Algorithm 1. Subsequently, we use this to derive the final result.

Before proceeding, we present a small lemma showing that, without loss of generality, one can assume $\eta_w \geq \frac{2}{l(w,\eta_w)}$ in Assumption 2.2.

**Lemma B.4.** *Suppose $f$ satisfies Assumption 2.2, then we can additionally assume $\eta_w \geq \frac{2}{l(w,\eta_w)}$.*

*Proof.* We argue by contradiction. Suppose there exists $w \in \mathbb{R}^n$ and $\eta_w > 0$ such that $l(w,\eta_w)$ is bounded, yet $\eta_w \leq \frac{2}{l(w,\eta_w)}$.

We define the continuous function $h : \mathbb{R}_{>0} \to \mathbb{R}$, $h(\eta) = \eta - \frac{2}{l(w,\eta)}$. By assumption $h(\eta_w) = \eta_w - \frac{2}{l(w,\eta_w)} < 0$. On the other hand,

$$h\left(\frac{2}{l(w,\eta_w)}\right) = \frac{2}{l(w,\eta_w)} - \frac{2}{l\left(w, \frac{2}{l(w,\eta_w)}\right)} \leq \frac{2}{l(w,\eta_w)} - \frac{2}{l(w,\eta_w)} = 0,$$

using $l(w,\eta)$ is non-decreasing in $\eta$.

Therefore, we can instead choose an $\tilde{\eta}_w \geq \frac{2}{l(w,\tilde{\eta}_w)}$ (for instance $\tilde{\eta}_w = \frac{2}{l(w,\eta_w)}$), to satisfy Assumption 2.2. $\square$

**Lemma B.5** (Descent Lemma on the segment). *Let $f \in C^1(\mathbb{R}^n)$, and let Assumption 2.2 hold at $w_k$ with a local Lipschitz constant $l(w_k, \eta_{w_k})$. Set $w_{k,\eta} := w_k - \eta \nabla f(w_k)$ for $\eta \in (0, \eta_{w_k}]$, then*

$$f(w_{k,\eta}) \leq f(w_k) + \nabla f(w_k)^T (w_{k,\eta} - w_k) + \frac{l(w_k, \eta_{\eta_k})}{2} \|w_{k,\eta} - w_k\|^2.$$

*Proof.* The fundamental theorem of calculus implies

$$f(w_{k,\eta}) = f(w_k) + \int_0^1 \nabla f((1-t)w_k + tw_{k,\eta})^T (w_{k,\eta} - w_k) \, dt$$

$$= f(w_k) + \int_0^1 \nabla f((1-t)w_k + tw_{k,\eta})^T (w_{k,\eta} - w_k) - \nabla f(w_k)^T (w_{k,\eta} - w_k) \, dt$$

$$\quad + \nabla f(w_k)^T (w_{k,\eta} - w_k)$$

$$\leq f(w_k) + \int_0^1 \| \nabla f((1-t)w_k + tw_{k,\eta}) - \nabla f(w_k) \| \cdot \|w_{k,\eta} - w_k\| \, dt$$

$$\quad + \nabla f(w_k)^T (w_{k,\eta} - w_k)$$

$$\leq f(w_k) + \int_0^1 l(w_k, \eta_{w_k}) \|t(w_{k,\eta} - w_k)\| \cdot \|w_{k,\eta} - w_k\| \, dt + \nabla f(w_k)^T (w_{k,\eta} - w_k)$$

$$= f(w_k) + l(w_k, \eta_{w_k}) \|w_{k,\eta} - w_k\|^2 \cdot \left.\frac{t^2}{2}\right|_0^1 + \nabla f(w_k)^T (w_{k,\eta} - w_k)$$

$$= f(w_k) + \nabla f(w_k)^T (w_{k,\eta} - w_k) + \frac{l(w_k, \eta_{w_k})}{2} \|w_{k,\eta} - w_k\|^2,$$

where the second inequality follows from Assumption 2.2, which implies

$$\|\nabla f(w) - \nabla f(w_\xi)\| \leq l(w,\eta) \|w - w_\xi\| \qquad \forall \xi \in [0, \eta].$$

$\square$

**Proposition B.6.** *Let $f \in C^1(\mathbb{R}^n)$, and let Assumption 2.2 hold at $w_k$ with a local Lipschitz constant $l(w_k, \eta_{w_k})$. Let $\eta_k$ be the result of a backtracking procedure employed to satisfy (3), then*

$$\begin{cases} \eta_k = \eta_{k,0} & \text{if } l_k = 0, \\ \eta_k \geq \frac{2\delta(1-c)}{l(w_k, \eta_{w_k})} & \text{if } l_k > 0. \end{cases} \tag{10}$$

*Proof.* We first distinguish between the following two cases:

**1st case:** There has not been a cut, hence $l_k = 0$, then by the algorithm $\eta_k = \eta_{k,0}$.

**2nd case:** We have cut at least once, so $l_k > 0$. This $\frac{\eta_k}{\delta}$ violates the termination rule (3). We denote the reference value as $R_k := f(w_k) + \epsilon_k$, with $\epsilon_k \geq 0$, and $g_k := \nabla f(w_k)$. As a first step, we apply the Descent Lemma B.5 on $w_k - \eta g_k$, with $\eta \in (0, \eta_{w_k}]$:

$$
\begin{aligned}
f(w_k - \eta g_k) &\leq f(w_k) + g_k^T(w_k - \eta g_k - w_k) + \frac{\eta^2 l(w_k, \eta_{w_k})}{2}\|g_k\|^2 \\
&= f(w_k) - \left(\eta - \frac{\eta^2 l(w_k, \eta_{w_k})}{2}\right)\|g_k\|^2.
\end{aligned}
\tag{11}
$$

Now we need to distinguish between the two scenarios: $\frac{\eta_k}{\delta} \in (0, \eta_{w_k}]$ or $\frac{\eta_k}{\delta} > \eta_{w_k}$.

If $\frac{\eta_k}{\delta} \in (0, \eta_{w_k}]$, we combine (11) with the termination rule (3), i.e., $f\left(w_k - \frac{\eta_k}{\delta} g_k\right) > R_k - c\frac{\eta_k}{\delta}\|g_k\|^2$, to establish

$$
f(w_k) - \left(\frac{\eta_k}{\delta} - \frac{\eta_k^2 l(w_k, \eta_{w_k})}{2\delta^2}\right)\|g_k\|^2 \geq R_k - c\frac{\eta_k}{\delta}\|g_k\|^2 \geq f(w_k) - c\frac{\eta_k}{\delta}\|g_k\|^2,
\tag{12}
$$

which leads to (10).

If $\frac{\eta_k}{\delta} > \eta_{w_k}$, then by Assumption 2.2 $\eta_{w_k} \geq \frac{2}{l(w, \eta_{w_k})}$. Combining this with $1 > 1 - c$, yields the desired inequality $\eta_k > \frac{2\delta(1-c)}{l(w, \eta_{w_k})}$. □

In light of this result on the backtracking line search, Proposition 2.5, which we restate here for the reader's convenience, follows directly.

**Proposition B.7** (Lipschitz-Awareness)**.** *Let $f$ be segment smooth and let $\eta_k$ be the result of an equality line search (Algorithm 2) employed to satisfy (3) with $\epsilon_k \geq 0$, then*

$$
\eta_k \geq \frac{2\delta(1-c)}{l(w_k, \eta_{w_k})}.
$$

*Proof.* The proof is analogue to that of Proposition B.6.

For the equality line search, we always make one cut. Therefore, we do not need the distinction between $l_k > 0$ and $l_k = 0$.

□

**Corollary B.8.** *Given $\eta_{k,0} > 0, c, \delta \in (0, 1), w_k \in \mathbb{R}^n$, the equality line search, Algorithm 2, terminates in at most*

$$
\log_{\frac{1}{\delta}}\left(\frac{\eta_{k,0} l(w_k, \eta_{w_k})}{2\delta(1-c)}\right) \text{ iterations.}
$$

*Proof.* From Lemma B.3 the line search algorithm terminates. From Proposition 2.5, we have

$$
\eta_{k,0}\delta^{l_k} = \eta_k \geq \frac{2\delta(1-c)}{l(w_k, \eta_{w_k})} \Leftrightarrow l_k \leq \log_{\frac{1}{\delta}}\left(\frac{\eta_{k,0} l(w_k, \eta_{w_k})}{2\delta(1-c)}\right).
$$

□

Finally, we give a precise statement of how large step-sizes arise in non-monotone line searches.

**Lemma B.9.** *Let $f \in C^2(\mathbb{R}^n)$ satisfy Assumption 2.2. Then for any $w_k$ there exist $c < 1, \delta > 0$ such that the equality line search of Algorithm 2, using a non-monotone reference value $R_k$, returns a step-size $\eta_k$ satisfying Definition 2.3 with $P = 0$.*

*Proof.* Since $R_k$ is a non-monotone reference value, $R_k = f(w_k) + \varepsilon_k$ with $\varepsilon_k > 0$, see also (3) where this concept is introduced.

Starting from (12) yields

$$\eta_k \geq \frac{2(1-c)\delta}{l(w_k, \eta_{w_k})} + \varepsilon_k \frac{2\delta^2}{\eta_k l(w_k, \eta_{w_k}) \|g_k\|_2^2}. \tag{13}$$

Suppose for all $c, \delta \in (0,1)$ $\eta_k \leq \frac{2}{l(w_k, \eta_{w_k})}$. Then $\frac{1}{\eta_k l(w_k, \eta_{w_k})} \geq \frac{1}{2}$, so (13) implies

$$\eta_k \geq \frac{2(1-c)\delta}{l(w_k, \eta_{w_k})} + \varepsilon_k \frac{\delta^2}{\|g_k\|_2^2}. \quad \text{Taking the limit} \lim_{\delta \to 1^-, c \to 0^+} \eta_k \geq \frac{2}{l(w_k, \eta_{w_k})} + \epsilon_k \|g_k\|_2^{-2} > \frac{2}{l(w_k, \eta_{w_k})}.$$

Which implies the existence $c > 0$, $\delta < 1$ such that $\eta_k$ is bounded below by $\frac{2}{l(w_k, \eta_{w_K})}$. $\qquad\square$

### B.3. Proof of Proposition 2.8

Fist we argue why we can assume wlog that $\eta_w \geq \frac{2}{dM_\nu(w, \eta_w)^{\frac{2}{1+\nu}}}$ in Assumption 2.7.

**Lemma B.10.** *Suppose f is Hölder segment smooth. Then for some fixed $d > 0$ and all $w \in \mathbb{R}^n$ we can assume* $\eta_w \geq \frac{2}{cM_\nu(w, \eta_w)^{\frac{2}{1+\nu}}}$.

*Proof.* Define $h : \mathbb{R}_+ \to \mathbb{R}$, as $h(\eta) = \eta - \frac{2}{dM_\nu(w, \eta)^{\frac{2}{1+\nu}}}$.

In the first case $h(\eta_w) \geq 0$ we are done.

If, however, $h(\eta_w) < 0$, then as $M_\nu(w, \eta)$ is monotonically increasing in $\eta$, we have $M_\nu\left(w, \frac{2}{dM_\nu(w, \eta_w)^{\frac{2}{1+\nu}}}\right) \geq M_\nu(w, \eta_w)$

and therefore $h\left(M_\nu(w, \eta_w)^{\frac{2}{1+\nu}}\right) \geq 0$. So we can choose $M_\nu(w, \eta_w)^{\frac{2}{1+\nu}}$ as $\eta_w$ instead. $\qquad\square$

Now we give a Descent Lemma for the Hölder segment smoothness.

**Lemma B.11** (Hölder Descent Lemma on the segment). *Let f satisfy Assumption 2.7. For all $w \in \mathbb{R}^n$ and $\eta \in [0, \eta_w]$*

$$f(w_\eta) - f(w) \leq \frac{M_\nu(w, \eta_w)}{\nu + 1} \|w - w_\eta\|^{\nu+1} + \nabla f(w)^T (w_\eta - w)$$

$$= \frac{M_\nu(w, \eta_w)}{\nu + 1} \eta^{\nu+1} \|\nabla f(w)\|^{\nu+1} - \eta \|\nabla f(w)\|^2$$

*Proof.* Works in the same manner as the proof of Lemma B.5, by replacing the Lipschitz smoothness with the Hölder smoothness

$$f(w_\eta) - f(w) = \int_0^1 \nabla f((1-t)w + tw_\eta)^T (w_\eta - w)\, dt$$

$$= \int_0^1 (\nabla f((1-t)w + tw_\eta) - \nabla f(w))^T (w_\eta - w)\, dt + \nabla f(w)^T (w_\eta - w)$$

$$\leq \int_0^1 \|\nabla f((1-t)w + tw_\eta) - \nabla f(w)\| \, \|w - w_\eta\|\, dt - \eta \|\nabla f(w)\|^2$$

$$\leq \int_0^1 M_\nu(w, \eta_w) \|t(w_\eta - w)\|^\nu \, \|w - w_\eta\|\, dt - \eta \|\nabla f(w)\|^2$$

$$= M_\nu(w, \eta_w) \|w - w_\eta\|^{\nu+1} \int_0^1 t^\nu dt - \eta \|\nabla f(w)\|^2$$

$$= \frac{M_\nu(w, \eta_w)}{\nu + 1} \|w - w_\eta\|^{\nu+1} - \eta \|\nabla f(w)\|^2,$$

where we have used that the definition of the local Hölder smoothness yields

$$\|\nabla f(w) - \nabla f(w_{k,\eta})\| \leq M_\nu(w, \eta_w) \|w - w_{k,\eta}\|^\nu \quad \forall \eta \in [0, \eta_w].$$

$\qquad\square$

Next we translate this Descent Lemma into a Descent Lemma with a squared gradient norm to compare against the conditions of the line search algorithm.

**Lemma B.12.** *Let $f$ satisfy Assumption 2.7. We define $\alpha(M_\nu(w, \eta_w), \varepsilon) := M_\nu(w, \eta_w)^{\frac{2}{1+\nu}} \cdot \left(\frac{1-\nu}{1+\nu} \cdot \frac{1}{\varepsilon}\right)^{\frac{1-\nu}{1+\nu}}$. Then for all $w \in \mathbb{R}^n$ and $0 \leq \nu \leq 1$:*

$$f(w_\eta) \leq f(w) + \frac{\alpha(M_\nu(w, \eta_w), \varepsilon)}{2}\eta^2 \|\nabla f(w)\|^2 - \eta \|\nabla f(w)\|^2 + \frac{\varepsilon}{2} \quad \forall \eta \in [0, \eta_w]. \tag{14}$$

*Proof.* We want to point out that the proof strategy is heavily follows (Nesterov, 2015). This is done by Young's inequality, i.e. for all $a, b \geq 0, 1 < p, q < \infty$ $\frac{1}{p} + \frac{1}{q} = 1$

$$\frac{a^p}{p} + \frac{b^q}{q} \geq a \cdot b. \tag{15}$$

We now choose $p = \frac{2}{1+\nu}, q = \frac{2}{1-\nu}, a = \|w - w_\eta\|^{\nu+1}, b = \left(\frac{1+\nu}{1-\nu}\frac{\varepsilon}{M_\nu(w, \eta_w)}\right)^{\frac{1-\nu}{1+\nu}}$. Then for all $t, s \geq 0$

$$\|w - w_\eta\|^{\nu+1} \leq (\nu+1)\frac{\|w - w_\eta\|^2}{2b} + (1-\nu)\frac{b^{q-1}}{2} \tag{16}$$

$$= (\nu+1)\frac{\|w - w_\eta\|^2}{2}\frac{\alpha(M_\nu(w, \eta_w), \varepsilon)}{M_\nu(w, \eta_w)} + \frac{1+\nu}{2}\frac{\varepsilon}{M_\nu(w, \eta_w)}. \tag{17}$$

Substituting this into the result of Lemma B.11 we get the claimed inequality. $\qquad \square$

Now we give a version of Proposition 2.8 for Algorithm 1.

**Proposition B.13.** *Let $f$ satisfy Assumption 2.7 with $d = \min\{1, e^{-\frac{f(x_0)-f^*}{2e}}\}$ in $w_k$ and let $\eta_k$ be the result of a backtracking procedure employed to satisfy (3) with $\epsilon_k > 0$, then*

$$\begin{cases} \eta_k = \eta_{k,0} & \text{if } l_k = 0, \\ \eta_k \geq \frac{2\delta(1-c)}{\alpha(M_\nu(w_k, \eta_{k_w}), 2\epsilon_k)} & \text{if } l_k > 0, \end{cases}$$

*with*

$$\alpha(M_\nu(w_k, \eta_{w_k}), \epsilon_k) := M_\nu(w_k, \eta_{w_k})^{\frac{2}{1+\nu}} \cdot \left(\frac{1-\nu}{1+\nu} \cdot \frac{1}{\epsilon_k}\right)^{\frac{1-\nu}{1+\nu}}.$$

*Proof.* This works similarly to the proof of Proposition 2.5. If $l_k = 0$ the result is immediate.

So, suppose we have at least one cut ($l_k > 0$), then by the stopping criterion (3)

$$f\left(w_k - \frac{\eta_k}{\delta}g_k\right) > f(w_k) - c\frac{\eta_k}{\delta}\|g_k\|^2 + \epsilon_k. \tag{18}$$

On the other hand Lemma B.12 yields for all $\eta \leq \eta_{w_k}$ that

$$f\left(w_k - \eta g_k\right) \leq f(w_k) - \eta\left(1 - \frac{\alpha(m_n u(w_k, \eta_{k,0}, 2\epsilon_k)}{2}\eta\right)\|g_k\|^2 + \epsilon_k. \tag{19}$$

Now we need to distinguish between the cases where $\frac{\eta_k}{\delta} \leq \eta_{w_k}$ is satisfied or not. In the first case combining (18) with (19) yields

$$f(w_k) + \epsilon_k - c\frac{\eta_k}{\delta}\|g_k\|^2 \leq f(w_k) - \frac{\eta_k}{\delta}\left(1 - \frac{\alpha(M_\nu(w_k, \eta_{w_k}), 2\epsilon_k)}{2}\frac{\eta_k}{\delta}\right)\|g_k\|^2 + \epsilon_k,$$

which results in

$$\eta_k \geq \frac{2\delta(1-c)}{\alpha(M_\nu(w_k, \eta_{k,0}), 2\epsilon_k)}.$$

If $\frac{\eta_k}{\delta} > \eta_{w_k}$, then by assumption $\frac{\eta_k}{\delta} \geq \frac{2}{dM_\nu(w_k, \eta_{w_k})^{\frac{2}{1+\nu}}}$.

**Claim:** $dM_\nu(w_k, \eta_{w_k})^{\frac{2}{1+\nu}} \leq \alpha(M_\nu(w_k, \eta_{k,0}), 2\varepsilon_k)$. This will directly yield the hypothesis.

Proof of the claim. From the definition $\alpha(M_\nu(w, \eta), 2\varepsilon_k) = M_\nu(w, \eta)^{\frac{2}{1+\nu}} \cdot \left( \frac{1-\nu}{1+\nu} \cdot \frac{1}{2\varepsilon_k} \right)^{\frac{1-\nu}{1+\nu}}$, $\nu \in [0, 1]$. So we just need

to show $d \leq \left( \frac{1-\nu}{1+\nu} \cdot \frac{1}{2\varepsilon_k} \right)^{\frac{1-\nu}{1+\nu}}$. To this end, define the function $g : [0, 1] \to \mathbb{R}$, $g(t) = \ln\left( \frac{t}{2\epsilon} \right)^t = t \ln\left( \frac{t}{2\varepsilon_k} \right)$ and see

$\left( \frac{1-\nu}{1+\nu} \cdot \frac{1}{2\varepsilon_k} \right)^{\frac{1-\nu}{1+\nu}} = \exp\left( g\left( \frac{1-\nu}{1+\nu} \right) \right)$. Note that since $g$ is not properly defined on the left end of the interval $[0, 1]$, we use the continuous extension $g(0) = \lim_{t \to 0^+} g(t)$ from here on.

Notice that the correspondence $t = \frac{1-\nu}{1+\nu}$ is one-to-one and hence minimizing $g$ w.r.t. $t$ is equivalent to minimizing the

expression $\left( \frac{1-\nu}{1+\nu} \cdot \frac{1}{2\varepsilon_k} \right)^{\frac{1-\nu}{1+\nu}}$ w.r.t. $\nu$. We calculate the derivatives

$$g'(t) = \ln\left( \frac{t}{2\varepsilon_k} \right) + 1, \quad g''(t) = \frac{1}{t},$$

and notice that $g$ is convex, hence it will obtain its minimum either on the boundaries or at a critical point.

$$g'(t) = 0 \Leftrightarrow t = \frac{2\varepsilon_k}{e}.$$

This leaves us with three possible candidates for the minimum of $\exp(g(t))$

$$\exp(g(0)) = 1, \quad \exp(g(1)) = \frac{1}{2\varepsilon_k}, \quad \exp\left( g\left( \frac{2\varepsilon_k}{e} \right) \right) = e^{-\frac{2\varepsilon_k}{e}}.$$

Now it is easy to see that $\frac{1}{x} \geq e^{-\frac{x}{e}}$ and $\varepsilon_k \leq f(x_0) - f^*$, hence $g(t) \geq \min\{1, e^{-2\frac{f(x_0)-f^*}{e}}\} = d$, which concludes the claim by observing $c(1 - \delta) < 1$ ☐

Now we are ready to prove Proposition 2.8, which we restate here for convenience.

**Proposition B.14.** *Let $f$ be Hölder segment smooth with $d = \min\{1, e^{-2\frac{f(x_0)-f^*}{e}}\}$ and let $\eta_k$ be the result of an equality line search employed to satisfy (3) with $\epsilon_k > 0$, then*

$$\eta_k \geq \frac{2\delta(1-c)}{\alpha(M_\nu(w_k, \eta_{w_k}), 2\epsilon_k)}. \tag{20}$$

*Proof.* This follows directly from Proposition B.13. As Algorithm 2 will always yield $l_k > 0$. ☐

Propositions B.6 and B.13 clarify the importance of the initial step size $\eta_{k,0}$, a value that has often played a secondary role in the optimization literature (Nocedal & Wright, 2006; Vaswani et al., 2019; Fan et al., 2023). For line search methods such as Algorithm 1, this parameter directly controls the final step size, i.e., $\eta_k = \eta_{k,0}\delta^{l_k}$. In particular, when no backtracks are performed ($l_k = 0$), the final and the initial step sizes coincide. Beyond that, if $\eta_{k,0}$ is too small (e.g., $\eta_{k,0} \ll \frac{1}{l(w_k, \eta_{w_k})}$), $\eta_{k,0}$ will not be reduced by the line search procedure and GD will ultimately behave exactly like a GD with a small constant step. In the context of edge of stability, this means that the iterates may never hit the nonmonotone phase (Cohen et al., 2021).

To achieve large enough step sizes and ensure that $\eta_k$ is lower bounded by the reciprocal of the Lipschitz/Hölder constant for all $k$, our idea is to design a line search that forces a cut on each iteration. To obtain this, equality line searches combine backtrack and extrapolation passes. In particular, if no cuts are performed, the step size is doubled until the line search condition is false (see Algorithm 2 for details). With the Definition 2.2 and lower boundedness of $f$, the procedure provably terminates (see Lemma B.3) and yields a step size for which the corresponding line search condition (either (2) or (3)) is approximately satisfied with equality. Notice that thanks to the finite convergence of the new line searches, the corresponding full-batch GD inherits the result of convergence for continuously differentiable functions that aren't necessarily globally $L$-smooth.

In Propositions B.6 and B.13, we have shown that the classical line searches are *often* (when $l_k > 0$) Lipschitz-aware, and when nonmonotone, also Hölder-aware. Thanks to the Propositions 2.5 and 2.8, we instead show that the step size $\eta_k$

yielded by the new equality line searches is *always* ($\forall k$) lower bounded by the reciprocal of the maximum sharpness along the segment $[w, w_{\eta_w}]$.

### B.4. Proof of Corollary 2.10

**Notation:** Recall that for a symmetric matrix $A \in \mathbb{R}^{n \times n}$, we denote its eigenvalues by

$$|\lambda_1(A)| \geq |\lambda_2(A)| \geq \cdots \geq |\lambda_n(A)|,$$

ordered by non-increasing absolute value.

The aim of this section is to prove Corollary 2.10 and therefore obtain a lower bound for the step size in terms of the sharpness at convergence, i.e. the maximal eigenvalue of the hessian at the limiting point $w^*$.

We recall a well known fact connecting the eigenvalues to the spectral norm.

**Lemma B.15.** *For a real symmetric matrix $A \in \mathbb{R}^{n \times n}$*

$$\|A\|_2 = |\lambda_1(A)|.$$

We start by connecting the segment smoothness (Definition 2.1) to the maximal (in absolute value) eigenvalue along the segment.

**Lemma B.16.** *Let $f$ be twice continuously differentiable and satisfy Assumption 2.2. Let $\eta_k$ be generated by Algorithm 2 and denote $H_\eta(w_k) := \nabla^2 f(w_k - \eta \nabla f(w_k))$ with $\eta \in [0, \eta_w]$, then*

$$\eta_k \geq \frac{2(1-c)\delta}{\max\limits_{\eta \in [0, \eta_w]} |\lambda_1(H_\eta(w_k))|}.$$

*Proof.* As $f$ is twice continuously differentiable, we can apply the mean value theorem to the gradient to see

$$l(w, \eta_w) = \sup_{\eta \in (0, \eta_w)} \frac{\| \nabla f(w_{k,\eta}) - \nabla f(w)\|}{\|w_{k,\eta} - w\|} \leq \sup_{\eta \in (0, \eta_w]} \frac{\|H_\eta(w_k)\, \nabla f(w)\|}{\|\nabla f(w)\|}$$
$$\leq \sup_{\eta \in (0, \eta_w]} \|H_\eta(w_k)\|_2 \leq \max_{\eta \in [0, \eta_{w_k}]} |\lambda_1(H_\eta(w_k))|,$$

where we have used that the Hessian is symmetric for the last inequality.

Finally, combining this result with the findings of Propsition 2.5, i.e.

$$\eta_k \geq \frac{2\delta(1-c)}{l(w_k, \eta_{w_k})},$$

yields the thesis. $\qquad\square$

Now we are ready to prove the main result, by exploiting the convergence and the continuity of the Hessian. We restate the result here for convenience.

**Corollary B.17.** *Let $f \in C^2(\mathbb{R}^n)$. Let $\{w_k\}$ be generated by GD where $\eta_k$ is generated by Algorithm 2. Let $f$ satisfy Assumption 2.2 on $w_k \; \forall k > \bar{k}$. Assume that $\{w_k\}$ converges to $w^*$. Then $\forall \varepsilon > 0, \exists \hat{k} > \bar{k} :$*

$$\eta_k \geq \frac{2(1-c)\delta}{|\lambda_1(H(w^*))| + \varepsilon} \quad \forall k \geq \hat{k}.$$

*Proof.* By assumption we know that $\lim_{k \to \infty} w_k = w^*$ and $\lim_{k \to \infty} \|\nabla f(w_k)\| = 0$, therefore $\lim_{k \to \infty} w_k - \hat{\eta}_k \nabla f(w_k) = w^*$, where $\hat{\eta}_k := \operatorname{argmax}_{\eta \in [0, \eta_{w_k}]} \lambda_1(H_\eta(w_k))$. This implies $H_{\hat{\eta}_k}(w_K)$ converges to $H(w^*)$ as $k \to \infty$. Then, by continuity of the Hessian of $f$, for every $\varepsilon > 0$ there exists an $\hat{k} > \bar{k}$ s.t. $\|H(w^*) - H_{\hat{\eta}_k}(w_k)\|_2 \leq \varepsilon$.

Next we use Weyl's inequality (Horn & Johnson, 2012), for singular values of matrices $A$, $B$, which tells us $\sigma_k(A) - \sigma_k(B) \leq \sigma_1(A - B) = \|A - B\|_2$ for ordered singular values $\sigma_1 > \sigma_2 > \dots$. Note that in our notation the eigenvalues are ordered by their magnitude in absolute value, we have $|\lambda_k(A)| = \sigma_k(A)$. Therefore,

$$|\lambda_1(H_{\hat{\eta}_k}(w_k))| = \sigma_1(H_{\hat{\eta}_k}(w_k)) \leq \sigma_1(H(w^*)) + \sigma_1(H_{\hat{\eta}_k}(w_k) - H(w^*)) = |\lambda_1(H(w^*))| + \|H_{\hat{\eta}_k}(w_k) - H(w^*)\|_2 \quad \text{for all } k.$$

Combining this with Corollary B.16 allows us to derive the thesis

$$
\begin{aligned}
\eta_k &\geq \frac{2(1-c)\delta}{\max_{\eta \in [0, \eta_{w_k}]} |\lambda_1(H_\eta(w_k))|} \\
&\geq \frac{2(1-c)\delta}{|\lambda_1(H(w^*))| + \|H_{\hat{\eta}_k}(w_k) - H(w^*)\|_2} \\
&\geq \frac{2(1-c)\delta}{|\lambda_1(H(w^*))| + \varepsilon}.
\end{aligned}
$$

$\square$

## B.5. Proof of Theorem 2.9

The following proof is directly derived from Grippo & Sciandrone (2023) and Zhang & Hager (2004).

**Theorem B.18.** *Let $f \in C^1(\mathbb{R}^n)$ be lower bounded by $f^*$, and let the level set $\mathcal{L}_0 := \{w \in \mathbb{R}^n : f(w) \leq f(w_0)\}$ be compact. Let $\{w_k\}$ be generated by GD and $\{\eta_k\}$ be generated by a nonmonotone equality line search of the type given in Zhang & Hager (2004)(Algorithm 4), then*

$$\lim_{k \to \infty} \|\nabla f(w_k)\| = 0.$$

*Proof.* From the monotone decrease of the reference value $R_k$ we can conclude that $w_{k+1} \in \mathcal{L}_0$, see

$$R_k = \frac{\xi Q_k}{Q_{k+1}} R_{k-1} + \frac{1}{Q_{k+1}} f(w_k) \leq \frac{\xi Q_k}{Q_{k+1}} R_{k-1} + \frac{1}{Q_{k+1}} \left( R_{k-1} - \eta_k c \|\nabla f(w_k)\|^2 \right) = R_{k-1} - \frac{\eta_k c}{Q_{k+1}} \|\nabla f(w_k)\|^2, \quad (21)$$

which also implies that $\{R_k\}$ is a decreasing sequence which is bounded from below by $f^*$, and in particular it converges to a finite value $R^*$. Using that $Q_k = 0$ is bounded (Lemma B.1), we conclude

$$\lim_{k \to \infty} \eta_k \|\nabla f(w_k)\|^2 = 0. \tag{22}$$

For the sequence $\eta_k$ we distinguish between three possible kinds of subsequences, that is

$$\eta_{k_i} \to \infty, \quad \eta_{k_i} \to \eta^* \in \mathbb{R}_+ \text{ or } \eta_{k_i} \to 0.$$

In the case that the subsequence converges to either infinity or a positive value the thesis $\lim_{i \to \infty} \|\nabla f(w_{k_i})\| = 0$ for the corresponding subsequence $\{\nabla f(w_{k_i})\}$ follows directly.

So consider the subsequence $\eta_{k_i} \to 0$, which implies

$$\lim_{i \to \infty} \eta_{k_i} \|\nabla f(w_{k_i})\| = 0. \tag{23}$$

Suppose the corresponding gradients do not converge to zero, i.e. $\{\nabla f(w_{k_i})\}_i \not\to 0$. W.l.o.g. also assume $\nabla f(w_{k_i}) \neq 0$ for all elements for the subsequence. Now, let $\hat{\eta}_{k_i} := \frac{\eta_{k_i}}{\delta}$, then also

$$\lim_{i \to \infty} \hat{\eta}_{k_i} \nabla f(w_{k_i}) = 0. \tag{24}$$

Also $\hat{\eta}_{k_i}$ does not satisfy the stopping criterion (3), so

$$f(w_{k_i} - \hat{\eta}_{k_i} \nabla f(w_{k_i})) > f(w_{k_i}) - c\hat{\eta}_{k_i} \|\nabla f(w_{k_i})\|^2 \quad \forall i. \tag{25}$$

Let $\hat{w}_{k_i} := w_{k_i} - \hat{\eta}_{k_i} \nabla f(w_{k_i})$ and let us apply the Mean Value Theorem, to achieve that there exists $z_{k_i} \in [w_{k_i}, \hat{w}_{k_i}]$ such that

$$f(\hat{w}_{k_i}) = f(w_{k_i}) + \nabla f(z_{k_i})^T (\hat{w}_{k_i} - w_{k_i}).$$

Combining this result with (25) yields

$$- \nabla f(z_k)^T \nabla f(w_{k_i}) > -c || \nabla f(w_{k_i})||^2 \quad \forall i. \tag{26}$$

which, using Cauchy-Schwarz, implies

$$(1-c) \| \nabla f(w_{k_i})\|^2 < \nabla f(w_{k_i})^T (\nabla f(w_{k_i}) - \nabla f(z_{k_i})) \le \| \nabla f(w_{k_i})\| \| \nabla f(w_{k_i}) - \nabla f(z_{k_i})\|.$$

Therefore, as $\nabla f(w_{k_i}) \neq 0$

$$(1-c) \| \nabla f(w_{k_i})\| \le \| \nabla f(w_{k_i}) - \nabla f(z_{k_i})\|, \tag{27}$$

For some $\gamma > 0$ denote by $L_0^\gamma$ the closed $\gamma$ neighborhood around $L_0$, i.e. $L_0^\gamma = \{x \in \mathbb{R}^n : \text{dist}\{x, L_0\} \le \gamma\}$. By construction $\|z_{k_i} - w_{k_i}\| \to 0$, hence for a $\gamma > 0$ there exists an $K > 0$ s.t. for all $k_i > K$ $\|z_{k_i} - w_{k_i}\| \le \gamma$, which implies $w_{k_i}, z_{k_i} \in L_0^\gamma$. As $\nabla f$ is continuous, it is uniformly continuous on the compact set, hence

$$\lim_{i \to \infty} \| \nabla f(z_{k_i}) - \nabla f(w_{k_i})\| = 0,$$

which by (27) gives the convergence to zero of the subsequence of the gradient. $\qquad \square$

Notice that the compactness of the $\mathcal{L}_0$ level set is implied by the coercivity of the function $f$, i.e., $\lim_{||w|| \to \infty} f(w) = \infty$. This property holds for MSE loss functions applied on neural networks with ReLU activation functions (or any other unbounded activation function), but it does not hold for unregularized cross-entropy losses. In this case, the compactness of the $\mathcal{L}_0$ level set can be replaced with the Kurdyka-Łojasiewicz property (Kanzow & Lehmann, 2025) or by assuming that $\eta_k \in [\eta_{\min}, \eta_{\max}]$. Both the lower ($10^{-6}$) and the upper ($10^6$) safety thresholds are employed in the implementation, but never met in practice by NLS or PoNLS.

**Corollary B.19.** *Let $f \in C^1(\mathbb{R}^n)$ be lower bounded by $f^*$. Let $\{w_k\}$ be generated by GD with nonmonotone equality line search of the type given in (Zhang & Hager, 2004) Algorithm 4 with step-sizes $\{\eta_k\}$. Denote by $\hat{\eta}_k = \frac{1}{k+1} \sum_{s=0}^{k} \eta_s$ the arithmetic mean of the step sizes. Then,*

$$\min_{s=0,\ldots,k} \{\| \nabla f(w_s)\|^2\} \le \frac{1}{(1-\xi)(k+1)} \hat{\eta}_k^{-1} (f(w_0) - f^*).$$

*If we also assume $f$ to be segment smooth (Assumption 2.2), and denote by $\hat{l}_k := \frac{k+1}{\sum_{s=0}^{k} \frac{1}{l(w_s, \eta_{w_s})}}$ the harmonic mean of the segment constant, we can conclude:*

$$\min_{s=0,\ldots,k} \| \nabla f(w_s)\|^2 \le \frac{1}{\quad}$$

*Proof.* We unroll (21) to gather

$$f^* \le R_{k+1} \le R_0 - c \sum_{s=0}^{k} \frac{\eta_s}{Q_{s+1}} \| \nabla f(w_s)\|^2,$$

which using $R_0 = f(w_0)$ implies

$$c \sum_{s=0}^{k} \frac{\eta_s}{Q_{s+1}} \| \nabla f(w_s)\|^2 \le f(w_0) - f^*,$$

using $Q_s \le \frac{1}{1-\xi}$ (Lemma B.1), this allows us to conclude

$$(1-\xi) \left( \frac{1}{k+1} \sum_{s=0}^{k} \eta_s \right) (k+1) \min \| \nabla f(w_s)\|^2 \le f(w_0) - f^*,$$

which leads to the thesis.

For $f$ satisfying Assumption 2.2 we use Proposition B.6 to lower bound each $\eta_s$ and conclude

$$(1 - \xi)2\delta(1 - c)\left(\frac{1}{k+1}\sum_{s=0}^{k}\frac{1}{l(w_s, \eta_{w_s})}\right)\min \| \nabla f(w_s)\|^2 \leq f(w_0) - f^*.$$

$\square$

## B.6. Lipschitz and Sharpness awareness of the Polyak step size, Proof of Proposition 2.11

In this section, we show similar results as Proposition B.6 and Proposition B.16 for Algorithm 3. That is we start by lower bounding the step-size in terms of the local smoothness constant.

**Lemma B.20.** *Let $f \in C^1$ satisfy Assumption 2.2. Let $\eta_k$ be generated by Algorithm 3, then*

$$\eta_k \geq \frac{\min\{2(1 - c)\delta, 1/(2c_p)\}}{l(w_k, \eta_{w_k})}.$$

*Proof.* We use the result from Proposition B.6 and just have to deal with the case $l_k = 0$, i.e. not having any cuts, that is $\eta_k = \frac{f(w_k) - f^*}{c_p \| \nabla f(w_k)\|_2^2}$, by definition.

Let $f^*$ be the minimum value obtained by the function then, we apply the Descent Lemma B.5 with $w_{k+1} = w_k - \frac{1}{l(w_k, \eta_{w_k})}\nabla f(w_k)$

$$f^* \leq f\left(w_k - \frac{1}{l(w_k, \eta_{w_k})}\nabla f(w_k)\right) \leq f(w_k) - \frac{1}{2l(w_k, \eta_{w_k})}\| \nabla f(w_k)\|_2^2$$

$$\Leftrightarrow \frac{f(w_k) - f^*}{\| \nabla f(w_k)\|_2^2} \geq \frac{1}{2l(w_k, \eta_{w_k})},$$

which gives a lower bound on the initial step size. So, all in all, we arrive at the claimed inequality. $\square$

Now we also relate this result to the local sharpness along the line.

**Corollary B.21.** *Let $f$ be twice continuously differentiable and let $f$ satisfy Assumption 2.2. Let $\eta_k$ be generated by Algorithm 3 and $H_\eta(w_k) := \nabla^2 f(w_k - \eta\nabla f(w_k))$, then*

$$\eta_k \geq \frac{\min\{2(1 - c)\delta, 1/(2c_p)\}}{\max_{\eta \in [0, \eta_{w_k}]}|\lambda_1(H_\eta(w_k))|}.$$

*Proof.* Based on the previous Lemma B.20 and following the steps of the proof of Corollary B.16, we reach the conclusion.
$\square$

## B.7. The flatness of minima, proof of Lemma 3.1

In this subsection, we shift our perspective. For a fixed data point $x$ let $n_x : \mathbb{R}^N \to \mathbb{R}^K$ denote the function that describes the output of the network w.r.t. the weights ($K$ is the number of classes), $N$ the number of weights. Let $y : \mathbb{R}^K \to \mathbb{R}^K$ be a function applied after the network, usually the identity or softmax.

Moreover, assume that last layer in $n_x$ is fully connected and linear. That is $n_x(w) = Wx' + b$, where $x' := n_x^{L-2}(w') \in \mathbb{R}^{N_{L-1}}$ is the output of the previous layer, $w'$ the set of weights without $(W, b)$, $W \in \mathbb{R}^{K \times N_{L-1}}, b \in \mathbb{R}^K$ a matrix and vector.

We consider the following two loss functions:

**Definition B.22** (Loss functions). Let $(x_i, y_i)_{i=1}^m \in \mathbb{R}^m \times \mathbb{R}^K$ be a fixed sample for a classification task with $K$ classes and $n : \mathbb{R}^n \times \mathbb{R}^m \to \mathbb{R}^K$ some classifier. We define the following loss functions:

1. The mean squared error (MSE)-loss: $l : \mathbb{R}^n \to \mathbb{R}, l(w) = \frac{1}{Km}\sum_{i=1}^m \|n(w, x_i) - y_i\|_2^2.$

2. The cross-entropy loss: $l : \mathbb{R}^n \to \mathbb{R}$, $l(w) = \frac{1}{m} \sum_{i=1}^m \sum_{k=1}^C -y_{i,k} \log(n(w, x_i)_k)$.

**Lemma B.23.** *Let $(x, \hat{y})$ be a data point, $f$ the MSE-loss for the output of the network $n_x : \mathbb{R}^n \to \mathbb{R}^K$ with identity afterwards. Assume that the last layer is linear, i.e. $n_x(w) = Wx'(w') + b$, where $x'(w')$ is the output of the previous layer and denote by $w' := w \backslash \{W, b\}$ all parameters but the ones in the last layer. Then the following are true*

1. *The subhessian of w.r.t. the weights of the last layer has the form*

$$\nabla_W^2 f(w) = \frac{2}{K} I_K \otimes x'x'^T, \ \nabla_b^2 f(w) = \frac{2}{K} I_K, \ \nabla_{W,b}^2 f(w) = \frac{2}{K} I_K \otimes x'$$

2. *The trace of the subhessian is given by $\text{trace}(\nabla_{(W,b)}^2 f(w)) = 2(\|x'\|^2 + 1)$.*
3. *The biggest eigenvalue of the hessian is at least $\frac{2}{K}$, i.e. $\lambda_1(\nabla_w^2 f(w)) \geq \frac{2}{K}$.*

*Proof.* We start with some general derivations for our loss functional:

$$f(w) = \frac{1}{K} \|n_x(w) - \hat{y}\|_2^2, \quad \nabla_w f(w) = \frac{2}{K} \sum_{i=1}^K (n_{x,i}(w) - \hat{y}_i) \nabla n_{x,i}(w)$$

$$\nabla^2 f(w) = \frac{2}{K} \sum_{i=1}^K \nabla_w n_{x,i}(w) \nabla_w n_{x,i}(w)^T + (n_{x,i}(w) - \hat{y}_i) \nabla_w^2 n_{x,i}(w).$$

1. As we are interested in the last layer of $n_x$, which is given by $Wx' + b$. Let $w_i$ denote the rows of $W$, then

$$\nabla_{w_j} n_{x,i}(w) = \delta_{i,j} x', \quad \nabla_{b_j} n_{x,i}(w) = \delta_{i,j}, \quad \nabla_W^2 n_{x,i}(w) = \nabla_b^2 n_{x,i}(w) = \nabla_{W,b}^2 n_{x,i} = 0,$$

$$\Rightarrow \nabla_W^2 f(w) = \frac{2}{K} \begin{pmatrix} x'x'^T & 0 & \dots & 0 \\ 0 & x'x'^T & \dots & 0 \\ \dots & \dots & \dots & \dots \\ 0 & 0 & \dots & x'x'^T \end{pmatrix}, \quad \nabla_b^2 f(w) = \frac{2}{K} \sum_{i=1}^K e_i e_i^T, \quad \nabla_{W,b} f(w) = \frac{2}{K} \begin{pmatrix} x' & 0 & \dots & 0 \\ 0 & x' & \dots & 0 \\ \dots & \dots & \dots & \dots \\ 0 & 0 & \dots & x' \end{pmatrix}.$$

2. With this at hand, it is straightforward to calculate the trace

$$\text{trace}(\nabla_{(W,b)}^2 f(w)) = \frac{2}{K} \sum_{i=1}^K \|\nabla_{(W,b)} n_{x,i}(w)\|^2 = \frac{2}{K} \sum_{i=1}^K \left( \|x\|^2 + 1 \right) = 2(\|x\|^2 + 1).$$

3. This can be seen by Cauchy's interlacing theorem (Horn & Johnson, 2012), which implies for a symmetric matrix

$$A = \begin{pmatrix} B & C \\ C^T & D, \end{pmatrix}, \qquad \lambda_1(A) \geq \lambda_1(D).$$

Which we use with the sub-block $\nabla_b^2 f(w)$ which has eigenvalues $\frac{2}{K}$, as calculated previously.

$\square$

As the previous Lemma was just for a single data point by the linearity of the loss function we can easily derive a result for the whole sample set.

**Corollary B.24.** *Let $\{x_i, y_i\}_{i=1}^m$ be a set of labeled training data, $n_{x_i} : \mathbb{R}^n \to \mathbb{R}^K$ the output of the network for the corresponding sample point. As in Lemma 3.1 we assume that the network has a linear last layer with bias and denote by $x_i'(w')$ the output of the second to last layer for the data point $x_i$. Let $f$ be the MSE loss over the whole training set, i.e. $f(w) = \frac{1}{Km} \sum_{i=1}^m \|Wx_i'(w') + b - y_i\|_2^2$. The subhessian is given by*

$$\nabla_W^2 f(w) = \frac{2}{K} I_K \otimes \frac{1}{m} \sum_{i=1}^m x_i' x_i'^T, \ \nabla_b^2 f(w) = \frac{2}{K} I_K, \ \nabla_{W,b} f(w) = \frac{2}{K} I_K \otimes \frac{1}{m} \sum_{i=1}^m x'.$$

*Therefore, again trace $\left( \nabla_{(W,b)}^2 f(w) \right) = 2 \left( \frac{1}{m} \sum_{i=1}^m \|x_i'\|_2^2 + 1 \right)$ and the biggest eigenvalue of the Hessian w.r.t. the whole data set is lower bounded by $\frac{2}{K}$.*

*Proof.* This follows from the structure of the loss, the linearity of the Hessian operator and the linearity of the trace aswell as the previous Lemma. □

Corollary B.24 provides us with some insight on why a trace of exactly 2 sometimes emerges in training. In this setting the output of the previous layer $x'$ seems to be almost zero (see Figures 27-31). This then only leaves the sub Hessian w.r.t. the bias in the last layer, yielding $K$ eigenvalues of exactly $\frac{2}{K}$.

So far we have considered the MSE loss with the identity function applied to the output of the network. We will now shift our focus to the cross entropy loss with the softmax function applied to the output.

**Lemma B.25.** *Let $(x, \hat{y})$ be a data point with one-hot encoding, $f$ the cross-entropy loss for the output of the network $n_x : \mathbb{R}^n \to \mathbb{R}^K$ with softmax afterwards. Assume that the last layer is linear, i.e. $n_x(w) = Wx'(w') + b$, where $x'(w')$ is the output of the previous layer and denote by $w' := w\backslash\{W, b\}$ all parameters except those in the last layer. Furthermore, we denote by $p_j = \frac{e^{n_x(w)_j}}{\sum_{k=1}^{K} e^{n_x(w)_k}}, j = 1, \ldots, K$ the output of the softmax function. Then the following are true:*

1. *The subhessian of w.r.t. the weights of the last layer has the form*

$$\nabla_W^2 f(w) = \left(diag(p) - pp^T\right) \otimes x'x'^T, \ \nabla_b^2 f(w) = \left(diag(p) - pp^T\right), \ \nabla_{W,b}^2 f(w) = \left(diag(p) - pp^T\right) \otimes x'$$

2. *The trace of the subhessian is given by $trace(\nabla_{(W,b)}^2 f(w)) = \left(1 - \|p\|_2^2\right)\left(1 + \|x'\|_2^2\right)$.*

*Proof.* As $\hat{y}$ is a one-hot encoding $\hat{y} = e_k$ for one $k \in \{1, \ldots, K\}$, the loss function of the network and its gradient can thus be written as

$$f(w) = -n_x(w)_k + \log\left(\sum_{j=1}^{K} e^{n(w)_j}\right),$$

$$\nabla_w f(w) = -\nabla_w n_{x,k}(w) + \sum_{j=1}^{K} p_j \nabla_w n_{x,j}(w),$$

$$\nabla_w^2 f(w) = -\nabla_w^2 n_{x,k}(w) + \sum_{j=1}^{K} p_l\left(\nabla_w n_{x,l}(w)\nabla_w n_{x,l}(w)^T + \nabla_w^2 n_{x,l}(w)\right) - \left(\sum_{l=1}^{K} p_l \nabla_w n_{x,l}(w)\right)\left(\sum_{l=1}^{K} p_l \nabla_w n_{x,l}(w)\right)^T.$$

The computations of $\nabla_{(W,b)} n_x(w')$ are as in Lemma B.23 and thus we arrive at

$$\nabla_W^2 f(W, b) = \sum_{l=1}^{K} p_l e_l \otimes x'(e_l \otimes x')^T - \left(\sum_{l=1}^{K} p_l e_l \otimes x'\right)\left(\sum_{l=1}^{K} p_l e_l \otimes x'\right)^T = \left(\text{diag}(p) - pp^T\right) \otimes x'x'^T,$$

$$\nabla_b^2 f(W, b) = \sum_{l=1}^{K} p_l e_l e_l^T - \left(\sum_{l=1}^{K} p_l e_l\right)\left(\sum_{l=1}^{K} p_l e_l\right)^T = \left(\text{diag}(p) - pp^T\right),$$

$$\nabla_{W,b}^2 f(W, b) = \sum_{l=1}^{K} p_l e_l e_l^T \otimes x' - \left(\sum_{l=1}^{K} p_l e_l \otimes x'\right)^T = \left(\text{diag}(p) - pp^T\right) \otimes x'.$$

The calculation of the trace is then straightforward, by

$$\text{trace}(\nabla_{(W,b)}^2 f(W, b)) = \text{trace}(\nabla_W^2 f(W, b)) + \text{trace}(\nabla_b^2 f(W, b)) = \text{trace}(\text{diag}(p) - pp^T)(1 + \text{trace}(x'x'^T)) = \left(1 - \|p\|_2^2\right)\|x'\|_2^2,$$

where we have used that the vector $p$ sums to one.

□

## C. Additional Experimental Details

The code to reproduce our experiments is given at https://github.com/curtfox/flatland.

### C.1. Datasets

For all deterministic experiments, we use stratified sampling to ensure that there are an equal number of images with each class label when sub-sampling the full dataset. For the stochastic experiments, we use the full dataset for training.

- CIFAR10 and CIFAR100 (Krizhevsky et al., 2009): For the deterministic experiments, we subsample 5000 of the 50,000 training images given in the full dataset.
- SVHN (Netzer et al., 2011): For the deterministic experiments, we subsample 4994 of the 73,257 training images given in the full dataset.
- EMNIST Letters (Cohen et al., 2017): For the deterministic experiments, we subsample 4992 of the 145,600 training images given in the full dataset.

### C.2. Models

All models use the PyTorch default for the initialization of model parameters. We include bias parameters in all of our models.

The CNN model is a convolutional neural network with the following structure:

- convolutional layer with 3 input channels and 32 output channels
- ReLU activation
- average pooling layer
- convolutional layer with 32 input channels and 32 output channels
- ReLU activation
- average pooling layer
- linear layer with input dimension 2048 and output dimension corresponding to the number of classes

The MLP is a multi-layer (three) perceptron model with the following structure:

- linear layer with input dimension depending on the dataset and output dimension 100
- ReLU activation
- linear layer with input dimension 100 and output dimension 100
- ReLU activation
- linear layer with input dimension 100 and output dimension equal to the number of classes in the dataset

For the resnet34, vgg11, densenet121, and wide_resnet50_2 experiments, we use the Torchvision implementations of these models. Similarly to Roulet et al. (2024), we remove all batch normalization layers in the resnet34 experiments, and do not use any dropout in the vgg11 experiments. For consistency, we also remove all batch normalization layers in the wide_resnet50_2 experiments, and do not use any dropout in the densenet121 experiments.

For our vision transformer model (tinyVIT), we use the SimpleViT model from the vit-pytorch package. See the following link for their code. We set the parameters as follows: $patch\_size = 8$, $dim = 256$, $depth = 4$, $heads = 8$, and $mlp\_dim = 512$. Finally, the $image\_size$ parameter is set equal to the height of the input images and the $num\_classes$ parameter is set equal to the number of classes in the dataset.

### C.3. Optimization

Unless otherwise specified, experiments in this paper are performed in full batch mode. For all line search methods, we set $c = 0.0001$ and $\delta = 0.5$ (except for Section G in which $\delta = 0.9$). For NLS and LS, the initial step size $\eta_{k,0} = \eta_{k-1}$ with $\eta_{0,0} = 1$, while for PoNLS the initial step size is set using the Polyak step size on each iteration with $f^* = 0$ (as in the work by Galli et al. (2023)). Note that for the nonmonotone line searches NLS and PoNLS, $R_k$ as given in (2) is computed using the method in Zhang & Hager (2004) (see (8) for more details). For the monotone LS method, the standard Armijo method where $R_k$ is set to $f(w_k)$ (Armijo, 1966) is used. For SAM, we select the step size with the best training loss over the grid $\{0.0001, 0.001, 0.01, 0.1\}$. Moreover, we do not use any regularization or momentum in our experiments except for SAM, where we use a weight decay factor of 0.0001 and a momentum factor of 0.9. It is worth noting that SAM has various

hyper-parameters (e.g., the neighborhood size of the perturbation) that were set to the default values in Foret et al. (2020) and not fine-tuned. We expect the results of SAM to improve when further hyper-parameter optimization is performed over these choices.

### C.4. Calculation of Sharpness

For the experiments where the sharpness is computed, we compute the sharpness using the *eigenvalues* function in the PyHessian package (Yao et al., 2020), with the *maxIter* parameter set to 100, *tol* parameter set to $10^{-3}$, and the *top_n* parameter set according to the number of the top eigenvalues we want to compute. Note that the sharpness corresponds to the top eigenvalue.

For the full batch experiments, we compute the sharpness every 100 iterations, with some exceptions. For the warmup experiment in D, we compute the sharpness every iteration. For the experiments zooming in on fewer iterations in Section 3.1 and Appendix I.2, we compute the sharpness every 2 iterations.

For the stochastic experiments in Appendix E, we compute the sharpness every 2 epochs.

### C.5. Calculation of the Random Guessing Value

Given a MSE loss for classification with a normalizing factor of $1/K$, a model that guesses randomly always outputs $1/K$ for any data point $x$. The corresponding loss value for a random guess with $K$ classes will be

$$f(w_K^r) = \frac{1}{K}\left(\left(1 - \frac{1}{K}\right)^2 + (K-1)\left(\frac{1}{K}\right)^2\right).$$

Thus, $f(w_{10}^r)) \approx 0.09$, $f(w_{100}^r)) \approx 0.009$, $f(w_{26}^r)) \approx 0.0369$.

### C.6. Compute

We run our experiments on a mix of NVIDIA A100, H100, L40S, and GeForce RTX 4090 GPU's. The total computation time of our experiments is around 1400 hours.

# D. Warmup Experiment

In this section, we show that GD with a warmed up step size can also hit the globally flat region. The GD-warmup run increases the step size to 125 (which is beyond the number of classes 100), and then decreases the step size to 90 to prevent divergence. This step size is used for the rest of training. Based on the sharpness plots (second row), we observe that this method hits the globally flat region within the first 50 iterations of training, without any line search. We compare this to the GD-warmup-ub, which increases the step size to 90 and uses this step size for the rest of training. Notably, the step size for this method is kept below 100 for all iterations. Just like in the line search setting, we see that upper bounding the step size so that it stays below the number of classes in the dataset prevents the method from reaching the globally flat region. Additionally, this also leads to improved convergence results, improving both the training loss and test accuracy.

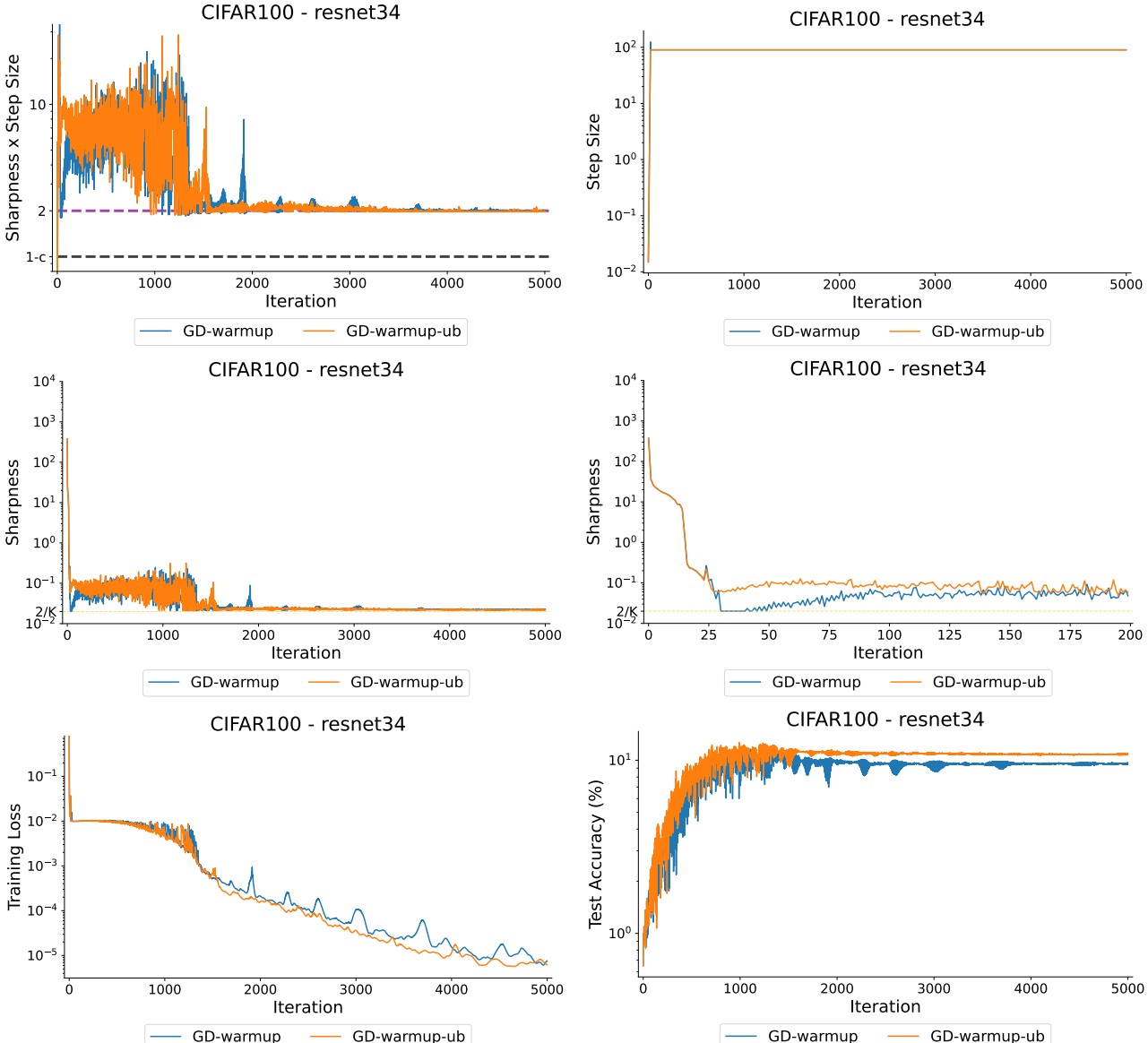

*Figure 5.* We plot the sharpness * step size (Top Left), step size (Top Right), sharpness (Middle Left), sharpness for first 200 iterations (Middle Right), training loss (Bottom Left), and test accuracy (Bottom Right) for two different runs of gradient descent using different variants of warmup. This experiment is performed for the resnet34 model on the CIFAR100 dataset.

# E. Stochastic Experiments

Similar to the deterministic case, for CDAT we set the stepping-on-the-edge parameter $\sigma$ to 2.06, as suggested in Roulet et al. (2024). We also select the step size for SAM via a grid search (see Appendix C). For the stochastic case, we compute the sharpness differently than in the full batch case. In particular, we compute the sharpness for each batch separately, and then calculate either the minimum or the average across all the batches at the end of each epoch. We present these values in the last two rows of Figure 6 and Figure 7.

Similar to the full batch case and looking at the average batch sharpness, minimum batch sharpness, and step size plots in Figures 6 and 7, we see that NLS uses large step sizes that bring the average and minimum batch sharpness to small and constant values. Now first focusing on the resnet34 experiments, we see that in the extreme case of NLS on SVHN×resnet34, the method is stuck at the same training loss throughout almost the entire training procedure. In addition, both the average and minimum batch sharpness remain at the $2/K$ threshold for this entire period. In a less extreme case, NLS reduces both the average and minimum sharpness values to the $2/K$ threshold for close to half of the training time on CIFAR100×resnet34. Therefore, even SGD may get "stuck" in globally-flat regions, and this issue is not exclusive to the full batch regime. However, similar to the deterministic case, NLS-ub is able to avoid globally flat regions. Additionally, we observe that NLS-ub either gives similar results to or significantly improves on NLS both in terms of training loss and test accuracy. Now instead focusing on the vgg11 experiments, we note that both the average and minimum sharpness values remain above the $2/K$ threshold. However, in the SVHN×vgg11 case, the minimum batch sharpness of NLS remains small and just above this threshold for a significant portion of training. Interestingly, even for the vgg11 runs where NLS avoids the globally flat region, employing the NLS-ub algorithm leads to improved convergence results while still maintaining low average and minimum batch sharpness. We leave further exploration of the observations made in the stochastic case to future work.

Moving onto the LS method, we see that when employing small batches, it does not increase the sharpness as much as the full batch case, and overall seems to have more comparable performance to the nonmonotone algorithms than in the full batch case. Interestingly, CDAT sometimes diverges entirely, which we see when training the vgg11 model on the CIFAR100 and SVHN datasets. This is in line with the work by Roulet et al. (2024), which suggests that different values of $\sigma$ smaller than 2.06 may be necessary for convergence in the stochastic case.

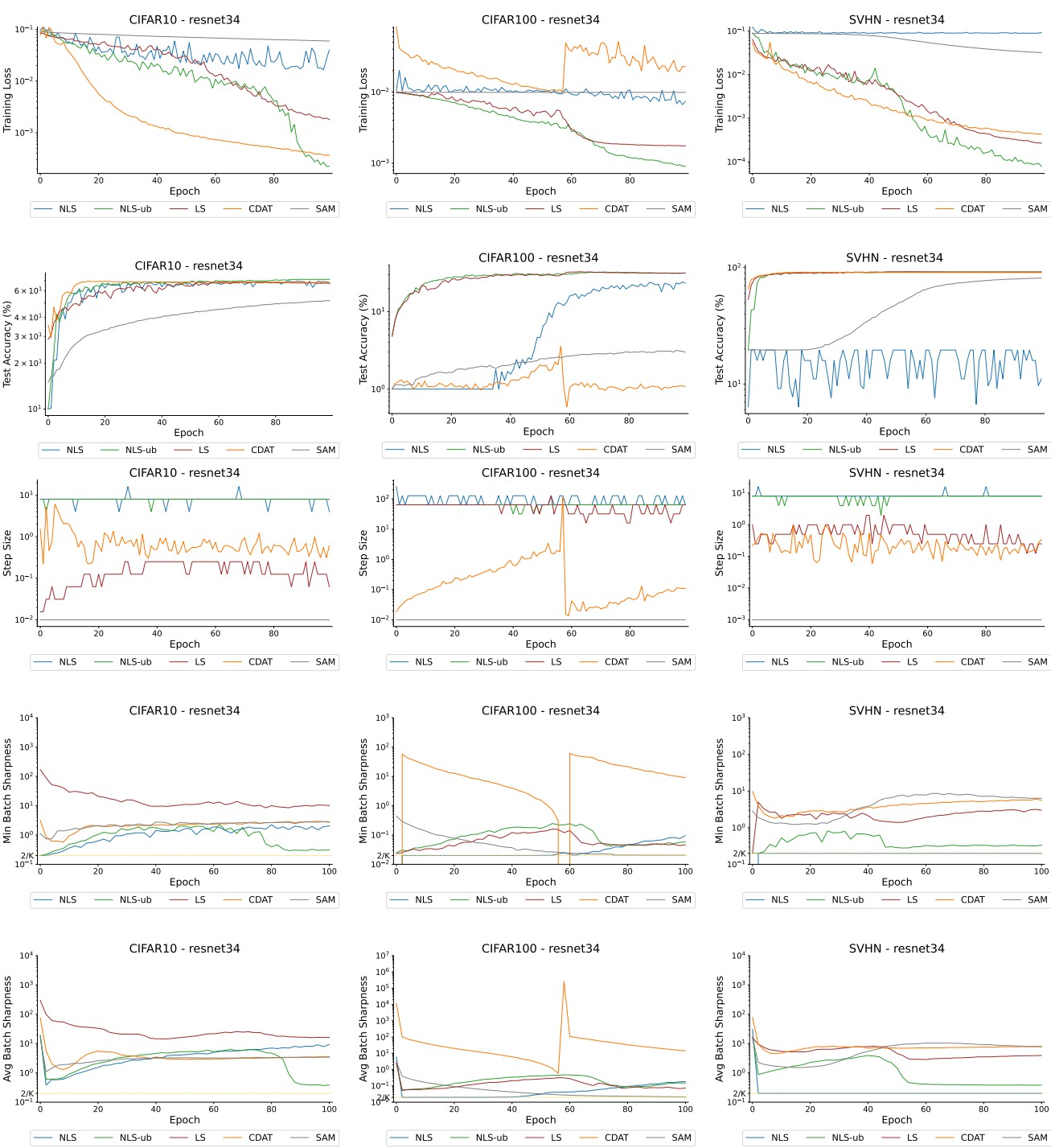

*Figure 6.* We plot the training loss (1st Row), test accuracy (2nd Row), step size (3rd Row), minimum sharpness across all batches in each epoch (4th Row), and average sharpness across all batches in each epoch (5th row) for five different methods. We compare gradient descent with the LS, NLS, and NLS-ub line searches as well as gradient descent with the CDAT step size selection and SAM. This is repeated on the CIFAR10, CIFAR100, and SVHN datasets for the resnet34 model using batches of size 256.

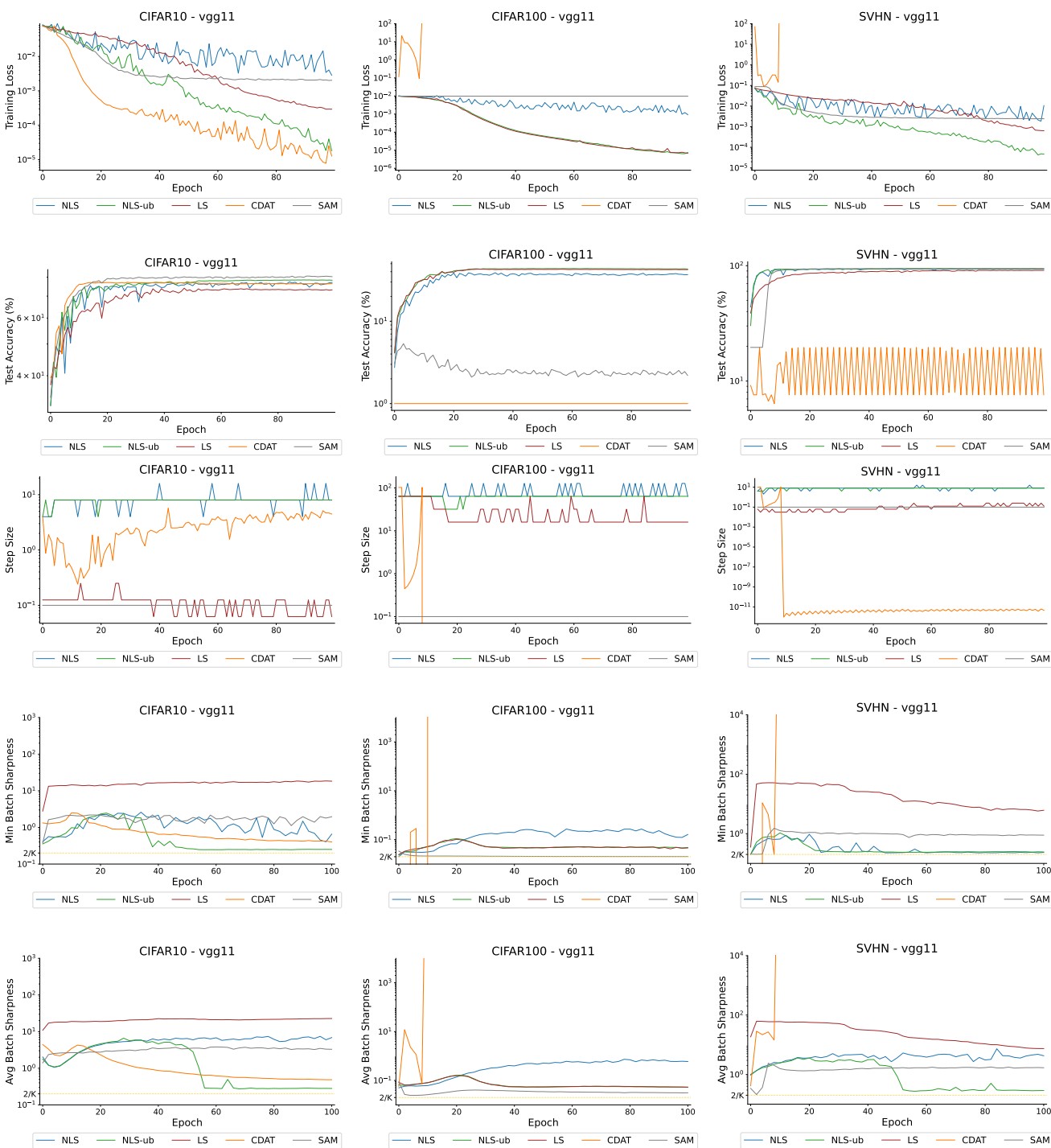

*Figure 7.* We plot the training loss (1st Row), test accuracy (2nd Row), step size (3rd Row), minimum sharpness across all batches in each epoch (4th Row), and average sharpness across all batches in each epoch (5th row) for five different methods. We compare gradient descent with the LS, NLS, and NLS-ub line searches as well as gradient descent with the CDAT step size selection and SAM. This is repeated on the CIFAR10, CIFAR100, and SVHN datasets for the vgg11 model using batches of size 256.

## F. Vision Transformer Experiments

In this section, we discuss the experimental results of our vision transformer experiments in Figure 8. Let us first notice that in the CIFAR10 and SVHN experiments, both NLS or PoNLS are far from hitting the globally-flat value of $2/K$. For the CIFAR100 dataset, although the sharpness does not exactly hit the $2/K$ threshold as in some earlier experiments with other models given in the paper, it hovers above $2/K$ for the NLS method. At the same time, we observe that the step sizes oscillate around $K$ and that the training loss does not improve beyond randomly guessing outputs (i.e., with $K = 100$, this corresponds to a loss of roughly $10^{-2}$) for many iterations. This behavior is similar to what we observed when the globally-flat regions are found on other architectures. It is also worth noticing that PoNLS encounters very small (and in fact negative values) sharpness values on some iterations when training the tinyVIT model on CIFAR100. Interestingly, we do not see this with the other architectures tested in this paper, that is, the largest eigenvalue is always positive. These differences between transformers and convolutional architectures will be addressed in future work.

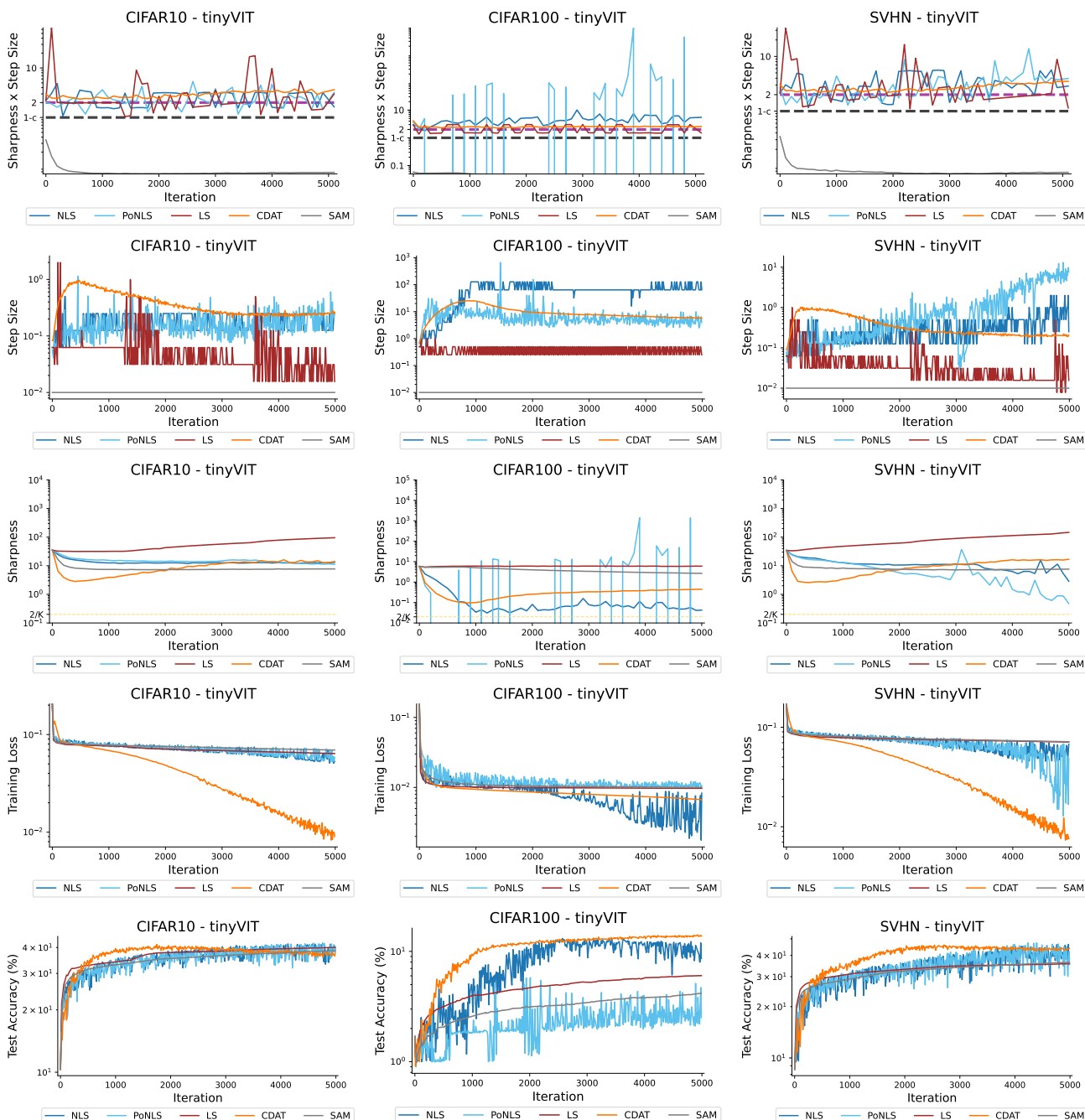

*Figure 8.* We plot the sharpness * step size (1st Row), step size (2nd Row), sharpness (3rd Row), training loss (4th Row), and test accuracy (5th row) for five different methods. We compare gradient descent with the LS, NLS, and PoNLS line searches as well as gradient descent with the CDAT step size selection and SAM. This is repeated for a vision transformer (tinyVIT) model on the CIFAR10, CIFAR100, and SVHN datasets.

## G. Delta Ablation Experiments

Recall that $\eta_k^*$ is the step size for which (8) is satisfied as an equality. From Lemma B.3 we know that by choosing $\delta$ arbitrarily close to 1, we have that $\eta_k$ will be consequently close to $\eta_k^*$. In particular, it will satisfy $\eta_k^* \in (\eta_k, \frac{\eta_k}{\delta})$. However, choosing a larger $\delta$ will increase the computational cost of finding $\eta_k$. Note that from Corollary B.8, we see that choosing $\delta = 0.5$ instead of $\delta = 0.9$ reduces the maximum number of function evaluations of the NLS line search by a factor of $\approx 7$, as this value is logarithmic with base $1/\delta$. Beyond the upper bound provided in Corollary B.8, we examine the experimental differences between using NLS with $\delta = 0.5$ and $\delta = 0.9$ in Figure 9 and Figure 10. In the last row of these figures, we show the number of function evaluations of the two methods, averaged over windows of size 25 to make the results more readable. We see that for both CNN experiments, using $\delta = 0.9$ generally leads to more function evaluations than using $\delta = 0.5$. For the resnet34 experiments, using $\delta = 0.9$ leads to significantly more function evaluations early in training compared to using $\delta = 0.5$, but both methods require very little evaluations afterwards. Finally, for the vgg11 experiments, again using $\delta = 0.9$ leads to significantly more function evaluations early in training. However, in later iterations the number of function evaluations for both is comparable.

Additionally, based on Figure 9 and Figure 10, using $\delta = 0.9$ in the line search seems to push NLS to operate at the EoS since the sharpness * step size is almost always at or above the threshold of 2. This is in line with the results given in Proposition 2.5. Another interesting observation is that when using $\delta = 0.9$ instead of $\delta = 0.5$, the step size is approximately $K$ and the sharpness $2/K$. In addition, the training loss stays mostly flat and the convergence is much worse than when using $\delta = 0.5$. These points suggest that using $\delta = 0.9$ forces NLS to get stuck in the globally flat region more often.

We conclude this section by comparing the computational costs between CDAT, SAM, and NLS. When $\delta = 0.5$, the number of function evaluations per iteration is about 4 for NLS. This, together with the cost of computing one gradient (roughly twice the cost of a forward pass) sums up to 6 forward passes per iterations. This value is the same as that of SAM and CDAT. In fact, beyond computing the GD step (1 forward pass + 1 backward pass), the cost of computing the denominator of CDAT's step size in (7) is 3 function evaluations (Roulet et al., 2024). Instead, SAM computes 2 GD passes at each iterations (Foret et al., 2020), i.e., 2 forward and 2 backward passes. We also point out that the memory requirement of computing the denominator of CDAT is approximately three times that of computing the objective function. It is worth mentioning that prior work showed that a nonmonotone line search with $\delta = 0.5$ (Galli et al., 2023) may be tweaked to employ only one additional function evaluation (on average), bringing the total to 5 per iteration. Finally, it would be interesting to combine NLS and PoNLS with an adaptive choice of $\delta$, as suggested in Cavalcanti et al. (2024).

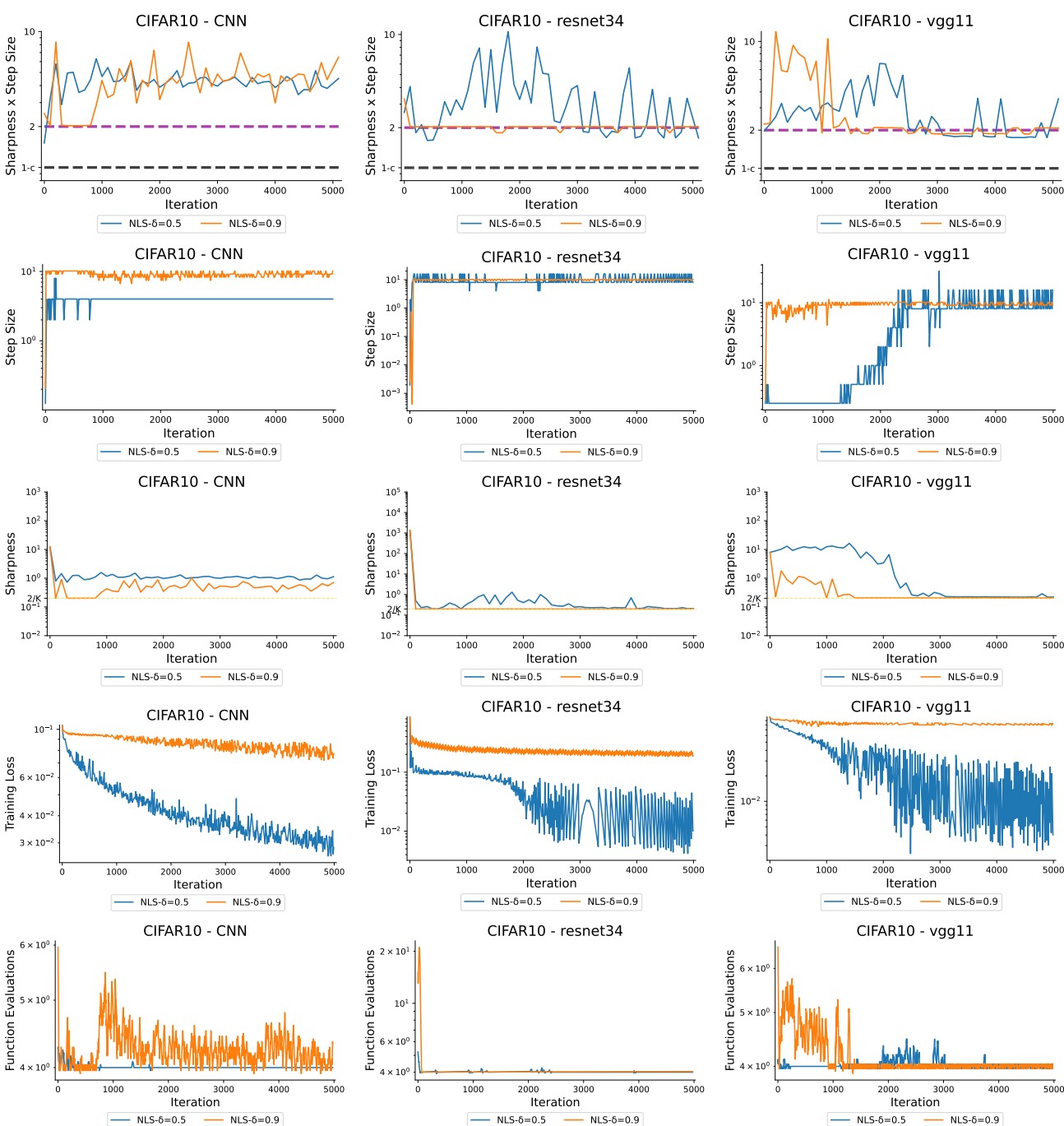

*Figure 9.* We plot the sharpness * step size (1st Row), step size (2nd Row), sharpness (3rd Row), training loss (4th Row), and function evaluations (5th row) for five different methods. We compare the NLS method with $\delta = 0.5$ and $\delta = 0.9$. This is repeated for the CNN, resnet34, and vgg11 models on the CIFAR10 dataset.

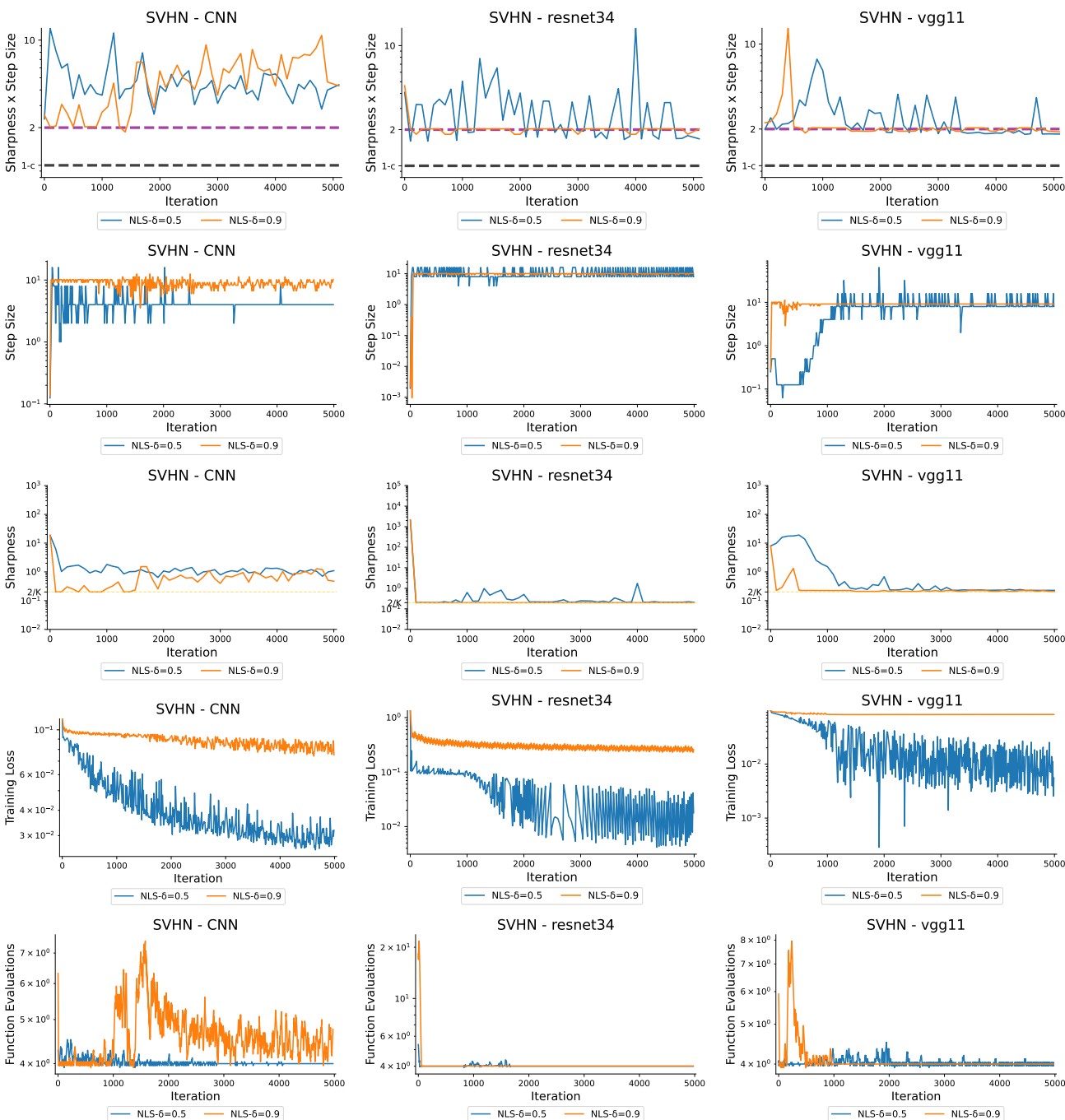

*Figure 10.* We plot the sharpness * step size (1st Row), step size (2nd Row), sharpness (3rd Row), training loss (4th Row), and function evaluations (5th row) for five different methods. We compare the NLS method with $\delta = 0.5$ and $\delta = 0.9$. This is repeated for the CNN, resnet34, and vgg11 models on the SVHN dataset.

# H. Cross Entropy Loss Experiments

In this section, we compare the different methods discussed in Section 3 using the cross entropy loss, rather than the MSE loss. In addition to the metrics previously plotted, we also plot the gradient norm at each iteration.

Concerning the globally-flat phenomenon, the main difference between MSE loss and cross-entropy loss is the shape of the sub-Hessian w.r.t. the bias of the last layer. As calculated in Lemma B.25, given the output $n_x(w)$ of the last layer of the network and the softmax renormalization $p_j = \frac{e^{n_x(w)_j}}{\sum_{k=1}^{K} e^{n_x(w)_k}}, j = 1, \ldots, K$ applied to this output, the Hessian w.r.t. the last bias is $\nabla_b^2 f(w) = \left(\mathrm{diag}(p) - pp^T\right)$, where $p$ is the $K$-dimensional vector with entries $p_j$. In particular, the diagonal entries of $\nabla_b^2 f(w)$ are not lower bounded by the constant value $2/K$ unlike for the MSE loss, but they may all become $0$. For instance, if $p = e_i$, i.e., the canonical basis vector with $1$ in the entry $i$ and $0$ everywhere else, all the entries of this sub-Hessian are $0$. In other words, the lower bound of the sharpness of the cross-entropy loss of this sub-Hessian is $0$, and not $2/K$. In fact, from the third row of Figures 11-18 we observe that NLS and PoNLS sometimes suddenly bring the sharpness to very small values (e.g., $10^{-4}$ for CIFAR10×vgg11 for PoNLS), and slowly decrease it. On the other hand, in some of the other experiments the sharpness is not as small but is approximately constant for a portion of or most of training (e.g., $10^{-2}$ on CIFAR10×resnet34 for PoNLS and NLS). While in the first case the corresponding value of the test accuracy is usually higher (around $40\%$ to $50\%$), in the second case the model is randomly guessing outputs and the test accuracy is low (around $10\%$) as long as the sharpness remains close to constant. Note that in the latter case, the expected sharpness for a uniform probability $p_j = \frac{1}{K}$ is indeed $\frac{1}{K}$. Overall, our discussion and experiments show that in the cross entropy setting, the NLS and PoNLS runs can also be distinguished into the same two classes as the MSE setting, successfully- and unsuccessfully-flat trainings, with a global minimum value of the sharpness that is now $0$ and not $2/K$.

Interestingly enough, the points where the sharpness reaches values that are remarkably below $\frac{1}{K}$ are the same points in which both the training loss and the gradient norm reach very small values. In other words, these points are globally-flat global minimizers of the training loss[2]. Additionally, for all these points with a very small gradient norm, the segment smoothness function $l(w_k, \eta_{w_k})$ is no longer meaningful. In fact, given $w^*$ such that $\nabla f(w^*) = 0$, the inequality $\|\nabla f(w^*) - \nabla f(w^* - \eta \nabla f(w^*))\| \leq l\|w^* - w^* + \eta \nabla f(w^*)\|$ holds trivially ($0 \leq 0$). In these cases, it is reasonable to observe from the first row of Figures 11-18 that the product $\eta_k \lambda_1(H(w_k))$ is below $1 - c$, despite Proposition 2.5.

In the comparison between using cross entropy or MSE loss for classification tasks, our results agree with those of Hui & Belkin (2021) which suggests that there is very little numerical evidence for the advantages of using the former over the latter. In our benchmark, for the experiments involving vgg11 models, the MSE loss achieves higher test accuracy than using the cross-entropy loss, while in the case of resnet34 and densenet121 it is the cross-entropy loss that works better than using the MSE loss.

To conclude, let us focus on the CIFAR10×CNN experiment. Between the test accuracy of CDAT and that of NLS (or PoNLS), there are roughly 5 to 10 percentage points. On the other hand, NLS and PoNLS globally minimize both sharpness and training loss, while CDAT does not achieve either the lowest training loss nor the lowest sharpness. In cases like the one just described, it seems obvious that the joint use of training loss and sharpness is not a good proxy for the test accuracy. However, considering classical approximation theory, it is possible that using a data subset of 5000 data points is not large enough to accurately approximate the underlying distribution. In other words, the responsibility of solving this aspect of the risk minimization problem does not lie in the hands of the optimization method.

---

[2]To be more precise, the global minimizers of the cross entropy losses are not attained.

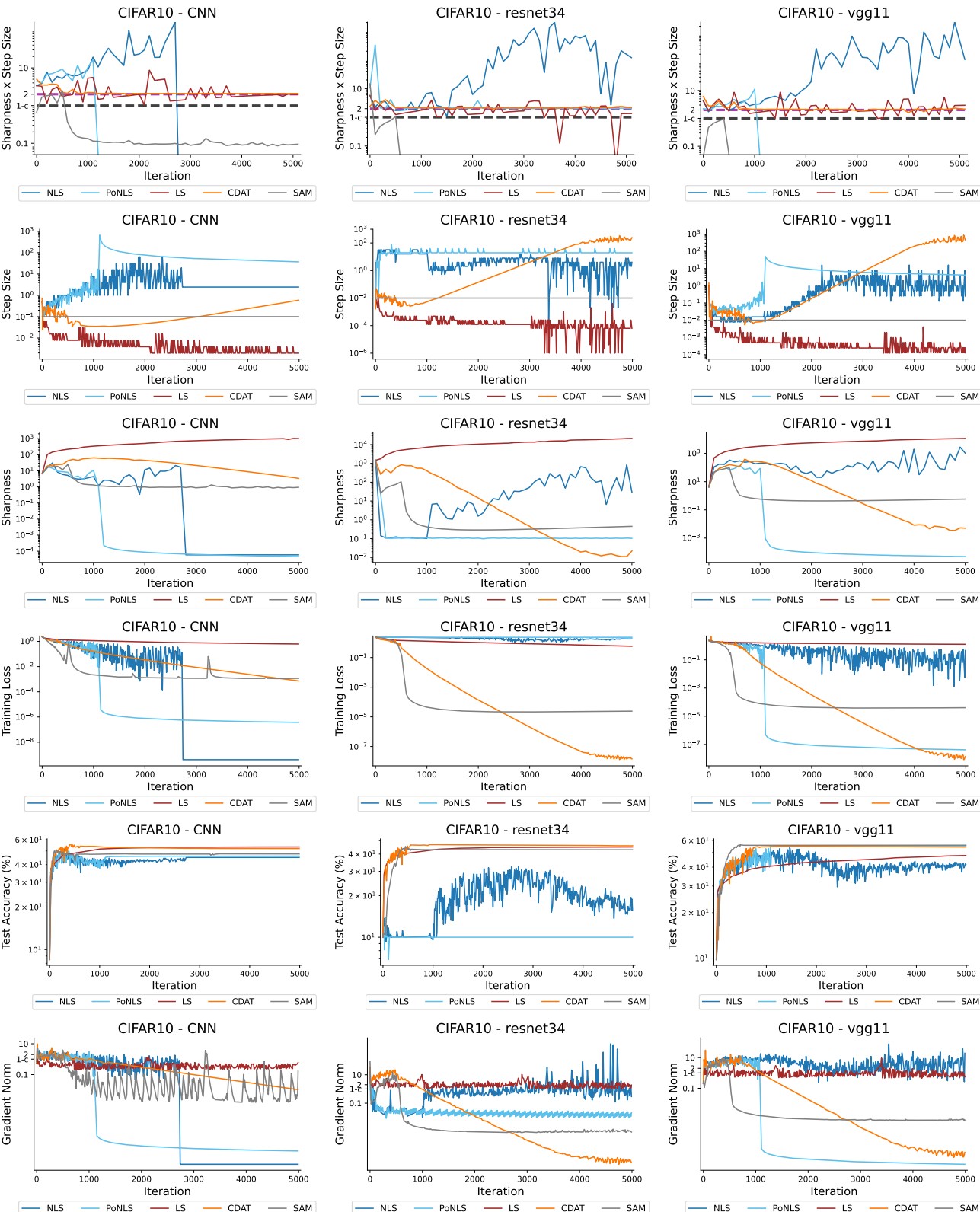

*Figure 11.* We plot the sharpness * step size (1st Row), step size (2nd Row), sharpness (3rd Row), training loss (4th Row), test accuracy (5th Row), and gradient norm (6th Row) for five different methods. We compare gradient descent with the LS, NLS, and PoNLS line searches as well as gradient descent with the CDAT step size selection and SAM. This is repeated for the CNN, resnet34, and vgg11 models on the CIFAR10 dataset using the cross entropy loss.

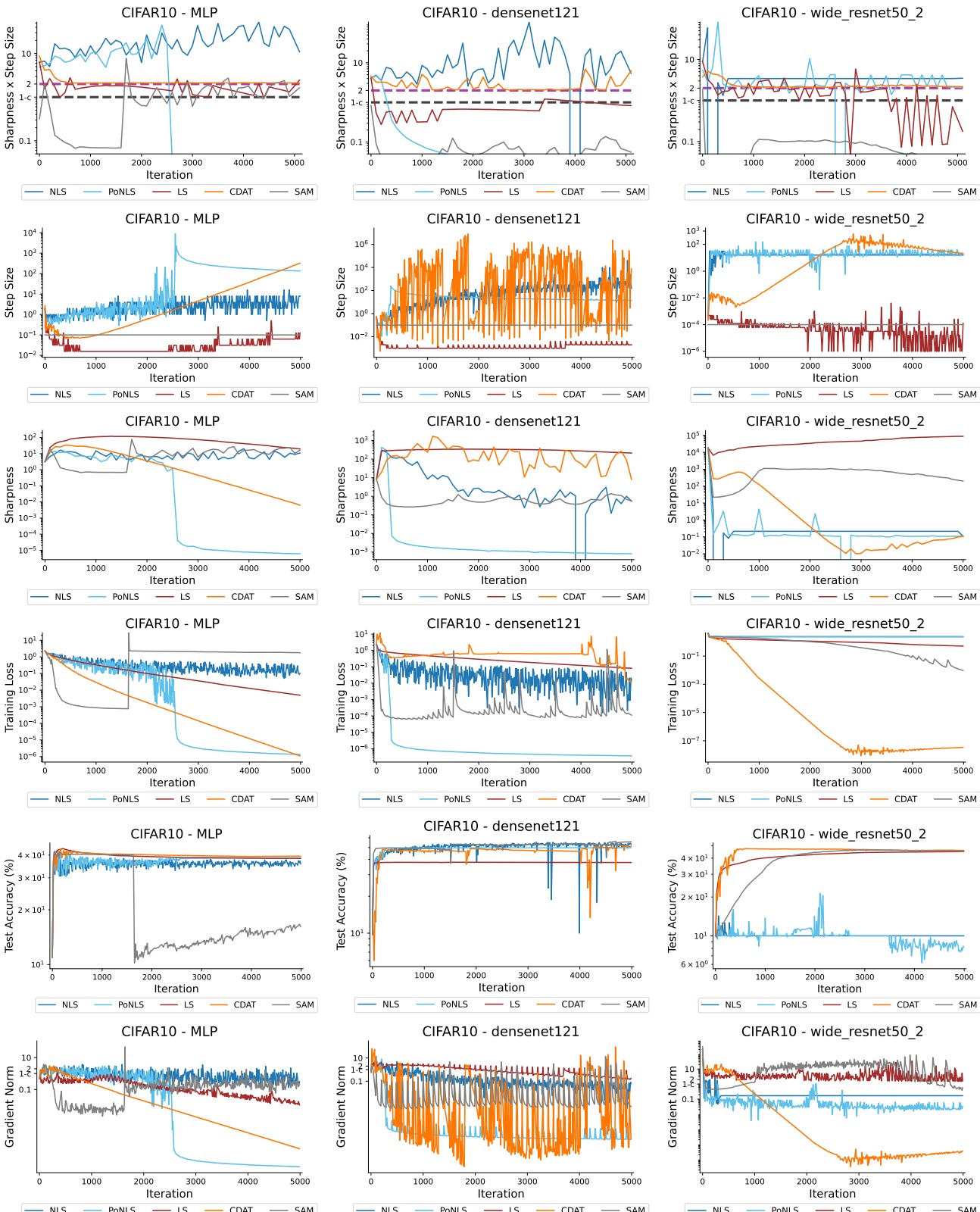

*Figure 12.* We plot the sharpness * step size (1st Row), step size (2nd Row), sharpness (3rd Row), training loss (4th Row), test accuracy (5th Row), and gradient norm (6th Row) for five different methods. We compare gradient descent with the LS, NLS, and PoNLS line searches as well as gradient descent with the CDAT step size selection and SAM. This is repeated for the MLP, densenet121, and wide_resnet50_2 models on the CIFAR10 dataset using the cross entropy loss.

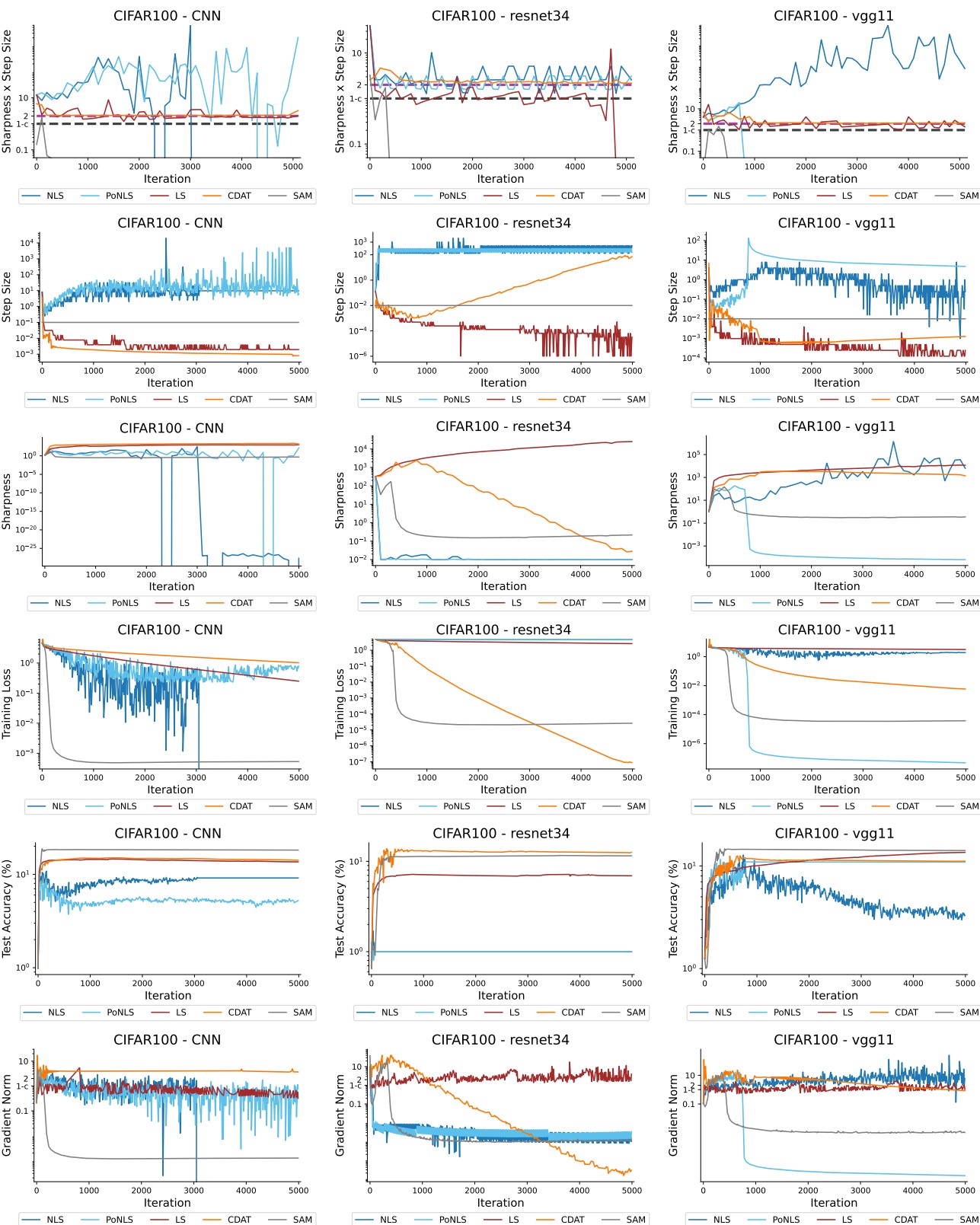

*Figure 13.* We plot the sharpness * step size (1st Row), step size (2nd Row), sharpness (3rd Row), training loss (4th Row), test accuracy (5th Row), and gradient norm (6th Row) for five different methods. We compare gradient descent with the LS, NLS, and PoNLS line searches as well as gradient descent with the CDAT step size selection and SAM. This is repeated for the CNN, resnet34, and vgg11 models on the CIFAR100 dataset using the cross entropy loss.

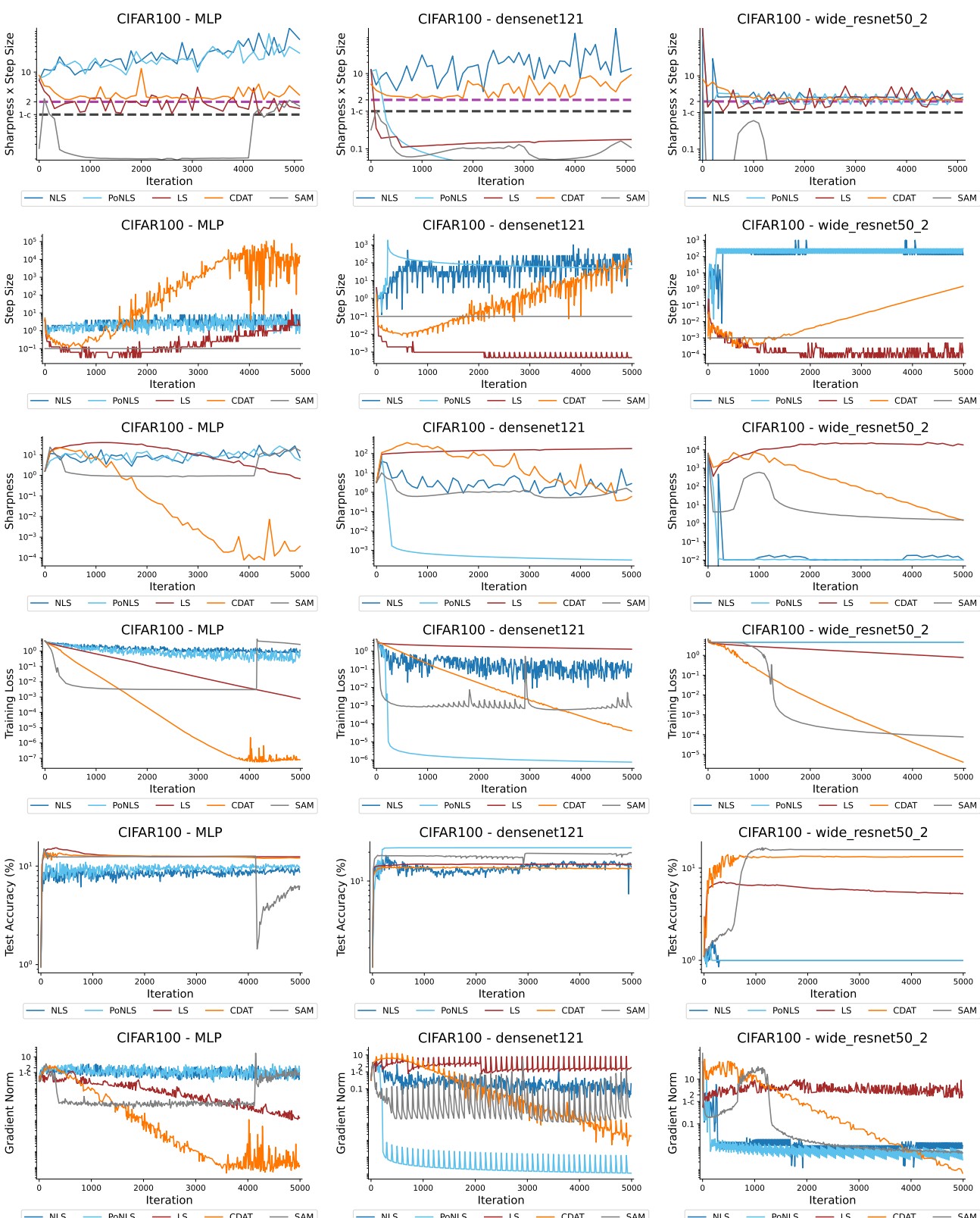

*Figure 14.* We plot the sharpness * step size (1st Row), step size (2nd Row), sharpness (3rd Row), training loss (4th Row), test accuracy (5th Row), and gradient norm (6th Row) for five different methods. We compare gradient descent with the LS, NLS, and PoNLS line searches as well as gradient descent with the CDAT step size selection and SAM. This is repeated for the MLP, densenet121, and wide_resnet50_2 models on the CIFAR100 dataset using the cross entropy loss.

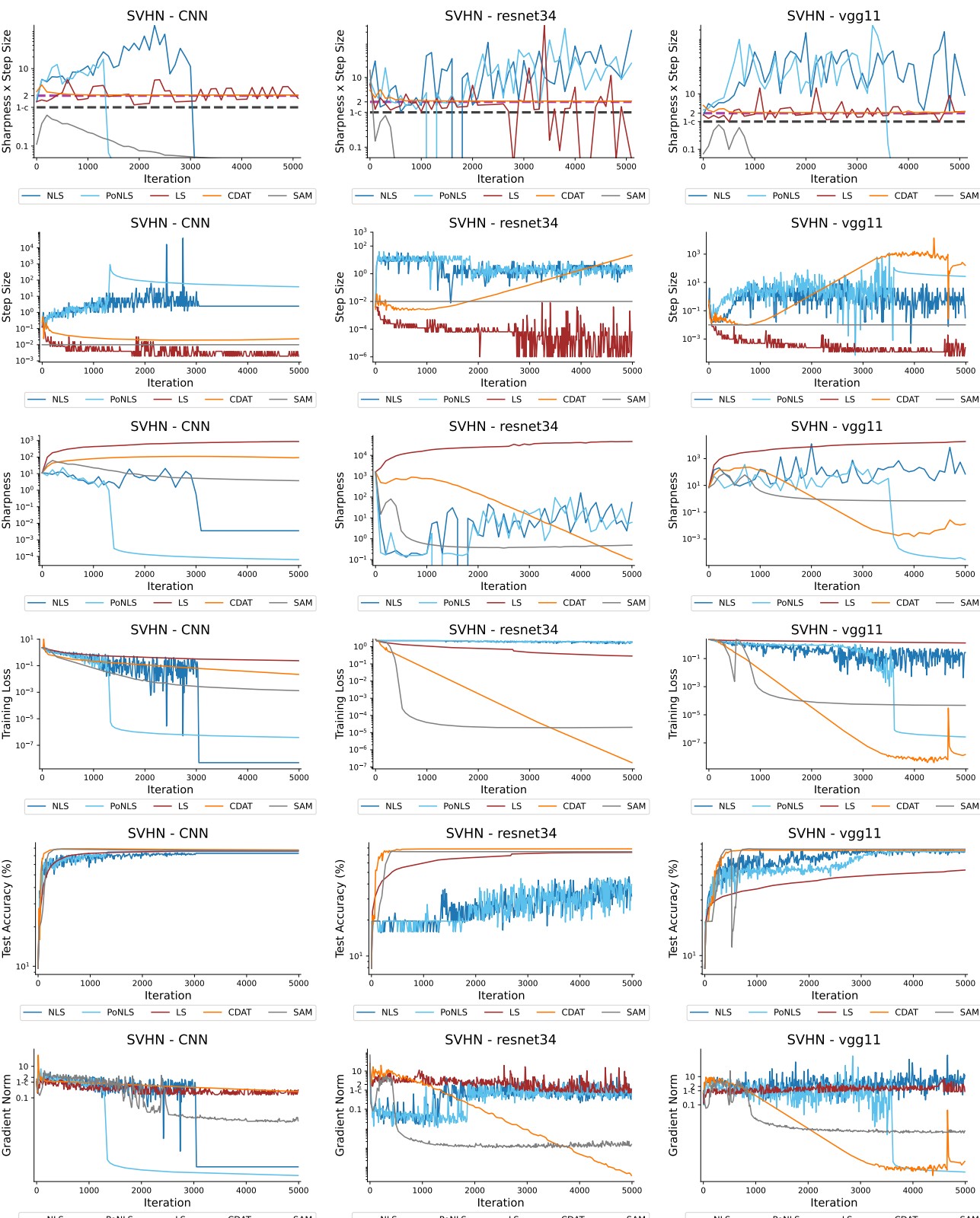

*Figure 15.* We plot the sharpness * step size (1st Row), step size (2nd Row), sharpness (3rd Row), training loss (4th Row), test accuracy (5th Row), and gradient norm (6th Row) for five different methods.. We compare gradient descent with the LS, NLS, and PoNLS line searches as well as gradient descent with the CDAT step size selection and SAM. This is repeated for the CNN, resnet34, and vgg11 models on the SVHN dataset using the cross entropy loss.

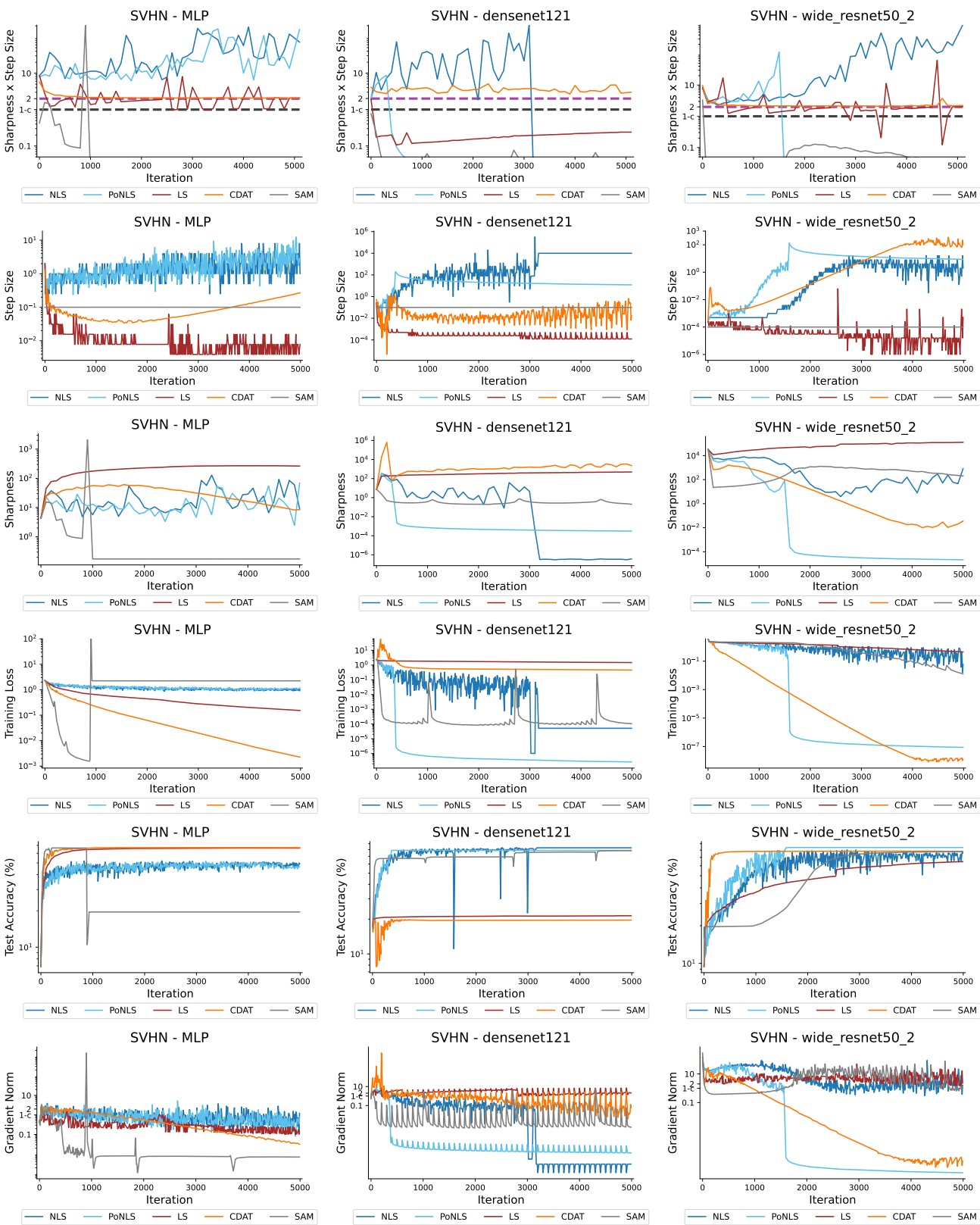

*Figure 16.* We plot the sharpness * step size (1st Row), step size (2nd Row), sharpness (3rd Row), training loss (4th Row), test accuracy (5th Row), and gradient norm (6th Row) for five different methods.. We compare gradient descent with the LS, NLS, and PoNLS line searches as well as gradient descent with the CDAT step size selection and SAM. This is repeated for the MLP, densenet121, and wide_resnet50_2 models on the SVHN dataset using the cross entropy loss.

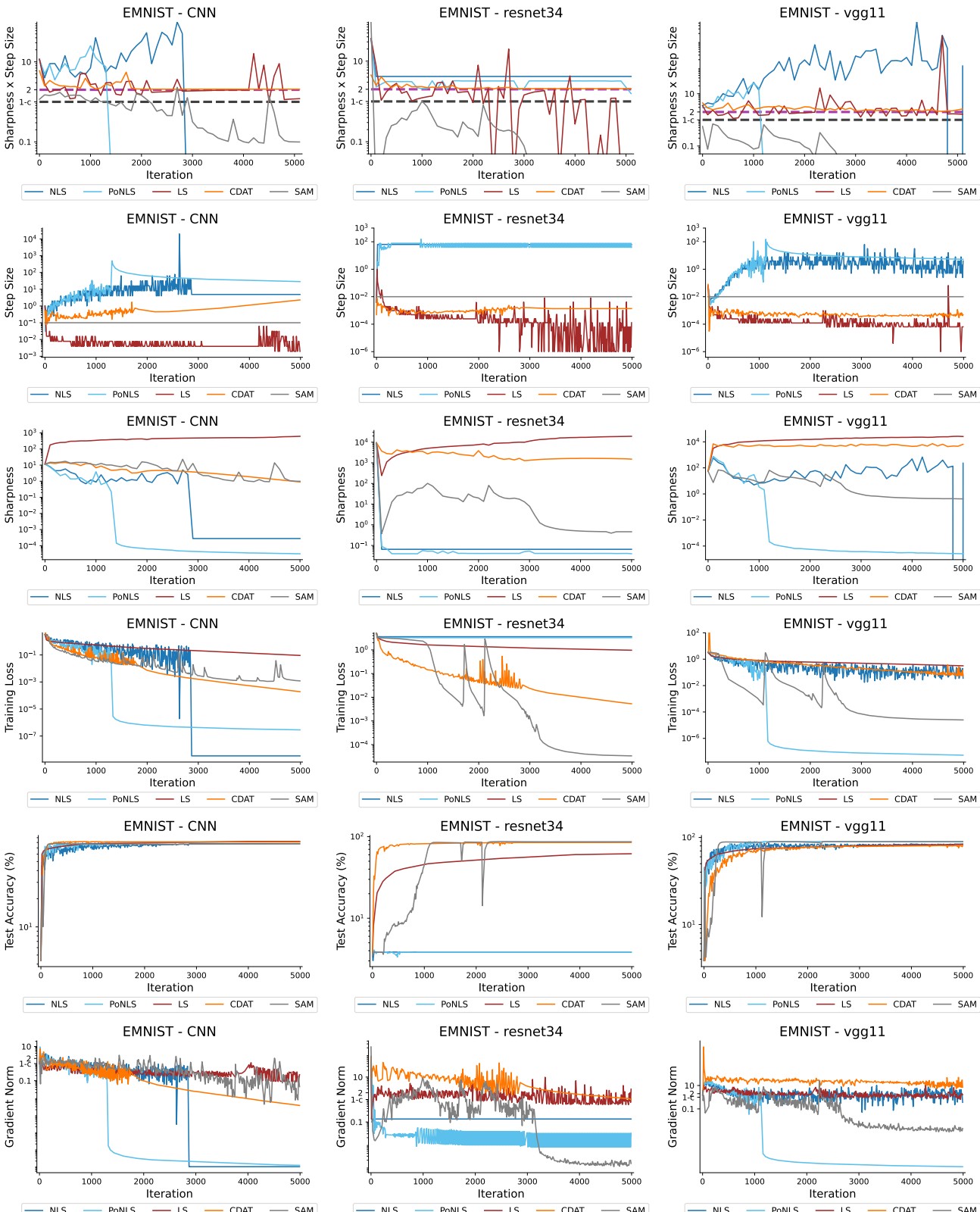

*Figure 17.* We plot the sharpness * step size (1st Row), step size (2nd Row), sharpness (3rd Row), training loss (4th Row), test accuracy (5th Row), and gradient norm (6th Row) for five different methods. We compare gradient descent with the LS, NLS, and PoNLS line searches as well as gradient descent with the CDAT step size selection and SAM. This is repeated for the CNN, resnet34, and vgg11 models on the EMNIST dataset using the cross entropy loss.

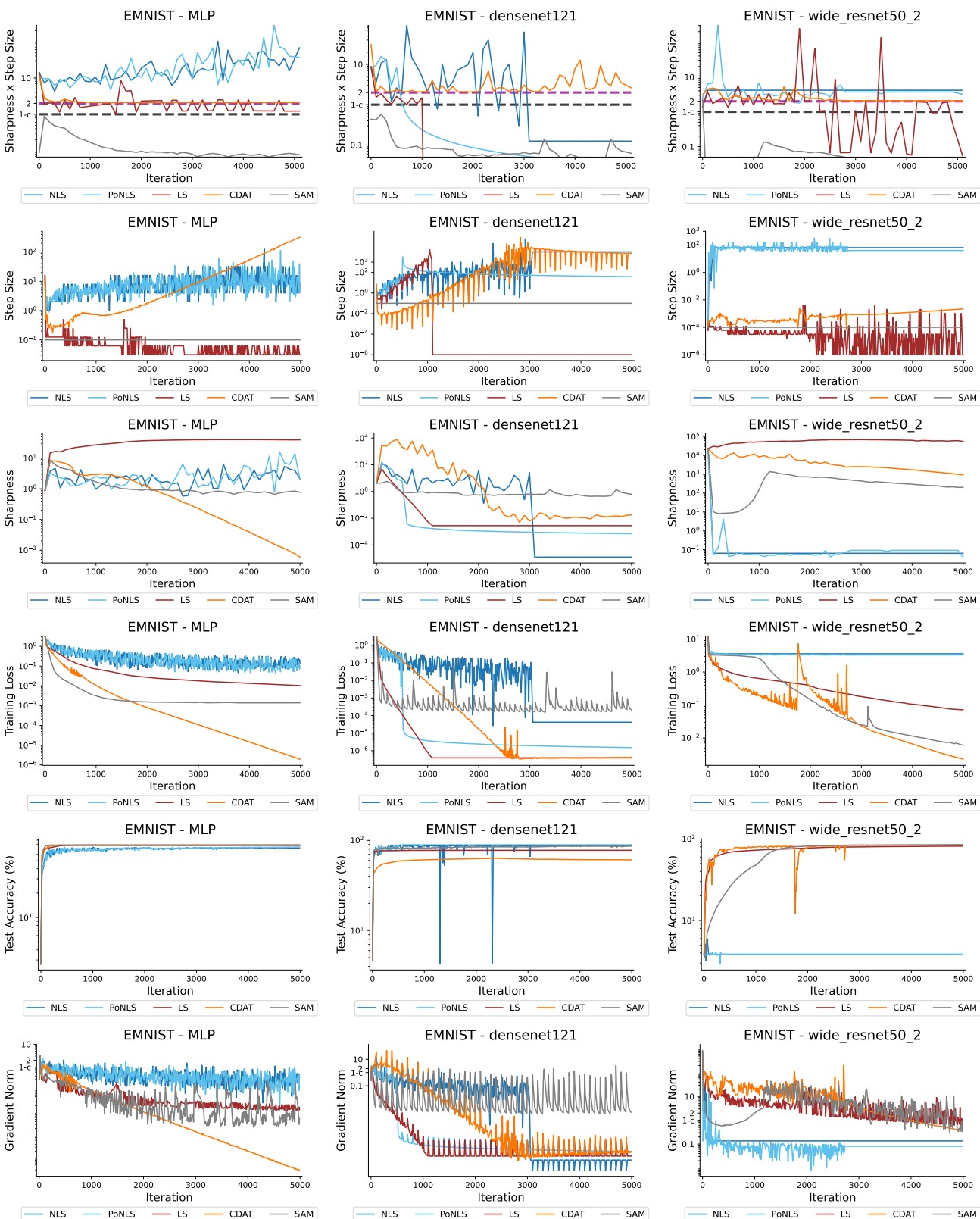

*Figure 18.* We plot the sharpness * step size (1st Row), step size (2nd Row), sharpness (3rd Row), training loss (4th Row), test accuracy (5th Row), and gradient norm (6th Row) for five different methods. We compare gradient descent with the LS, NLS, and PoNLS line searches as well as gradient descent with the CDAT step size selection and SAM. This is repeated for the MLP, densenet121, and wide_resnet50_2 models on the EMNIST dataset using the cross entropy loss.

# I. Additional Experiments

## I.1. Additional Sharpness Aware Method Experiments

In this section, we provide additional experiments comparing line searches with CDAT and SAM on various metrics using the MSE loss. Continuing our discussion in Section 3, we now give more details on the LS method. We see from the first row of Figure 19 that the step sizes yielded by a monotone line search method LS are not always above the threshold of $\frac{1-c}{\lambda_1(H(w_k))}$. Despite the fact that Proposition 2.5 also holds for monotone line searches, we find that LS encounters cancellation errors. This occurs when $f(w_{k+1})$ is numerically identical to $f(w_k)$, and the backtracking procedure continues to reduce the step size $\eta_k$ until it reaches the safeguard value of $10^{-6}$. Nonmonotone line searches of the type proposed in Zhang & Hager (2004) avoid this issue by default, as $\epsilon_k > 0 \; \forall k$. Focusing on the sharpness values obtained by the monotone line search, we notice that they do not flatten out and continue increasing as previously noticed in the work by Roulet et al. (2024). Thanks to Proposition 2.5, more precisely the Corollary B.16 in the Appendix, and the reverse bound $\max_{\eta \in [0,\eta_{w_k}]} \lambda_1(H_\eta(w_k)) \geq \frac{1-c}{\eta_k}$, we can now provide a full picture of the behavior of LS. While the sharpness increases, one can notice from the plots in the second row that the step size $\eta_k$ chosen by LS decreases slowly and nonmonotonically (when not affected by cancellation errors). In view of the reverse bound, this forces the maximum sharpness along the segment $[0, \eta_{w_k}]$ to increase. In other words, the monotone requirement of the line search forces the method to enter sharper and sharper valleys by reducing the step size. The nonmonotone line search instead takes larger steps and allows for increases in the training loss as one can see in Figure 19, which prevents the method from entering sharper regions. In summary, NLS maintains the optimization trajectory on flatter grounds, with sharpness values that are various magnitudes lower than that of LS.

From the second row of Figures 19-26, one can see that the step sizes yielded by NLS and PoNLS are upper bounded by $2^{i_K}$, where $i_K$ is the smallest integer $i$ such that $2^i \geq K$. Loosely speaking, when $\min_{w \in \mathbb{R}^n} \lambda_1(H(w)) = 2/K$ and $\eta_k > K$, there do not exist flat enough valleys that can contain the oscillations of GD with step size $\eta_k$. In other words, GD with $\eta_k > K$ will always hit a "wall" of the valley and the line search will consequently reduce the step size.

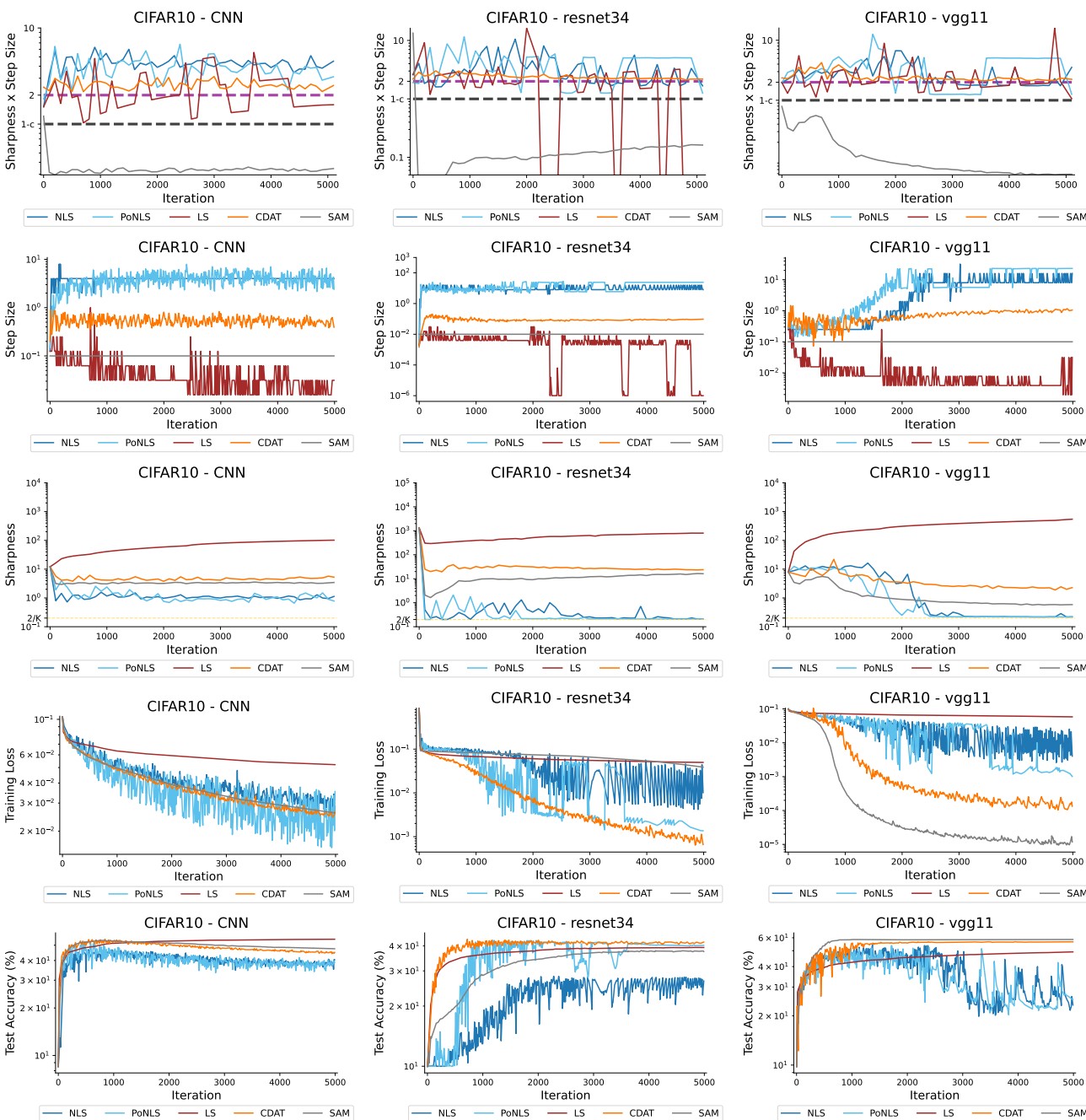

*Figure 19.* We plot the sharpness * step size (1st Row), step size (2nd Row), sharpness (3rd Row), training loss (4th Row), and test accuracy (5th Row) for five different methods. We compare gradient descent with the LS, NLS, and PoNLS line searches as well as gradient descent with the CDAT step size selection and SAM. This is repeated for the CNN, resnet34, and vgg11 models on the CIFAR10 dataset.

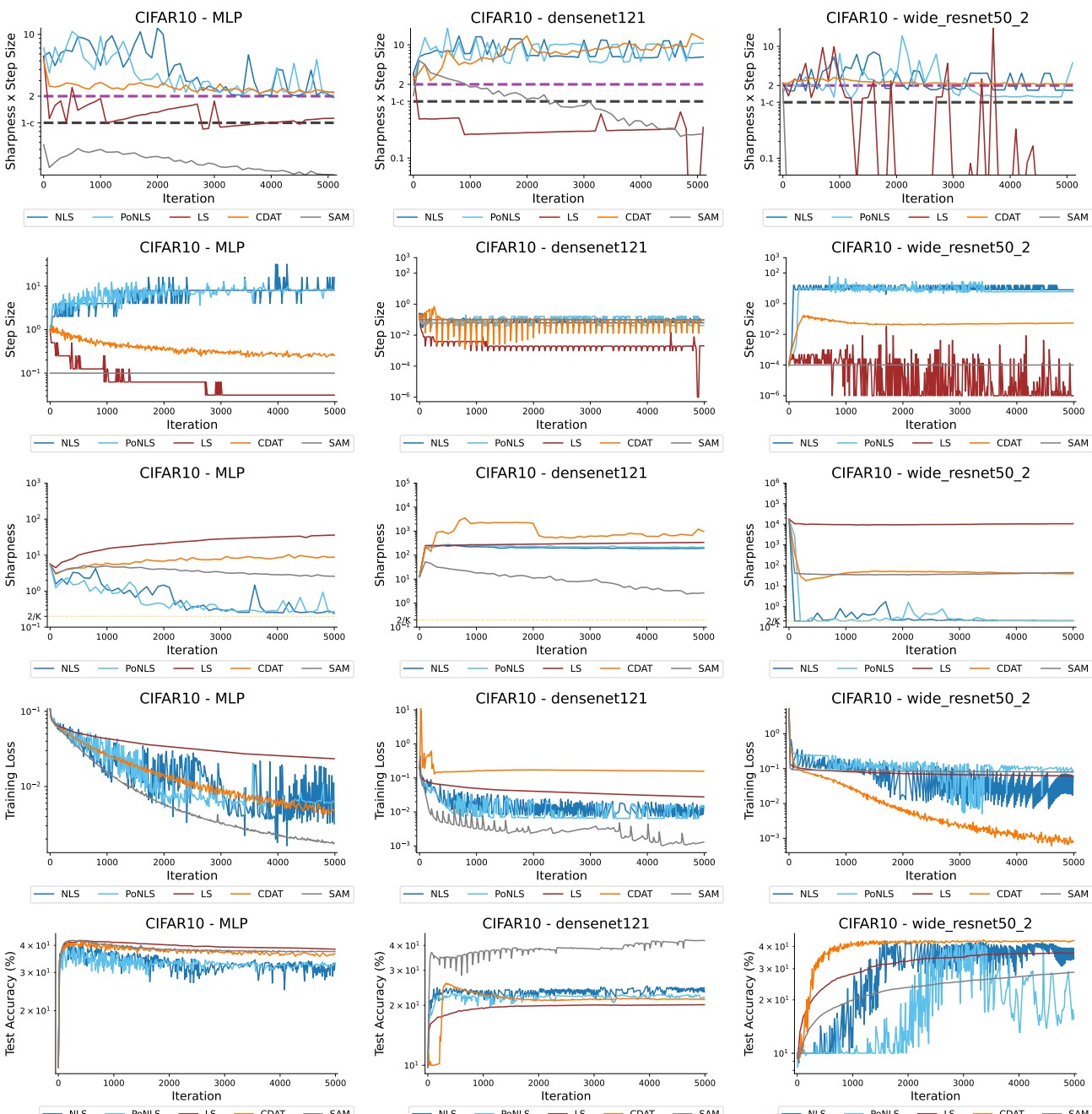

*Figure 20.* We plot the sharpness * step size (1st Row), step size (2nd Row), sharpness (3rd Row), training loss (4th Row), and test accuracy (5th Row) for five different methods. We compare gradient descent with the LS, NLS, and PoNLS line searches as well as gradient descent with the CDAT step size selection and SAM. This is repeated for the MLP, densenet121, and wide_resnet50_2 models on the CIFAR10 dataset.

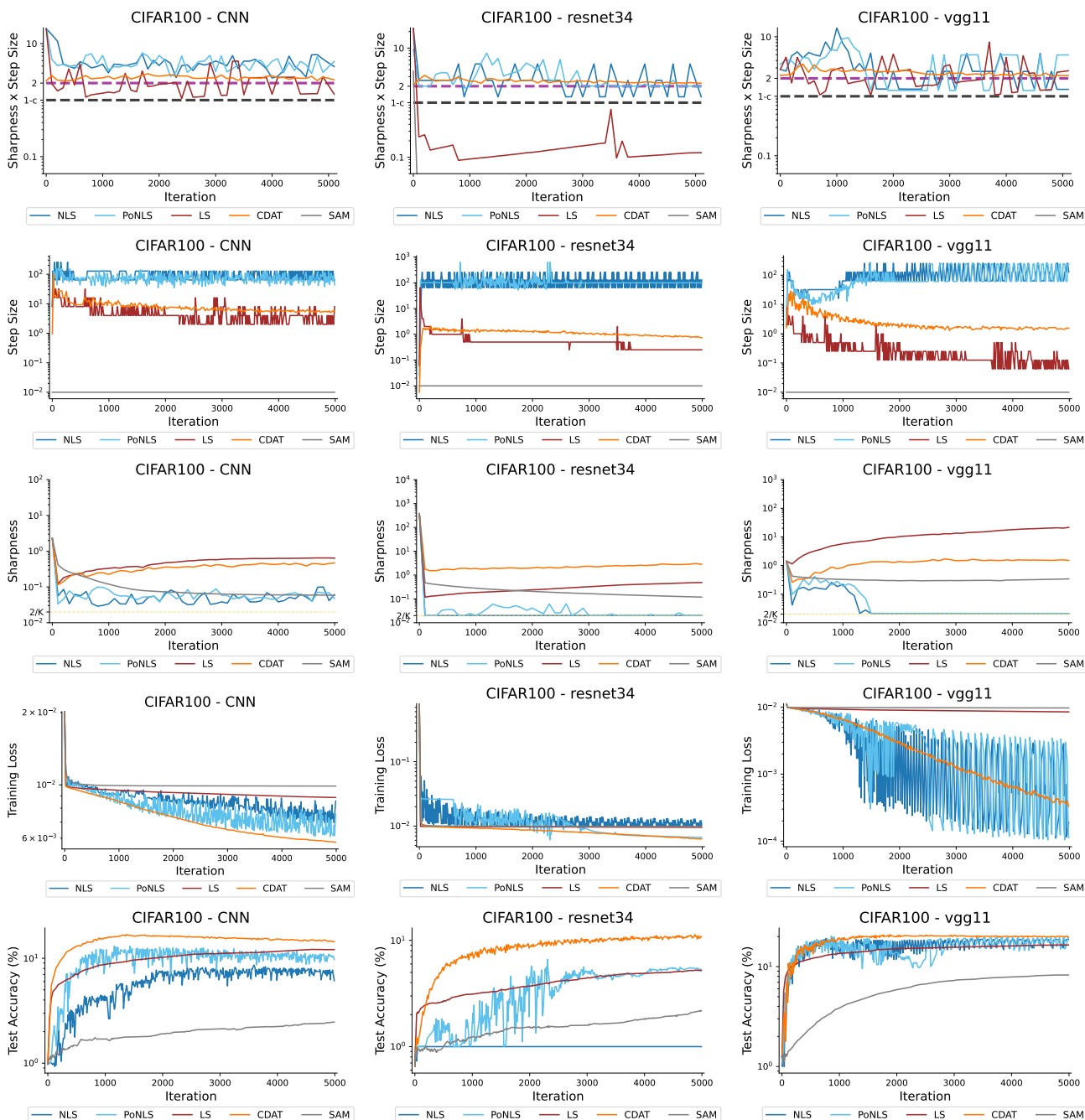

*Figure 21.* We plot the sharpness * step size (1st Row), step size (2nd Row), sharpness (3rd Row), training loss (4th Row), and test accuracy (5th Row) for five different methods. We compare gradient descent with the LS, NLS, and PoNLS line searches as well as gradient descent with the CDAT step size selection and SAM. This is repeated for the CNN, resnet34, and vgg11 models on the CIFAR100 dataset.

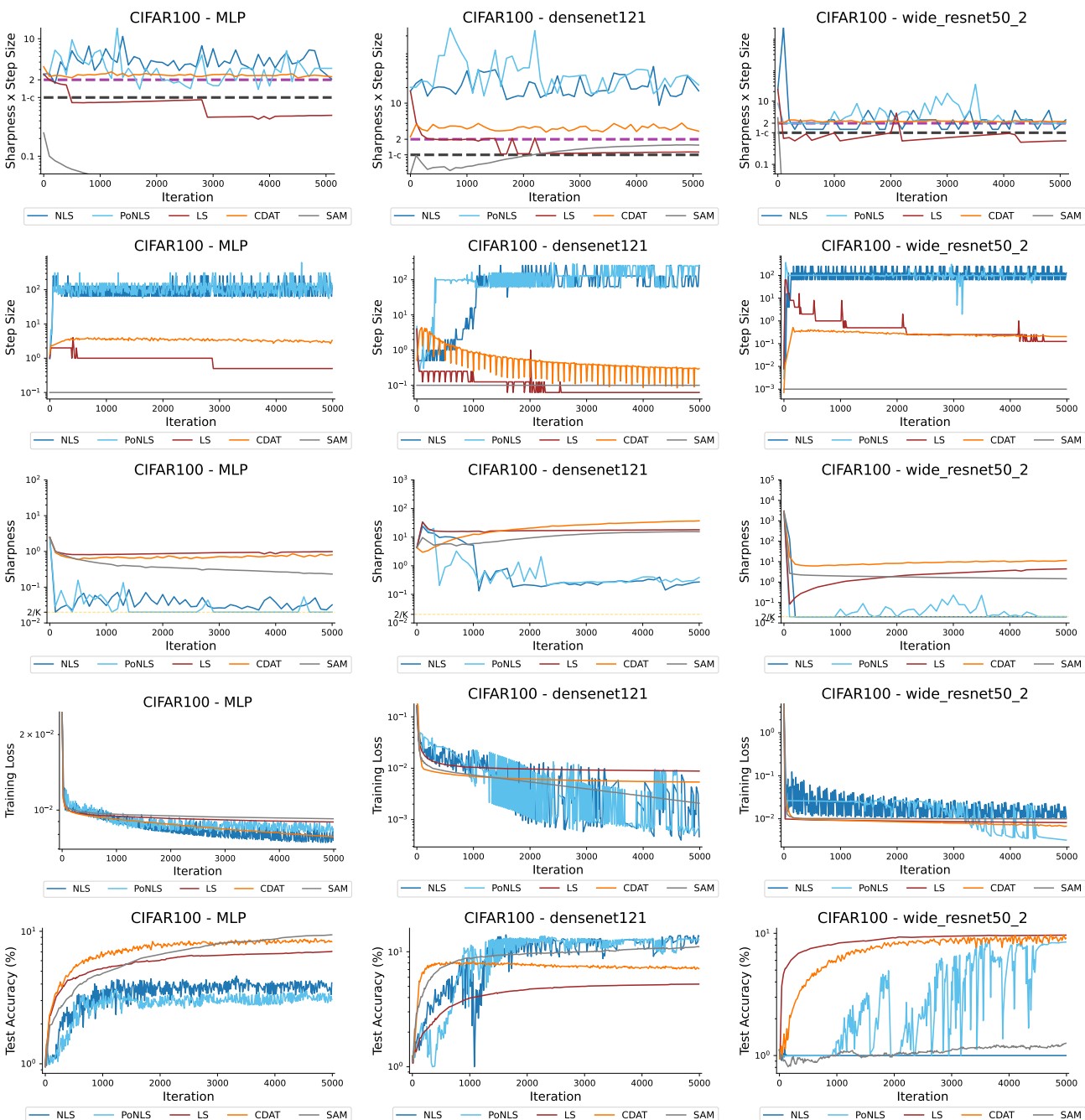

*Figure 22.* We plot the sharpness * step size (1st Row), step size (2nd Row), sharpness (3rd Row), training loss (4th Row), and test accuracy (5th Row) for five different methods. We compare gradient descent with the LS, NLS, and PoNLS line searches as well as gradient descent with the CDAT step size selection and SAM. This is repeated for the MLP, densenet121, and wide_resnet50_2 models on the CIFAR100 dataset.

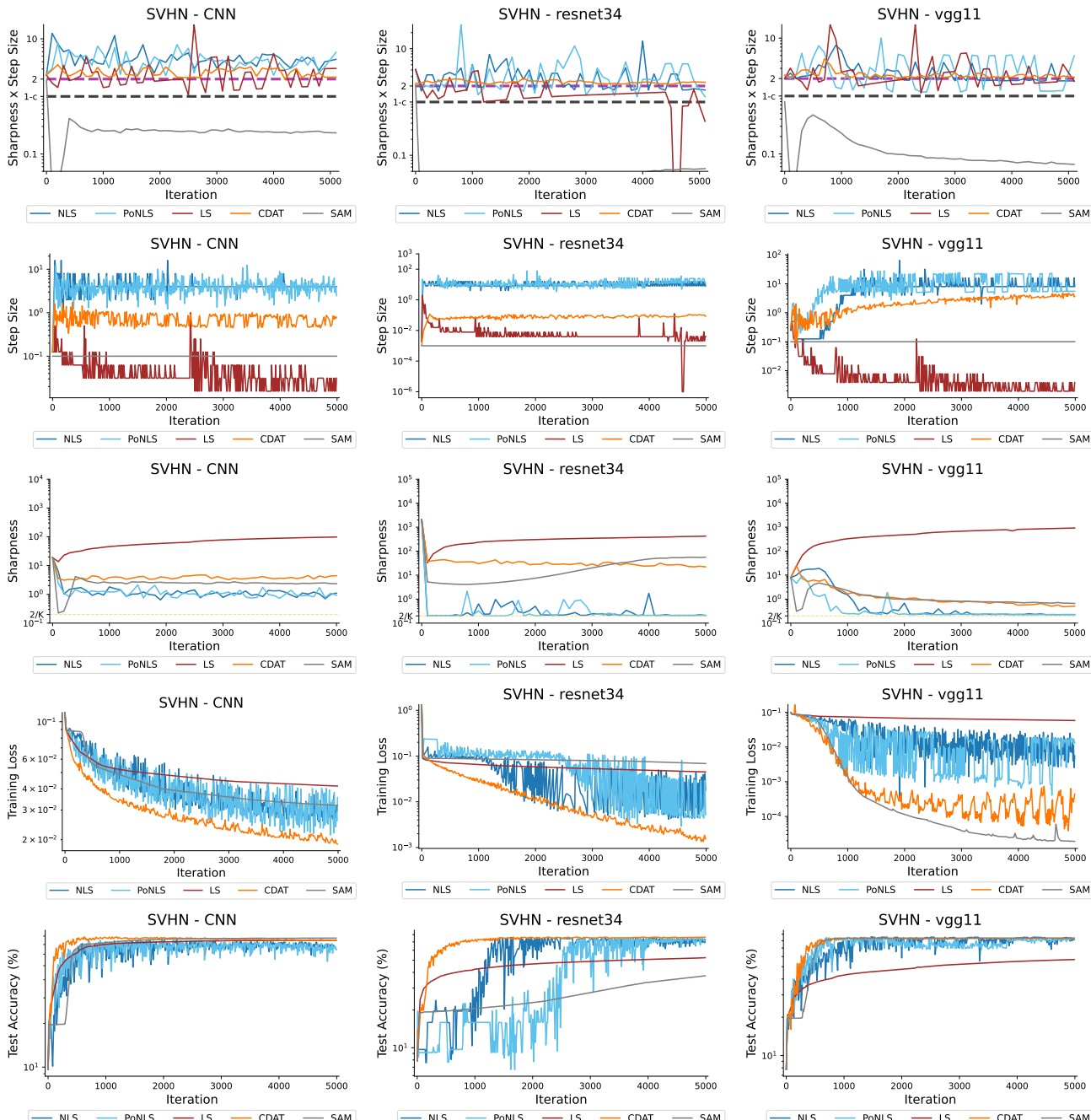

*Figure 23.* We plot the sharpness * step size (1st Row), step size (2nd Row), sharpness (3rd Row), training loss (4th Row), and test accuracy (5th Row) for five different methods. We compare gradient descent with the LS, NLS, and PoNLS line searches as well as gradient descent with the CDAT step size selection and SAM. This is repeated for the CNN, resnet34, and vgg11 models on the SVHN dataset.

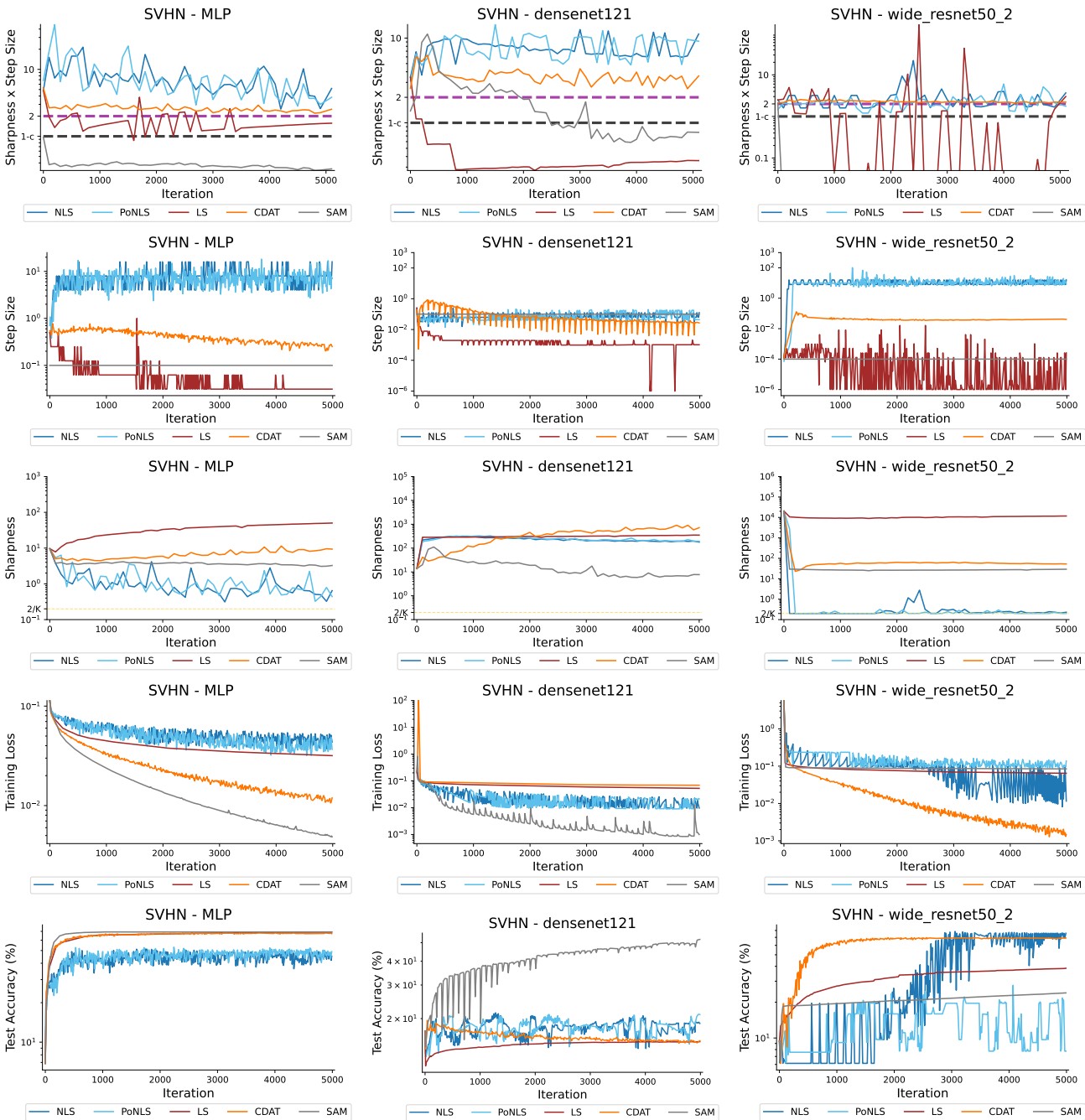

*Figure 24.* We plot the sharpness * step size (1st Row), step size (2nd Row), sharpness (3rd Row), training loss (4th Row), and test accuracy (5th Row) for five different methods. We compare gradient descent with the LS, NLS, and PoNLS line searches as well as gradient descent with the CDAT step size selection and SAM. This is repeated for the MLP, densenet121, and wide_resnet50_2 models on the SVHN dataset.

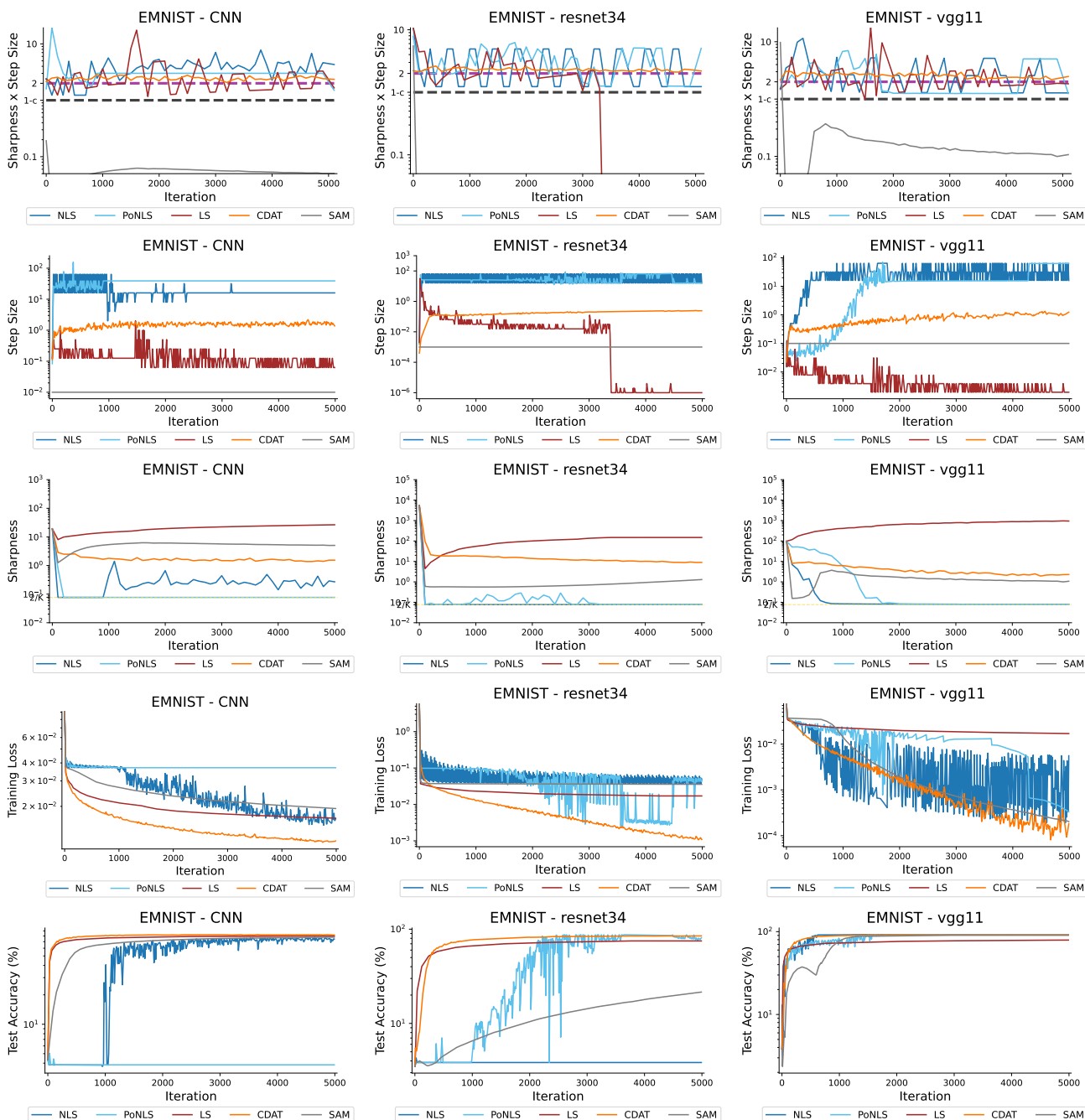

*Figure 25.* We plot the sharpness * step size (1st Row), step size (2nd Row), sharpness (3rd Row), training loss (4th Row), and test accuracy (5th Row) for five different methods. We compare gradient descent with the LS, NLS, and PoNLS line searches as well as gradient descent with the CDAT step size selection and SAM. This is repeated for the CNN, resnet34, and vgg11 models on the EMNIST dataset.

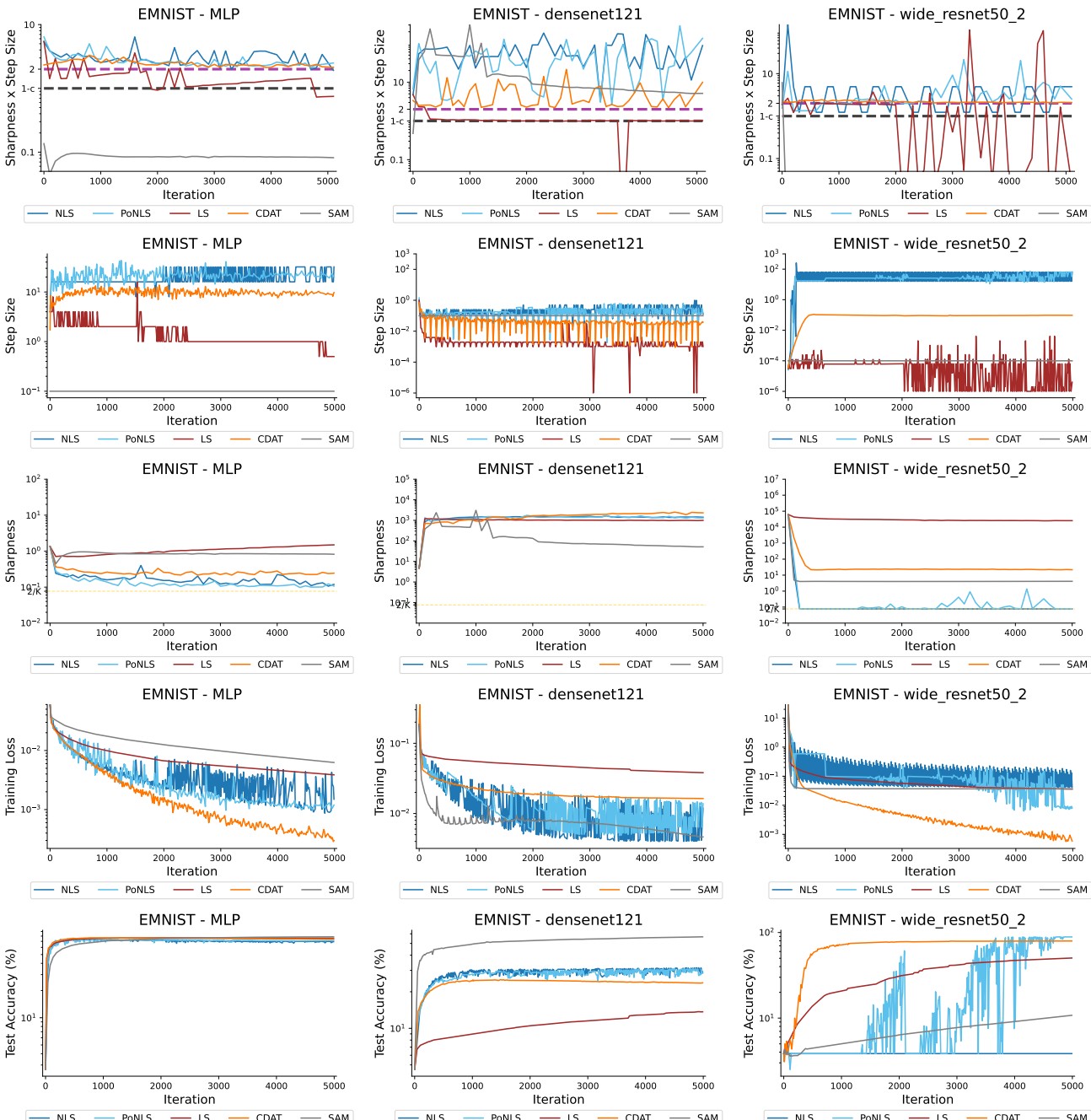

*Figure 26.* We plot the sharpness * step size (1st Row), step size (2nd Row), sharpness (3rd Row), training loss (4th Row), and test accuracy (5th Row) for five different methods. We compare gradient descent with the LS, NLS, and PoNLS line searches as well as gradient descent with the CDAT step size selection and SAM. This is repeated for the MLP, densenet121, and wide_resnet50_2 models on the EMNIST dataset.

### I.2. Additional Experiments on Characterizing Globally-Flat Points

In Figures 27 through 31, we provide additional experiments that focus on a small number of iterations to perform a finer analysis of the points where the sharpness reaches $2/K$, where $K$ is the number of classes in the dataset.

Note that for all the experiments in this section (and Figure 2 in the main paper), an activation is considered approximately zero if its value is less than $10^{-6}$. In particular, the percentage of approximately zero activations in row 3 is computed over all neurons of the layer and over all the points in the 5000 subset of the dataset. Additionally, in the layer-wise gradient norm

plots, the upper bound of the shaded region corresponds to the maximum per layer gradient norm across all the hidden layers. The lower bound similarly corresponds to the minimum. From Figures 27-30, we can observe that while for resnet34 and wide_resnet5_2 the sharpness always precisely hits $2/K$, this value is slightly above it in the case of the vgg11. This suggests that the possibility of precisely finding (or not fidning) the globally-flat saddle points may be architecture-dependent. In fact, all the runs of NLS in our benchmark on resnet34 and wide_resnet5_2 are unsuccessfully-flat, while all of those on vgg11 are successfully-flat. Figure 31 shows that the globally-flat points can also be found for the CNN model, while they were not precisely found for the MLP architecture. Interestingly enough, from our experiments it appears that the higher the number of classes, the easier it is for NLS to find globally-flat points.

One further characteristic that differentiates successfully- and unsuccessfully-flat experiments can be observed in the fourth row of Figures 27-31. While not reported in the main paper, this plot shows that the percentage of zero entries of the gradient (this time over the whole gradient, and not per layer) is often higher for unsuccessfully-flat trainings than for successfully-flat ones. Notice that in this case, with zero entries we mean exactly 0 and not below a certain threshold.

Finally, observe from the first row of these figures that the trace sometimes suddenly decreases to small values below the shown y-axis limit, such as for CIFAR10×resnet34, CIFAR100×resnet34, and SVHN×vgg11 experiments. This occurs when the trace becomes negative on these iterations. Notice that is not a contradiction to Lemma B.23, as the sum of the eigenvalues may become negative for all points that are not globally-flat.

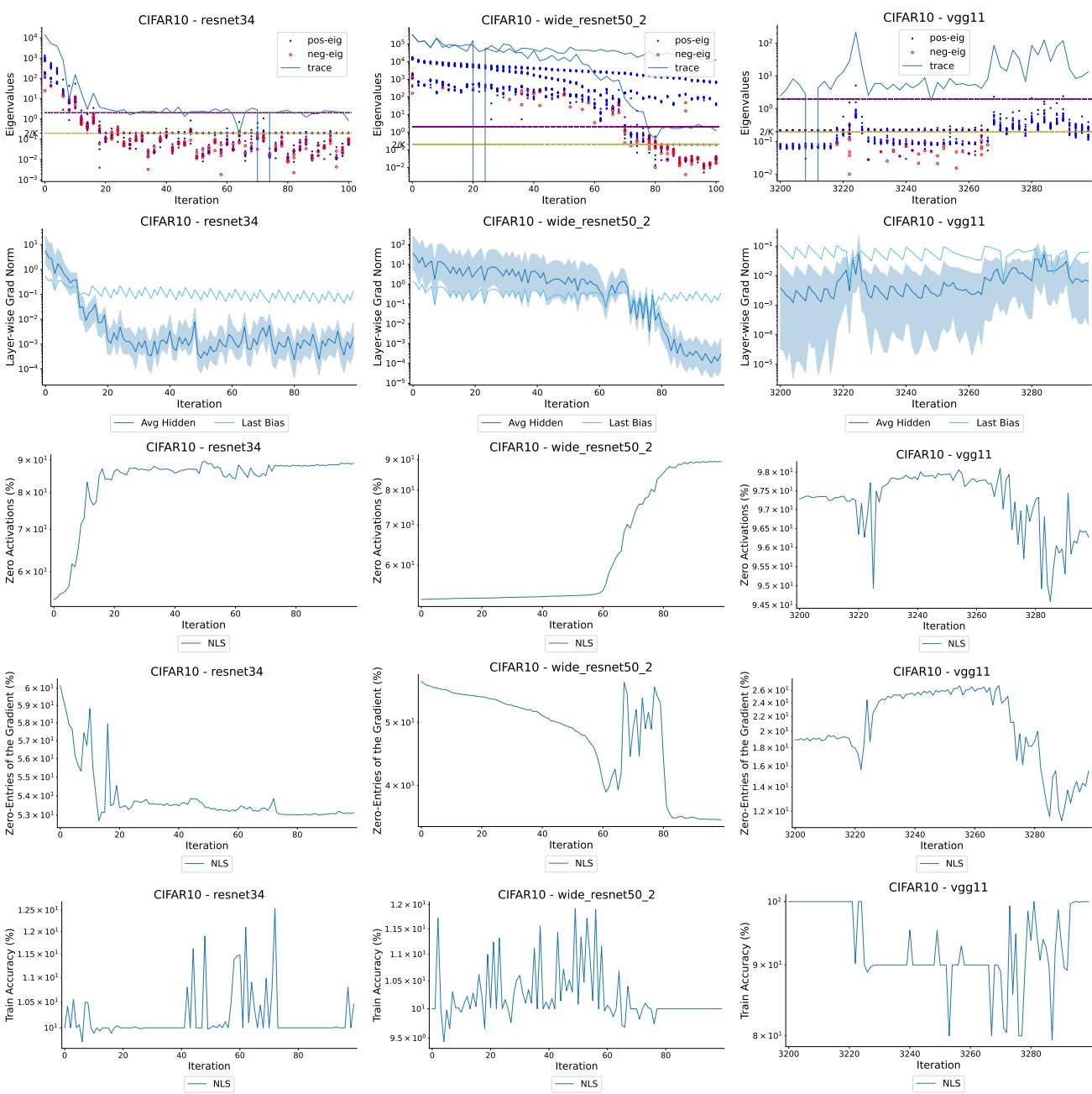

*Figure 27.* We plot the top 20 eigenvalues (1st row), layer-wise gradient norm for the hidden layers (which we average across all hidden layers) and gradient norm of the bias parameters in the last layer (2nd Row), the maximum per layer percentage of neural activations approximately equal to zero (3rd Row), the percentage of the gradient entries equal to 0 (4th Row), and training accuracy (5th Row) for the NLS line search method. This is repeated for the resnet34 (first 100 iterations), wide_resnet50_2 (first 100 iterations), and vgg11 (iterations 3200-3300) models on the CIFAR10 dataset.

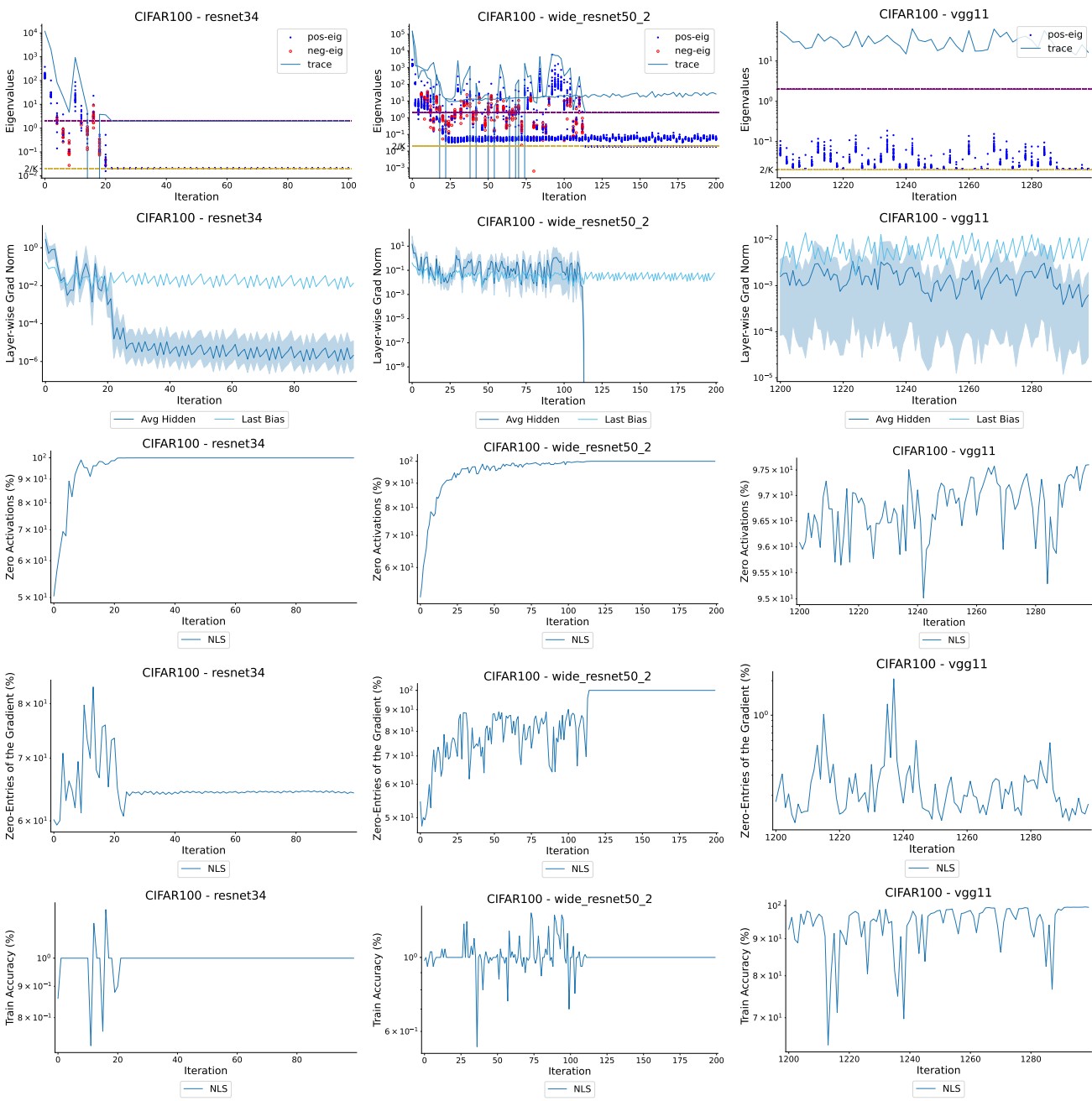

*Figure 28.* We plot the top 20 eigenvalues (1st row), layer-wise gradient norm for the hidden layers (which we average across all hidden layers) and gradient norm of the bias parameters in the last layer (2nd Row), the maximum per layer percentage of neural activations approximately equal to zero (3rd Row), the percentage of the gradient entries equal to 0 (4th Row), and training accuracy (5th Row) for the NLS line search method. This is repeated for the resnet34 (first 100 iterations), wide_resnet50_2 (first 200 iterations), and vgg11 (iterations 1200-1300) models on the CIFAR100 dataset.

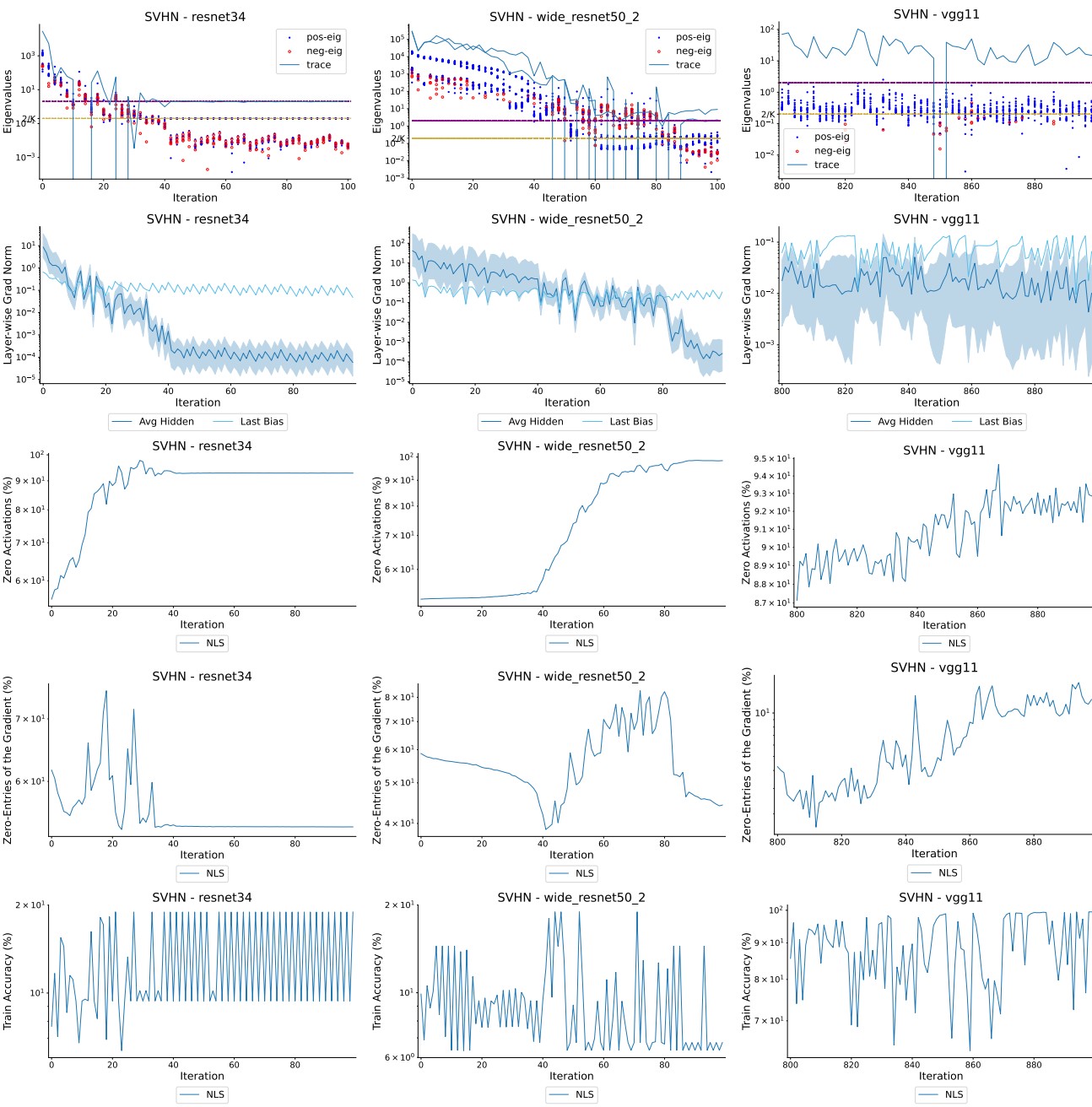

*Figure 29.* We plot the top 20 eigenvalues (1st row), layer-wise gradient norm for the hidden layers (which we average across all hidden layers) and gradient norm of the bias parameters in the last layer (2nd Row), the maximum per layer percentage of neural activations approximately equal to zero (3rd Row), the percentage of the gradient entries equal to 0 (4th Row), and training accuracy (5th Row) for the NLS line search method. This is repeated for the resnet34 (first 100 iterations), wide_resnet50_2 (first 100 iterations), and vgg11 (iterations 800-900) models on the SVHN dataset.

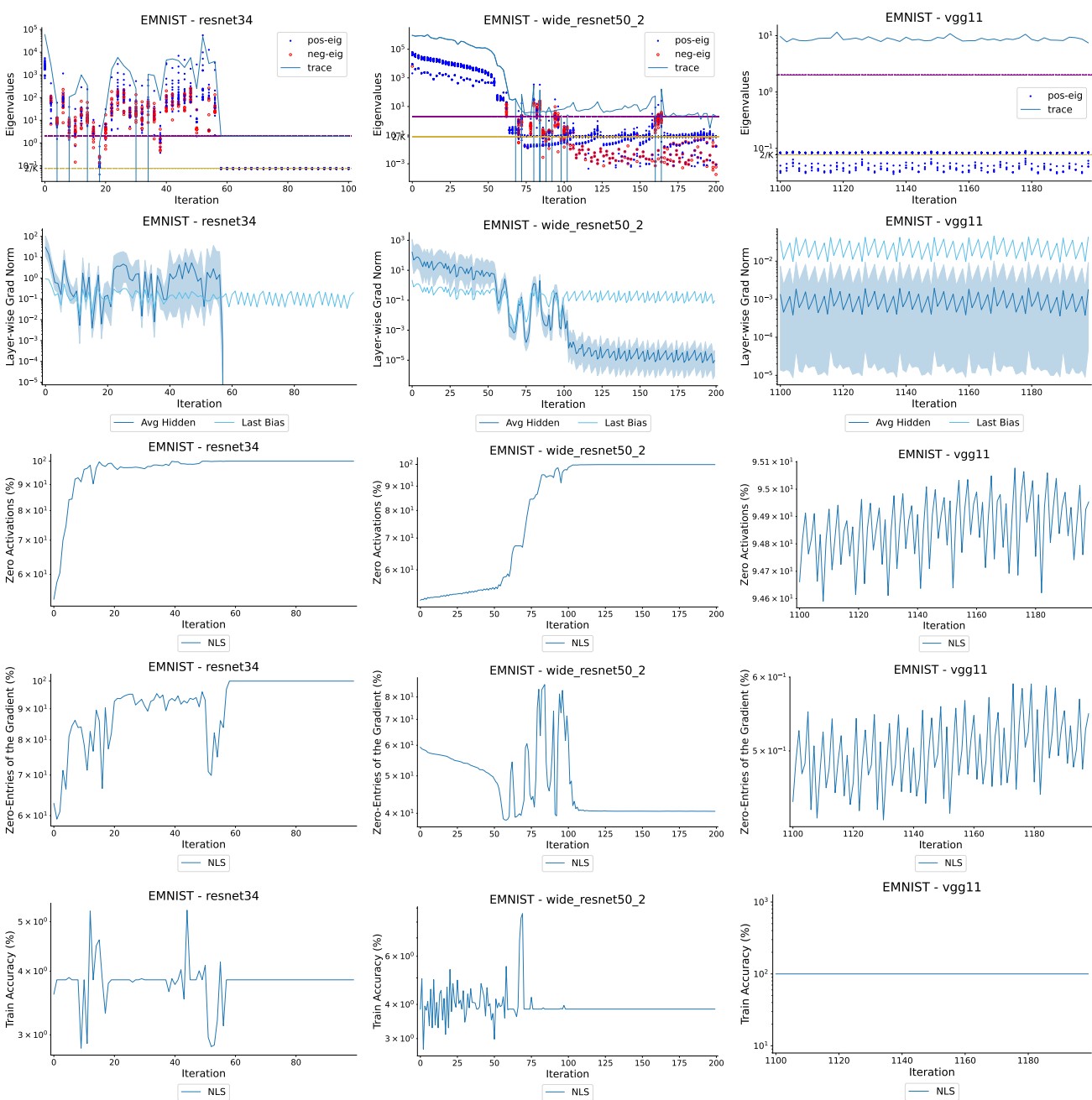

*Figure 30.* We plot the top 20 eigenvalues (1st row), layer-wise gradient norm for the hidden layers (which we average across all hidden layers) and gradient norm of the bias parameters in the last layer (2nd Row), the maximum per layer percentage of neural activations approximately equal to zero (3rd Row), the percentage of the gradient entries equal to 0 (4th Row), and training accuracy (5th Row) for the NLS line search method. This is repeated for the resnet34 (first 100 iterations), wide_resnet50_2 (first 100 iterations), and vgg11 (iterations 1100-1200) models on the EMNIST dataset.

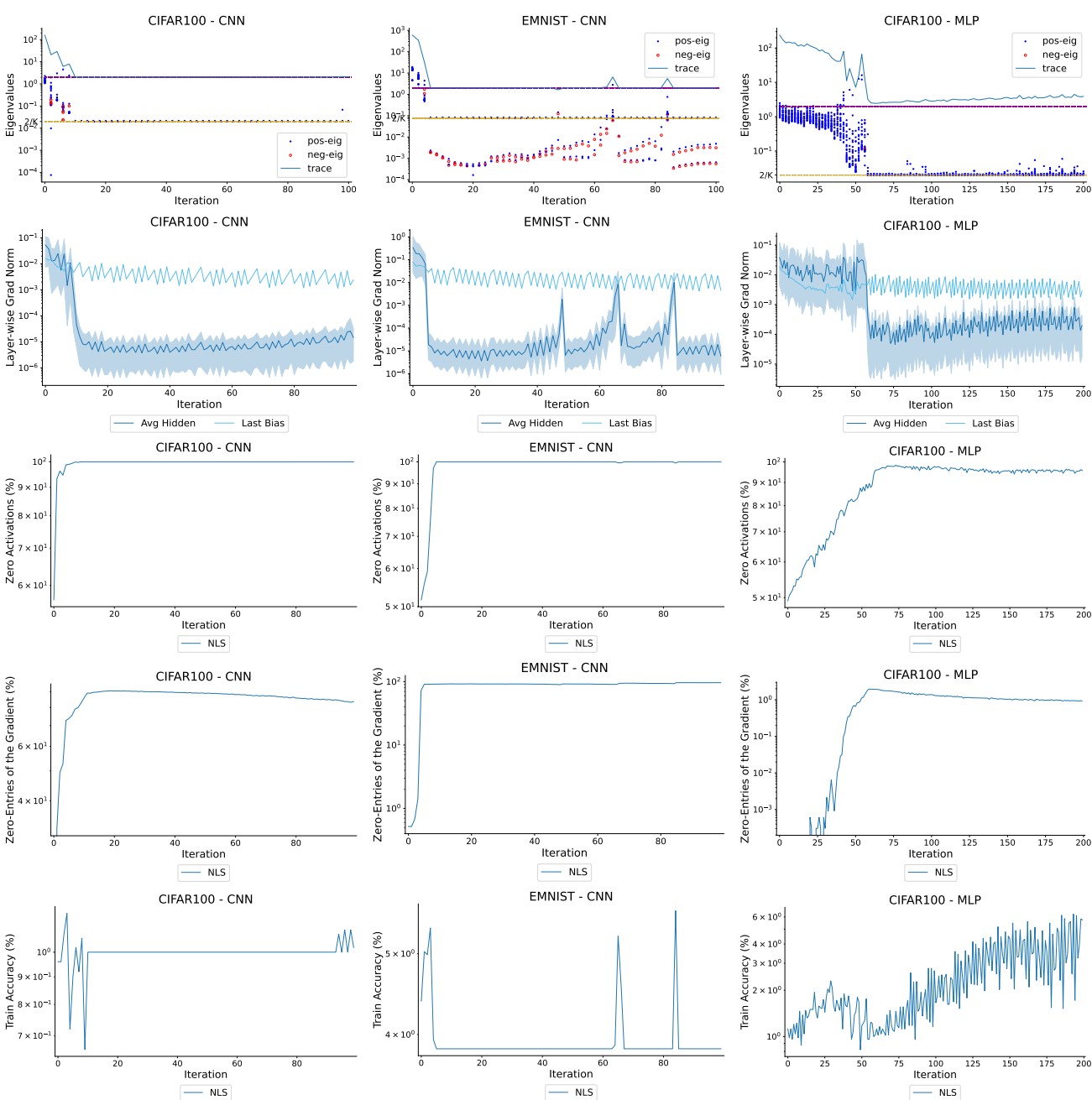

*Figure 31.* We plot the top 20 eigenvalues (1st row), layer-wise gradient norm for the hidden layers (which we average across all hidden layers) and gradient norm of the bias parameters in the last layer (2nd Row), the maximum per layer percentage of neural activations approximately equal to zero (3rd Row), the percentage of the gradient entries equal to 0 (4th Row), and training accuracy (5th Row) for the NLS line search method. This is repeated for the experiment CIFAR100×CNN (first 100 iterations), EMNIST×CNN (first 100 iterations), and CIFAR100×MLP (first 200 iterations).

## I.3. Additional Experiments on Avoiding the Globally-Flat Points

In Figures 32 through 39, we show further experiments highlighting the ability of NLS-ub to avoid globally flat points while improving upon the performance of NLS.

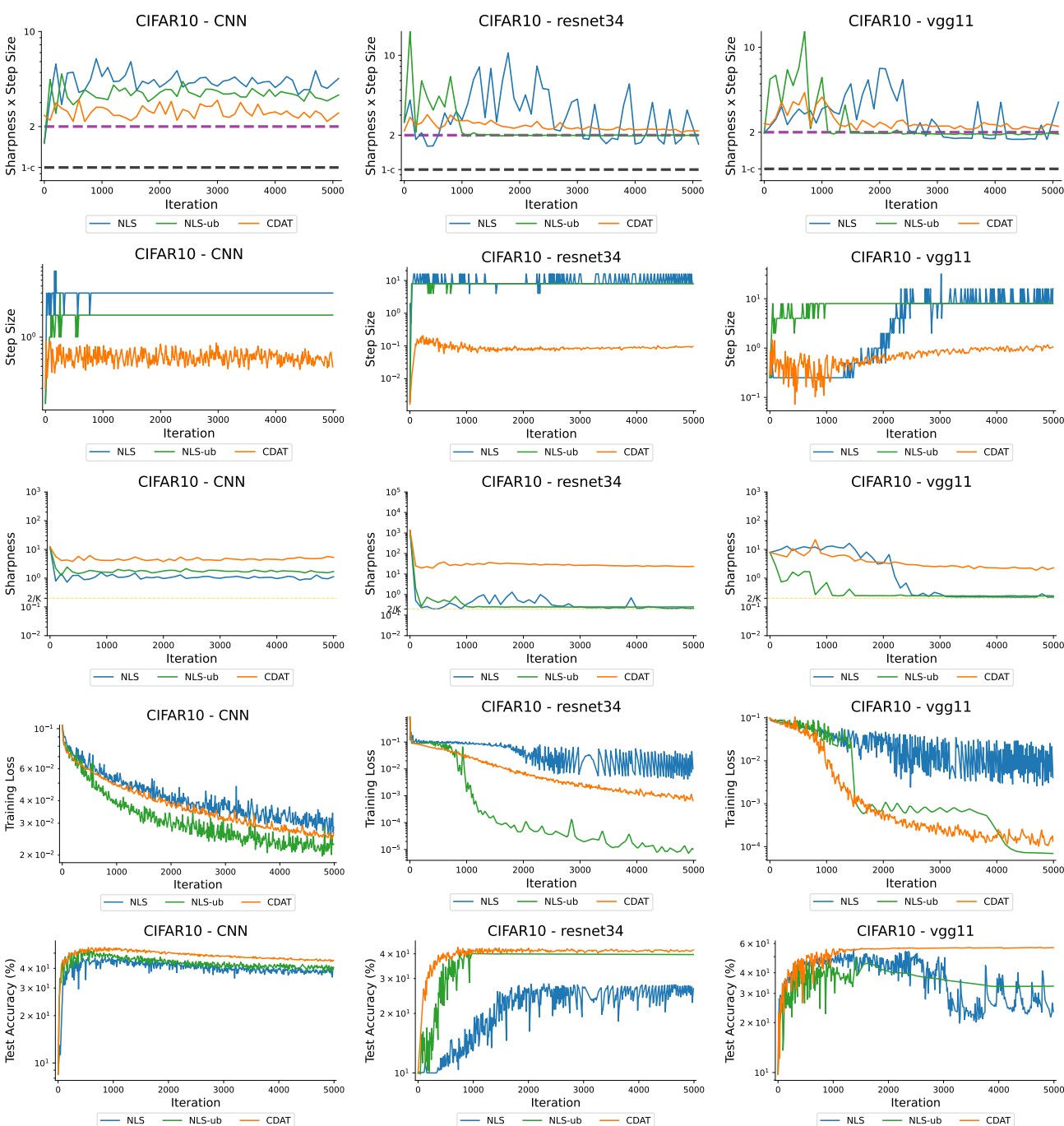

*Figure 32.* We plot the sharpness * step size (1st Row), step size (2nd Row), sharpness (3rd Row), training loss (4th Row), and test accuracy (5th Row) for five different methods. We compare gradient descent with the NLS and NLS-ub line searches as well as gradient descent with the CDAT step size selection. This is repeated for the CNN, resnet34, and vgg11 models on the CIFAR10 dataset.

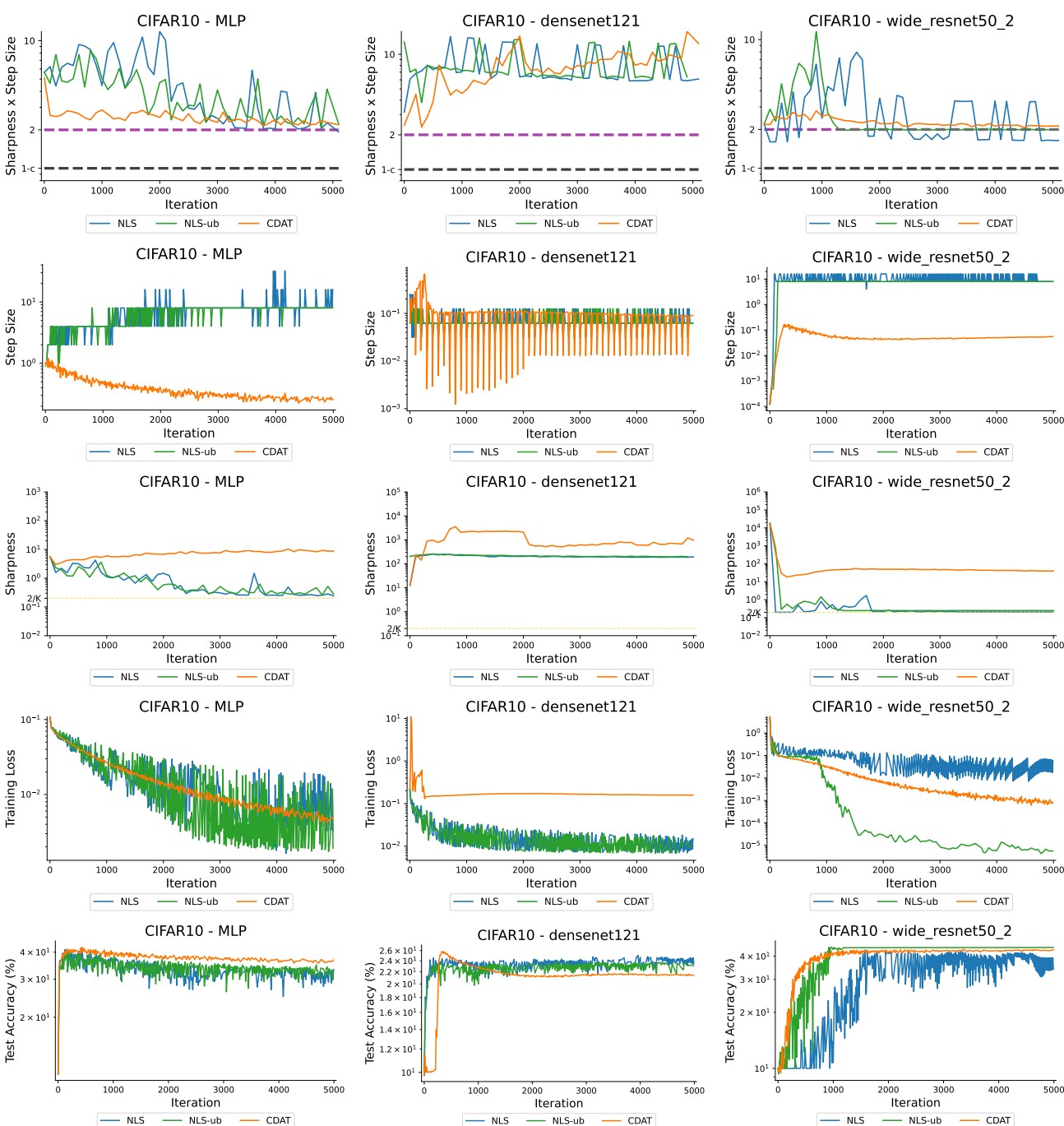

*Figure 33.* We plot the sharpness * step size (1st Row), step size (2nd Row), sharpness (3rd Row), training loss (4th Row), and test accuracy (5th Row) for five different methods. We compare gradient descent with the NLS and NLS-ub line searches as well as gradient descent with the CDAT step size selection. This is repeated for the MLP, densenet121, and wide_resnet50_2 models on the CIFAR10 dataset.

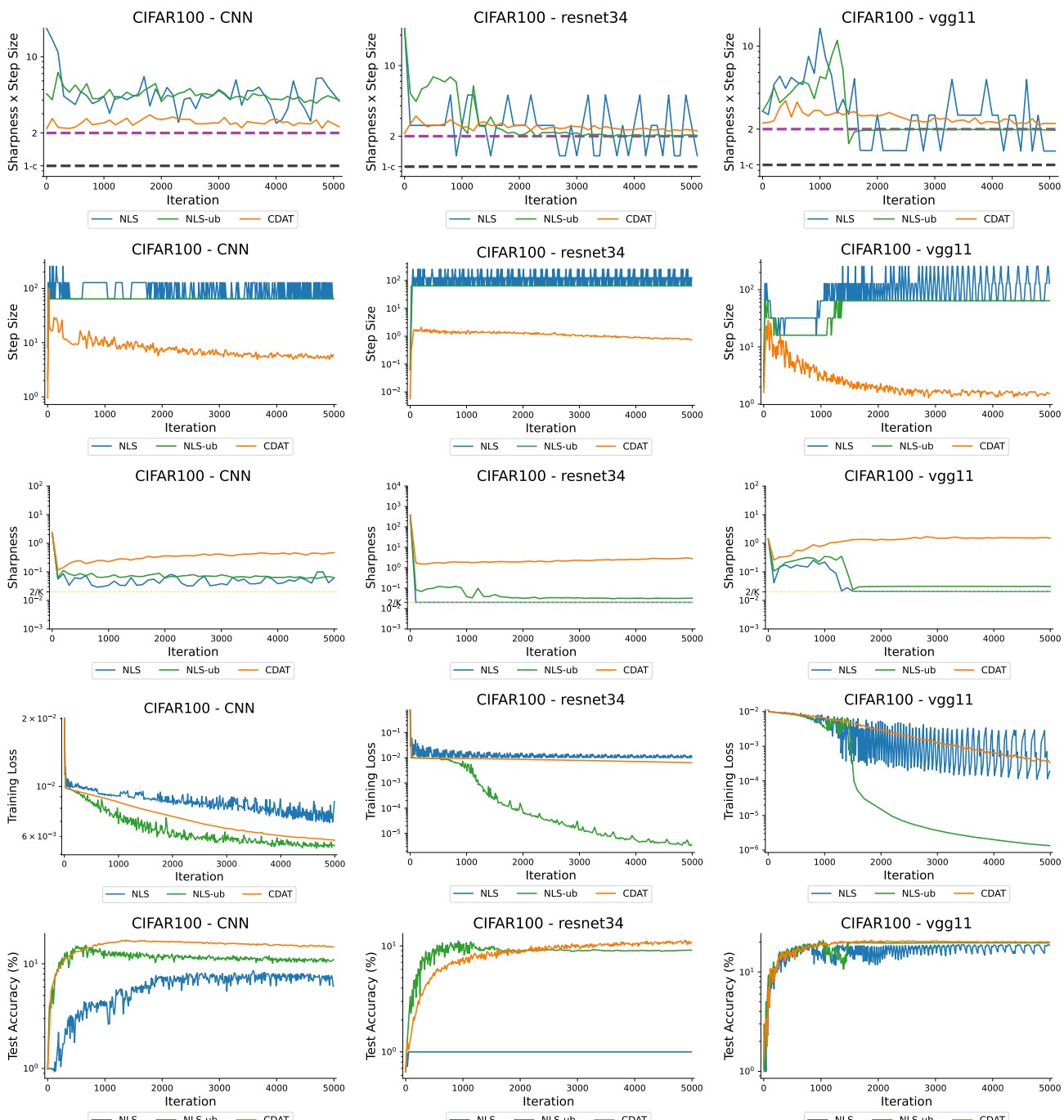

*Figure 34.* We plot the sharpness * step size (1st Row), step size (2nd Row), sharpness (3rd Row), training loss (4th Row), and test accuracy (5th Row) for five different methods. We compare gradient descent with the NLS and NLS-ub line searches as well as gradient descent with the CDAT step size selection. This is repeated for the CNN, resnet34, and vgg11 models on the CIFAR100 dataset.

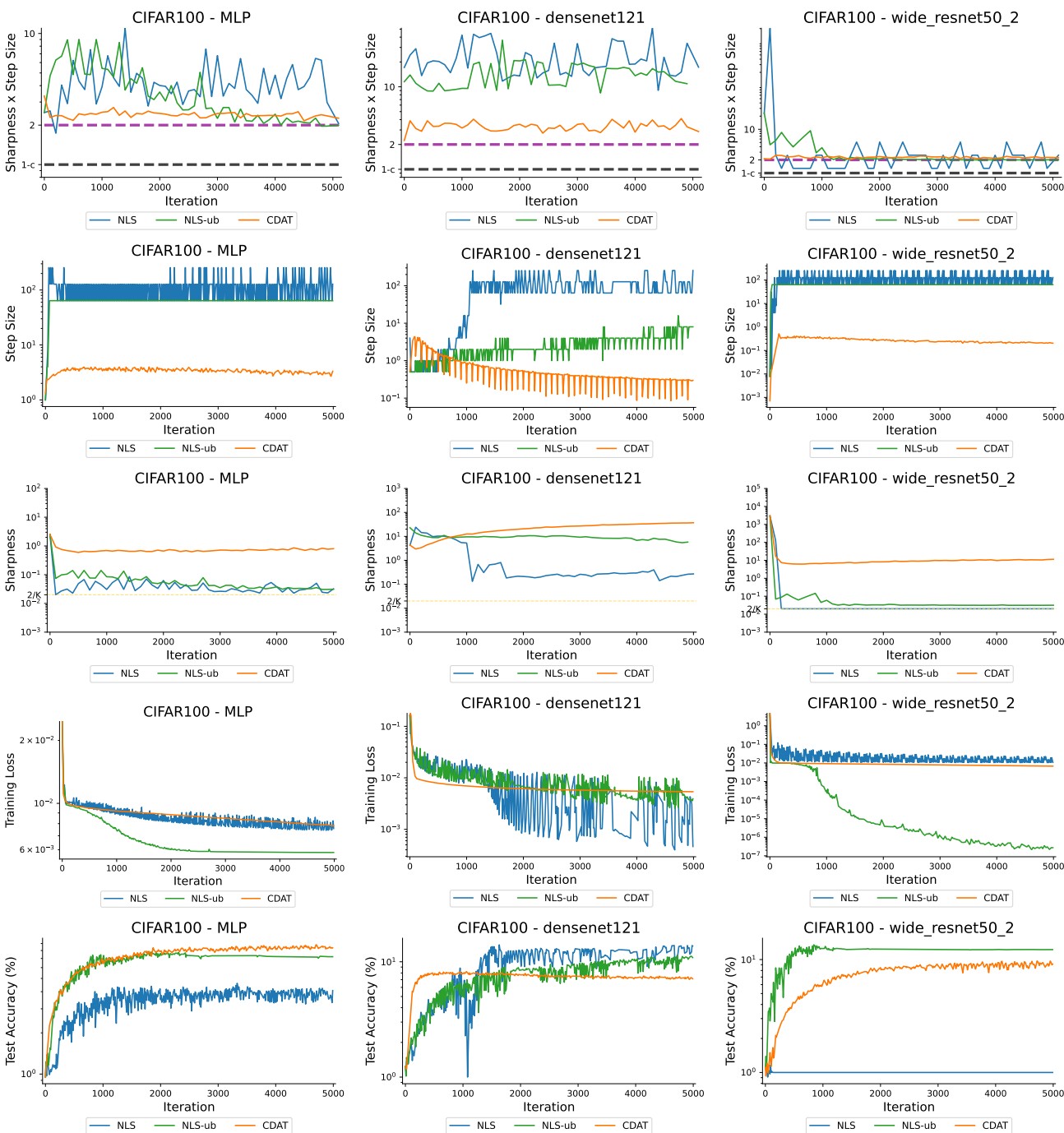

*Figure 35.* We plot the sharpness * step size (1st Row), step size (2nd Row), sharpness (3rd Row), training loss (4th Row), and test accuracy (5th Row) for five different methods. We compare gradient descent with the NLS and NLS-ub line searches as well as gradient descent with the CDAT step size selection. This is repeated for the MLP, densenet121, and wide_resnet50_2 models on the CIFAR100 dataset.

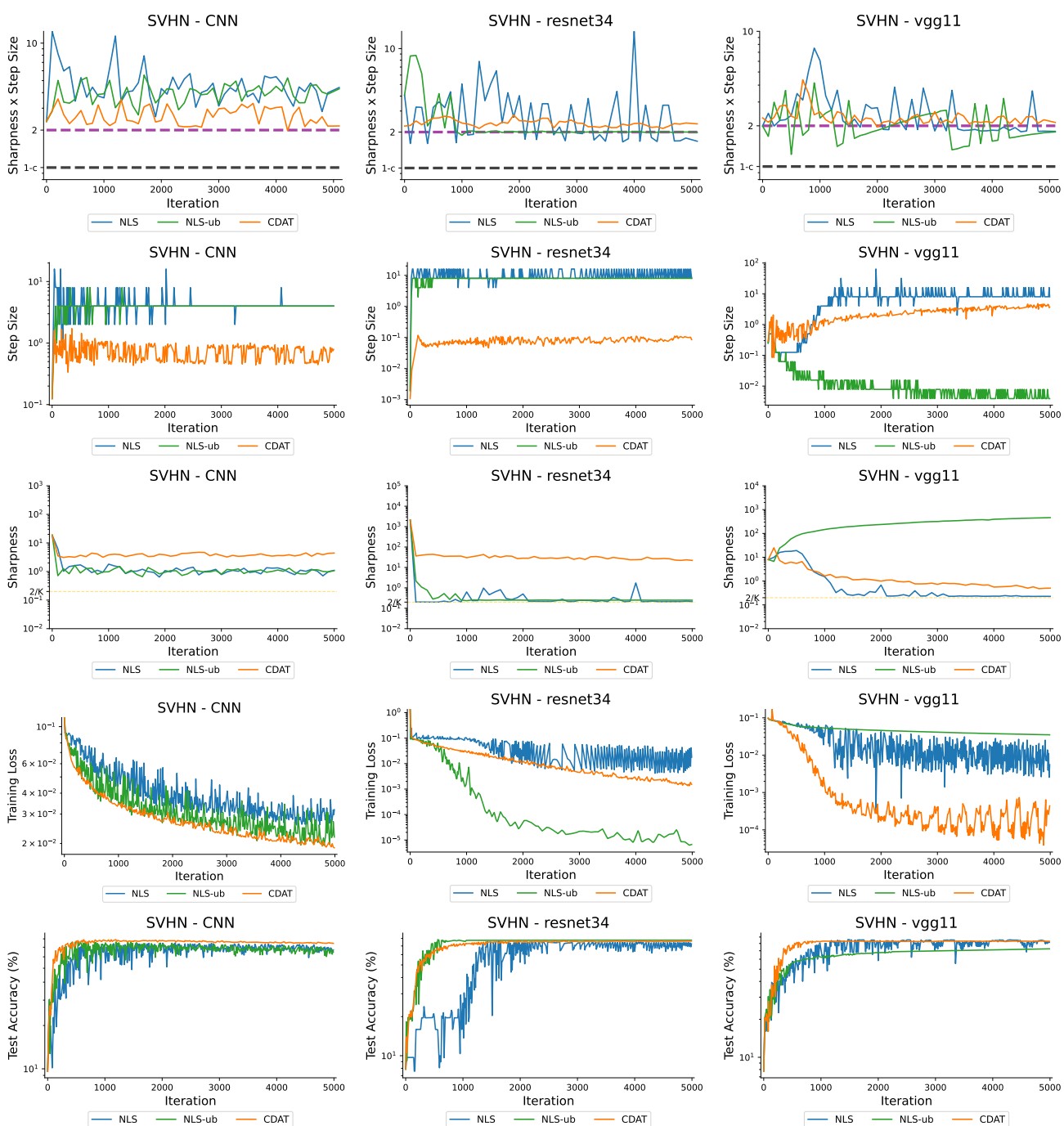

*Figure 36.* We plot the sharpness * step size (1st Row), step size (2nd Row), sharpness (3rd Row), training loss (4th Row), and test accuracy (5th Row) for five different methods. We compare gradient descent with the NLS and NLS-ub line searches as well as gradient descent with the CDAT step size selection. This is repeated for the CNN, resnet34, and vgg11 models on the SVHN dataset.

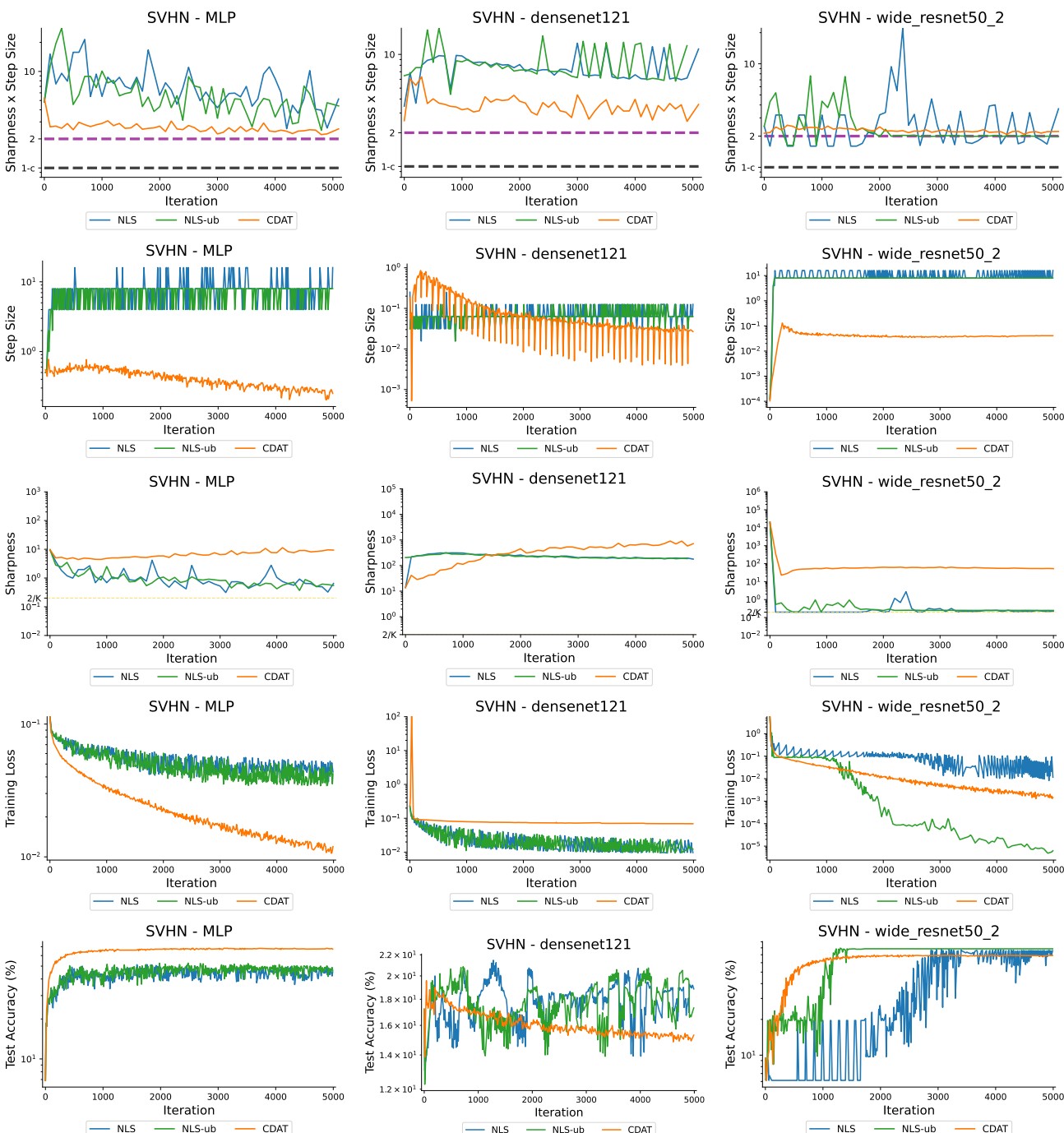

*Figure 37.* We plot the sharpness * step size (1st Row), step size (2nd Row), sharpness (3rd Row), training loss (4th Row), and test accuracy (5th Row) for five different methods. We compare gradient descent with the NLS and NLS-ub line searches as well as gradient descent with the CDAT step size selection. This is repeated for the MLP, densenet121, and wide_resnet50_2 models on the SVHN dataset.

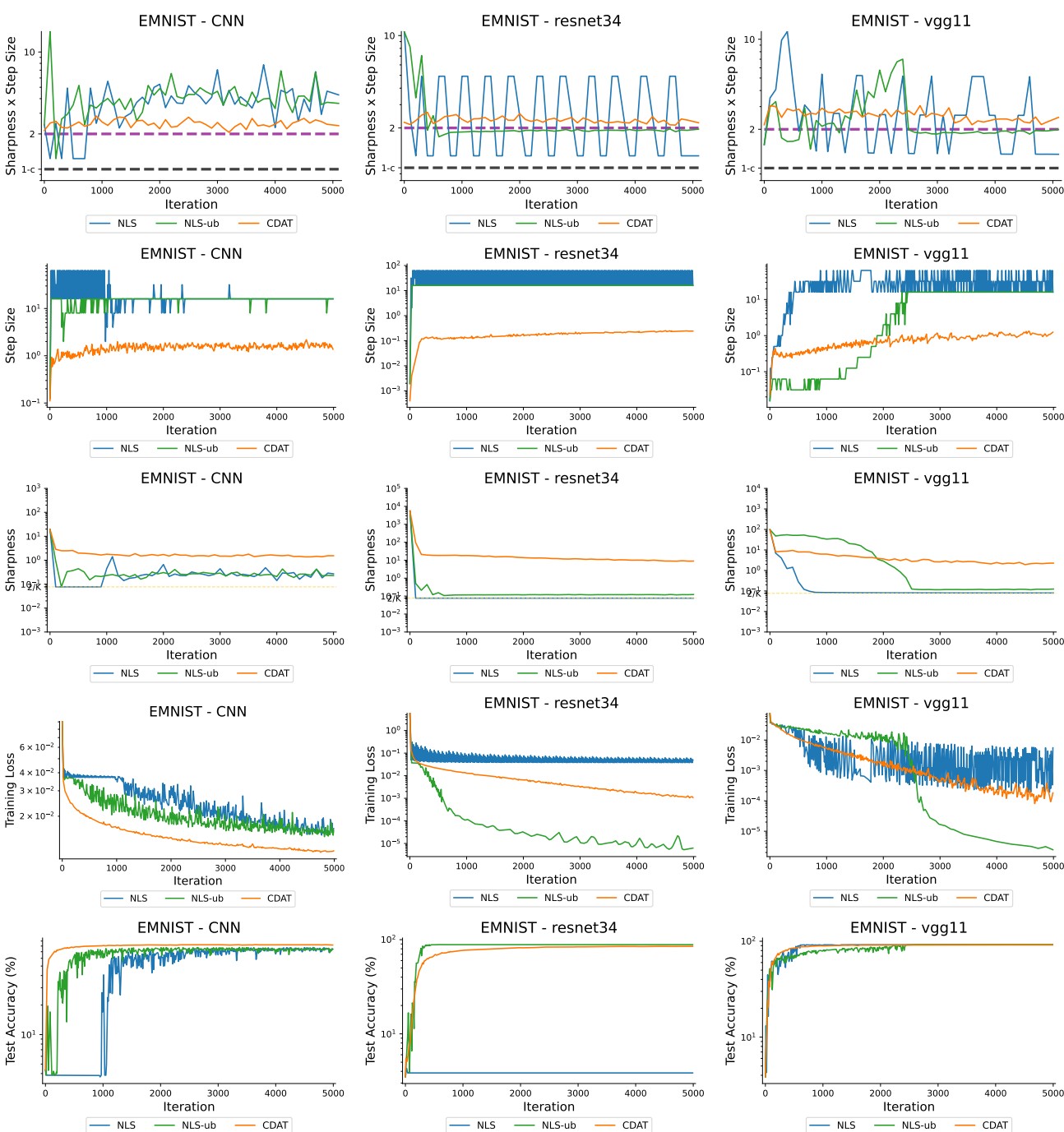

*Figure 38.* We plot the sharpness * step size (1st Row), step size (2nd Row), sharpness (3rd Row), training loss (4th Row), and test accuracy (5th Row) for five different methods. We compare gradient descent with the NLS and NLS-ub line searches as well as gradient descent with the CDAT step size selection. This is repeated for the CNN, resnet34, and vgg11 models on the EMNIST dataset.

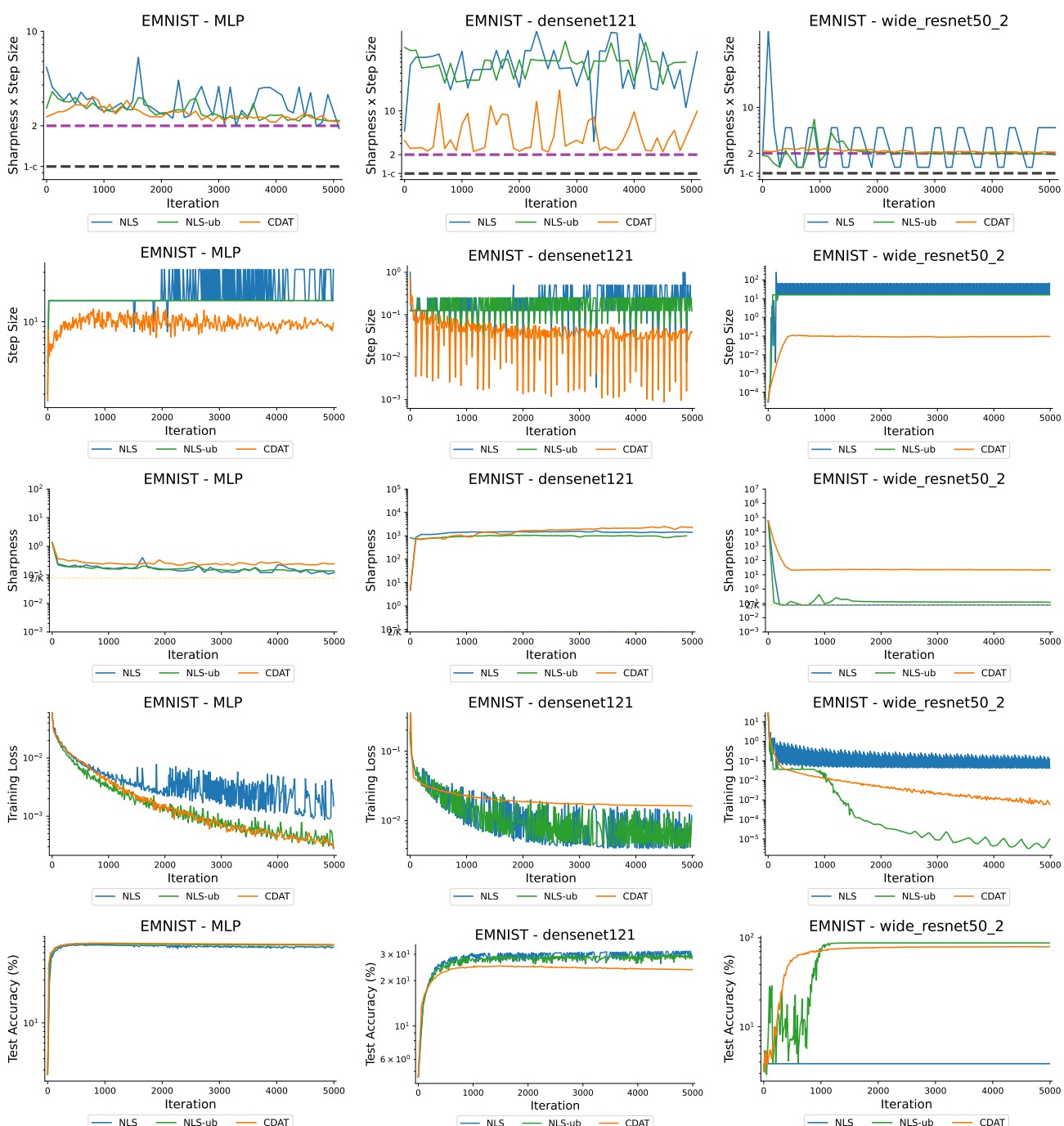

*Figure 39.* We plot the sharpness * step size (1st Row), step size (2nd Row), sharpness (3rd Row), training loss (4th Row), and test accuracy (5th Row) for five different methods. We compare gradient descent with the NLS and NLS-ub line searches as well as gradient descent with the CDAT step size selection. This is repeated for the MLP, densenet121, and wide_resnet50_2 models on the EMNIST dataset.

## I.4. Additional Experiments on the validity of Local Segment Smoothness

In Figures 40 and 41, we show further experiments that verify segment smoothness (Assumption 2.2) holds in a variety settings. Continuing our discussion in Section 3.3, in Figures 40 and 41 we plot the Lipschitz constant $l'(w_k, \eta) := \frac{\|\nabla f(w_k) - \nabla f(w_\eta)\|}{\|\eta \nabla f(w_k)\|}$ along the gradient line at various iterations (0, 100, 500, 1000, and 5000) achieved by NLS. These "instantaneous" values are lower bounds for $l(w_k, \eta_{w_k})$ of Definition 2.1. For a fixed $w_k$, if we assume the gradient to be

continuous and Assumption 2.2 to hold, the value $l(w_k, \eta_{w_k})$ is finite for any finite $\eta \geq \eta_{w_k}$. On the other hand, if we take $\eta \to \infty$, $l(w_k, \eta)$ can still go to infinity. This is what we observe for the magenta line, that is, $l(w_k, \eta)$ is numerically `np.inf` for values of $\eta$ that are larger than 0.01 (e.g., EMNIST×wide_resnet50_2). This proves that some of these neural networks (e.g., resnet34, wide_resnet50_2) do not have globally directionally $L$-smooth objective functions. Moreover, if we exclude what happens at initialization, we observe that $l'(w_k, \eta)$ is stable (or even decreasing) along the gradient line for all step sizes $\eta$ from $10^{-3}$ to 1. Only when moving further away from $w_k$ (e.g., with a step size of 10) do we see that $l(w_k, \eta_{w_k})$ is no longer stable.

Note that the small horizontal dashed lines represent the sharpness at the corresponding iterate, e.g., the magenta dashed line is $\lambda_1(H(w_0))$. As described in the main text, it is difficult to precisely verify Proposition 2.5 because of the difficulty of calculating $l(w_k, \eta_{w_k})$. The instantaneous values $l'(w_k, \eta)$ and the horizontal dashed line can help us check this. In particular, every time the value $l'(w_k, \eta)$ is at least once above the dashed line, this implies that $l(w_k, \eta_{w_k}) \geq \lambda_1(H(w_k))$. This, together with the fact that $\eta_k \cdot \lambda_1(H(w_k))$ is always greater than $2\delta(1-c)$ in Figures 19-26, implies that Proposition 2.5 is verified. We note that this seems to happen in the large majority of our experiments. At the same time, when the values of $l'(w_k, \eta)$ are not similar to the corresponding dashed line, the value $l(w_k, \eta_{w_k})$ is not well approximated by $\lambda_1(H(w_k))$.

From the results achieved on EMNIST, one can observe that the self-stabilization semi-formal bound derived in Damian et al. (2022), i.e., $\lambda_1(\nabla^2 f(w_k)) \leq \frac{2}{\eta}$, may not hold at the globally-flat points. For instance, the step sizes selected in EMNIST×CNN experiment reach values of 64, but the corresponding sharpness and Lipschitz constant do not decrease below 2/26.

Now, looking at the vertical dashed lines, we note that they represent $\eta_k$ and are of the same color as the iteration $k$ at which they are selected. From these lines, we observe that the large majority of the step sizes $\eta_k$ chosen by the nonmonotone line search are all within the regions for which the corresponding $l'(w_k, \eta)$ is stable. Despite the fact that $\eta_k \leq \eta_{w_k}$ is not required to derive Propositions 2.5 and 2.8, it is a good sign that nonmonotone line searches select step sizes in the region where $l'(w_k, \eta)$ is stable. In summary, Assumption 2.2 is numerically satisfied on all the points we have tested it, and in these cases, the corresponding values of $\eta_{w_k}$ are far from infinitesimal.

Finally, we note that Assumption 2.2 is weaker than global $L$-smoothness, global directional $L$-smoothness (Mishkin et al., 2024) and $(L_0, L_1)$-smoothness (also when defined without involving the Hessian) (Zhang et al., 2020). Moreover, Assumption 2.2 is also weaker than local smoothness, as it only needs to hold from $w$ along the gradient line and not everywhere around $w$.

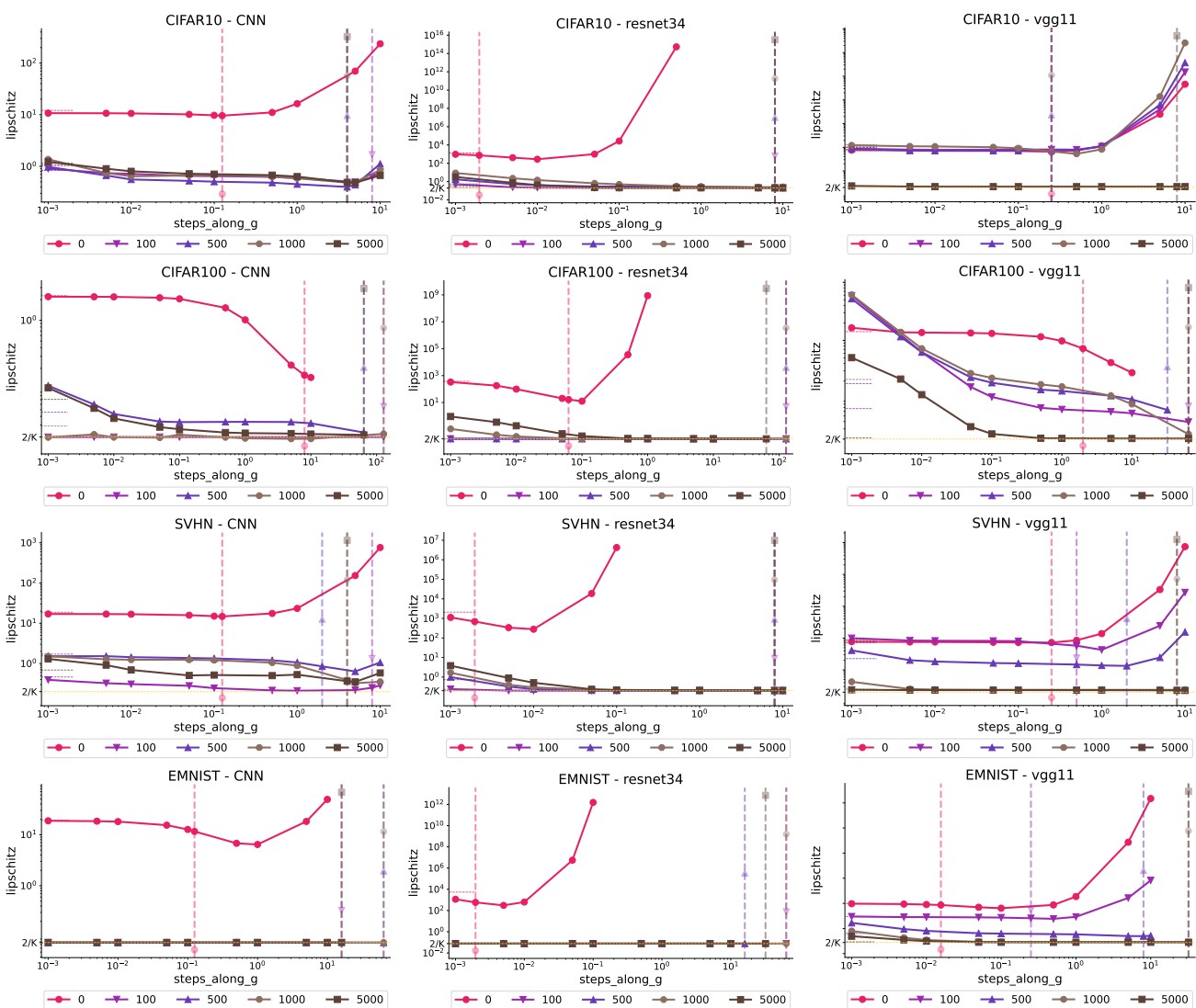

*Figure 40.* We plot the segment smoothness (2.1) for different steps along the gradient direction at varying training iterations for the NLS line search method. In addition, the vertical dashed lines correspond to the selected step size at each iteration. Finally, the horizontal dashed lines correspond to the sharpness at each iteration. This is repeated for the CNN, resnet34, and vgg11 models on the CIFAR10, CIFAR100, SVHN, and EMNIST datasets.

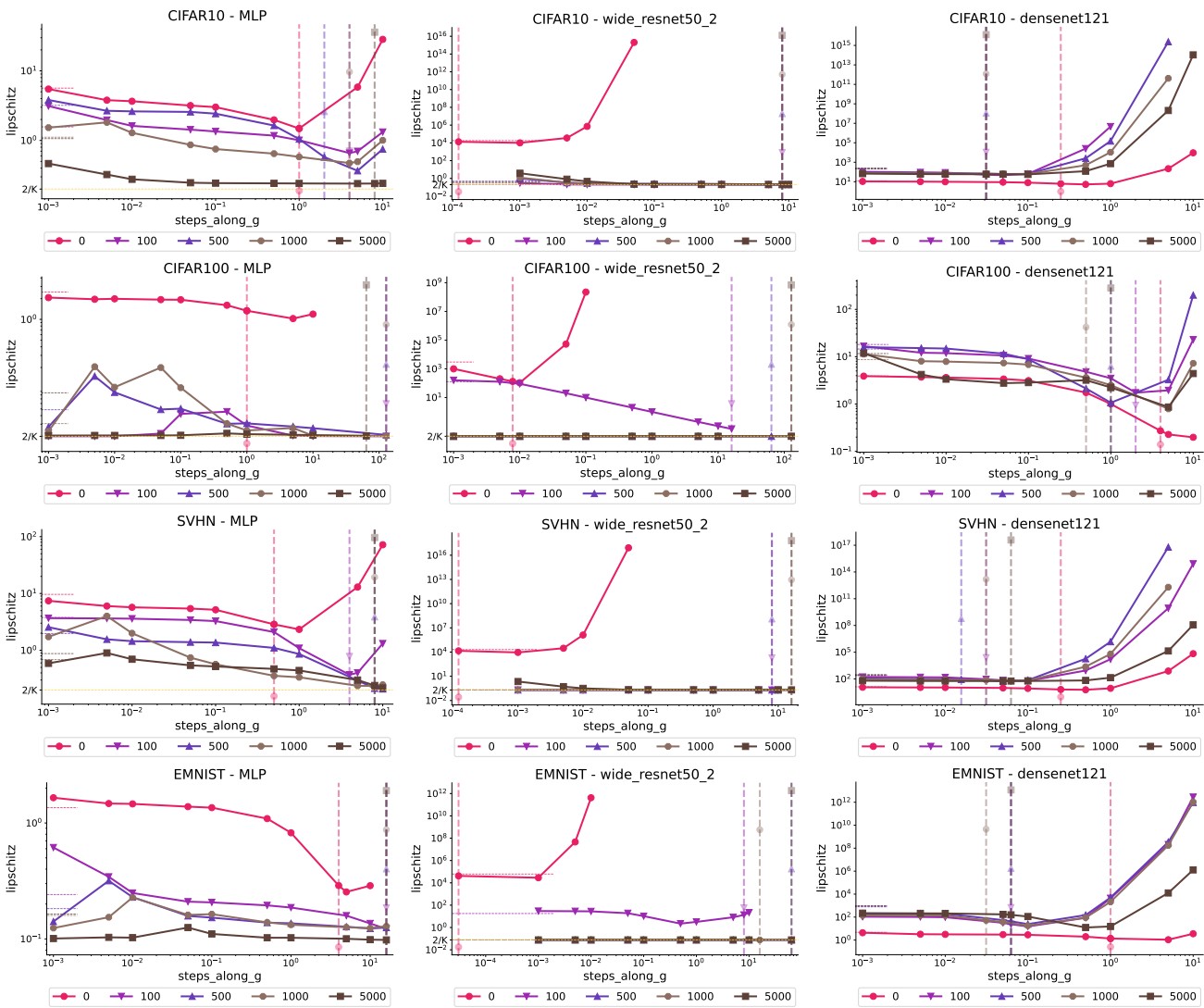

*Figure 41.* We plot the segment smoothness (2.1) for different steps along the gradient direction at varying training iterations for the NLS line search method. In addition, the vertical dashed lines correspond to the selected step size at each iteration. Finally, the horizontal dashed lines correspond to the sharpness at each iteration. This is repeated for the MLP, densenet121, and wide_resnet50_2 models on the CIFAR10, CIFAR100, SVHN, and EMNIST datasets.

## J. No-Bias Experiments

In this section, we discuss what can occur when removing biases from neural network architectures when training with gradient descent with large step sizes. In many of our previous experiments, we show that NLS converges to a globally-flat point. When training some networks where the biases have been removed, we instead show that NLS no longer converges to a globally-flat point but instead collapses to the 0-network (a trivial stationary point that outputs zero regardless of the input). To verify this claim, in Figure 42 we provide the training loss, sharpness, and gradient norm in three different settings, two of which showing convergence to the 0-network. The training loss converges to the random guessing values of $1/26$ and $1/10$, for the EMNIST and SVHN experiments, respectively. The sharpness and gradient norms converge to zero in both experiments. Finally, we checked the output of the network on each image of the dataset and this is constantly 0, showing that NLS is converging to the 0-network. We also provide one experiment where a model with the biases removed is trained on CIFAR10, yet NLS does not converge to the 0-network. We leave to future work exploring the conditions under which this network collapse occurs, since removing the biases does not always lead to NLS converging to the 0-network for all network architectures we tested.

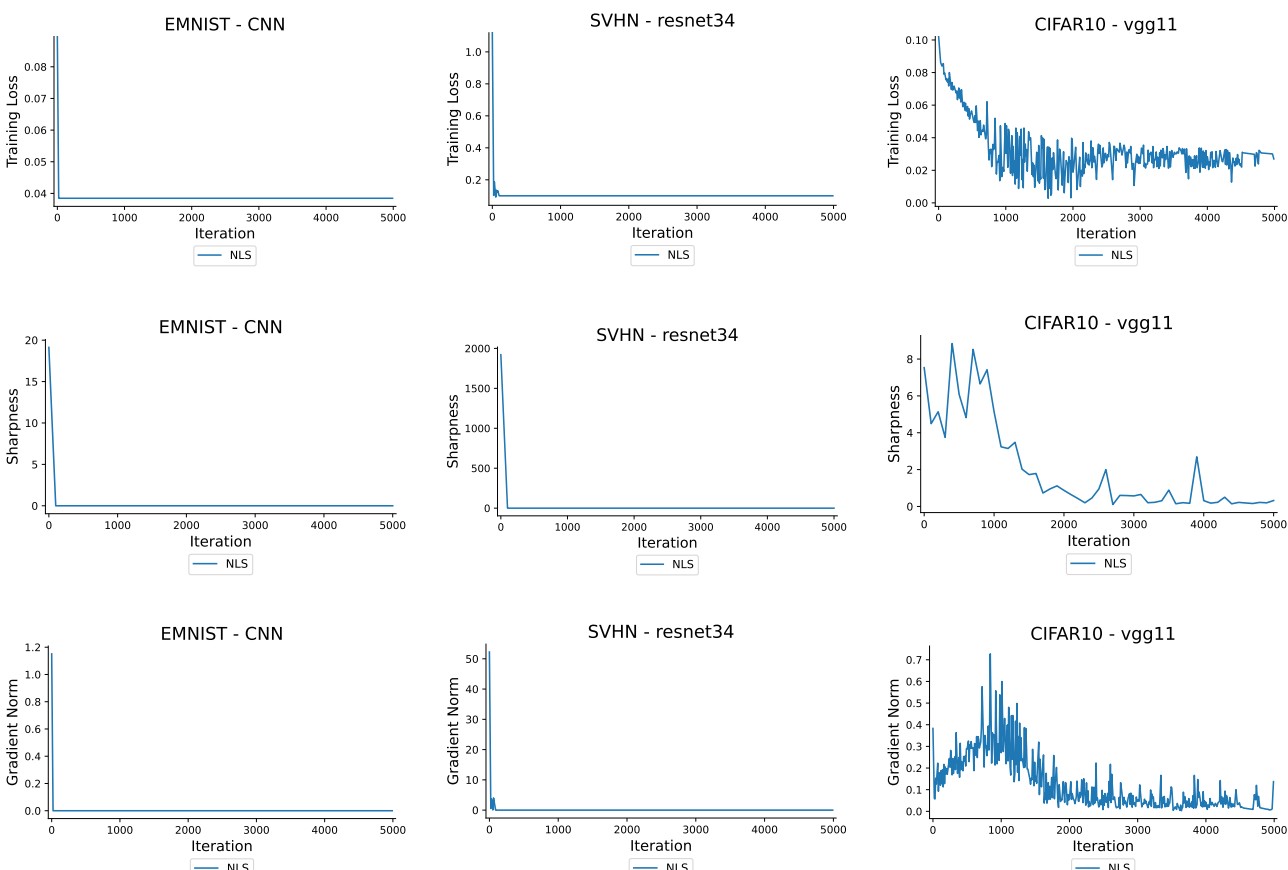

*Figure 42.* We plot the training loss (1st Row), sharpness (2nd Row), and gradient norm (3rd row) for the NLS line search method. This is repeated for the EMNIST×CNN, SVHN×resnet34, and CIFAR10×vgg11 experiments, where the biases have been removed from the last layer of the networks being trained.

