# OpenReview forum: "Flatland: The Adventures of Gradient Descent with Large Step Sizes"
_ICML.cc/2026/Conference — ICML 2026 regular_

### Official Review · Reviewer_b9Sb · 2026-02-25

**Soundness:** 3
**Presentation:** 2
**Significance:** 3
**Originality:** 3
**Overall Recommendation:** 3
**Confidence:** 4

**Summary:**

This paper studies the gradient descent (GD) with large step sizes, where the edge of stability phenomenon emerges. It seeks to address the question of how to find the largest step sizes possible under which GD still converges. Such step sizes can be preferred in practice as they can lead to faster convergence and help find flatter solutions. A definition of large step sizes is proposed, and several line search methods are shown to be capable of finding approximately the desired large step sizes. The validity of these approaches is justified both theoretically and through extensive experiments. It was also discovered in the experiments that the globally-flat solutions might not be desirable as they admit inferior performance. To avoid these solutions, the authors propose to enforce an upper bound on step sizes in the line search.

**Compliance With Llm Reviewing Policy:**

Affirmed.

**Final Justification:**

There is little doubt that the experimental findings presented in the paper are quite impressive. My questions were mostly on the theoretical parts, and the authors addressed some of them in their rebuttals.

In my opinion, the theoretical results of this paper are far from ready for publication; it contains many typos, unreasonable assumptions, and questionable claims. From the authors' responses, one could also see that they are not fully familiar with relevant topics and sometimes make inaccurate claims. While I am sure that all of these could be fixed eventually, I am not confident enough to recommend an acceptance score for this paper before I take a look at the revised version.

**Key Questions For Authors:**

1. Why do you need two parts in the definition of large step sizes? For the large step sizes that you can theoretically justify, is it always the case that P = 0 so that the first part of the definition becomes redundant?

1. Why can Lipschitz-Awardness be obtained with monotone line search (Proposition 2.4) but Holder-awareness cannot (Proposition 2.7)?

1. What is the training loss for the globally-flat points found by the algorithm? If it is high, then maybe they have poor performance, not because they are too flat, but simply because they do not correspond to a well-trained network.

**Limitations:**

yes.

**Strengths And Weaknesses:**

Strengths:
1. This paper recovers the surprising link between line search methods and the EoS phenomenon of GD.
1. The theoretical justifications are mostly rigorous, and the newly proposed definition for large step sizes could be of interest.
1. The experimental results are extensive: The authors test the proposed methods on different datasets, neural network structures, and compare with many baselines.
1. The observation that one should use slightly smaller step sizes to avoid the bad globally-flat region is interesting and could be useful.


Weaknesses:
1. Clarity of the paper (especially Section 2) can be improved. There are quite a lot of typos and inconsistencies in notations/arguments, which make this section hard to read, for example:

a. page 2, footnote 1: Where do you use continuity of h?

b. page 3 line 147: What is $\delta$?

c. page 3, Assumption 2.6: What is $\eta_k$ abd $\eta_{w_k}$?

d. page 3, Theorem 2.8: What is $\mathcal L_0$ and what is $\eta_{max}$?

e. page 4, line 184: What is $f^*$?

f. page 4, equation (6): What is $\eta_{w_k}$?

g. Do $\mu_1$ and $\lambda_1$ both refer to the maximal eigenvalue? If so, why two different notations?

I understand that some of these notations are defined or mentioned in the appendix or after they are first introduced. However, this should be mitigated as it causes significant confusion for the readers.

2. Some of the theoretical results come with unjustified/unclear assumptions. For example:

a. Theorem 2.8: Why do you need the assumption that both level sets are bounded and the step sizes are upper-bounded? It appears that one of them would be sufficient. What is the requirement on $\eta_{max}$, do you just need the sup of step sizes to be finite?

b. Corollary 2.9: The sequential convergence of the iterates is a strong assumption. Can you explain when it holds in your settings?

3. The technical novelty is limited. The line search methods studied are classical in optimization literature, and the theoretical results are mostly expected with standard proofs.

---

> ### Author Rebuttal · Authors · 2026-03-30
>
> We thank the reviewer for the thorough reading and for recognizing the surprising link between line search methods and the EoS phenomenon as one of our key contributions. We believe the concerns raised by the reviewer are addressable, and we do our best to demonstrate this in what follows.
>
> In the revised submission, we will fix the various typos and inconsistencies following the reviewer’s suggestions. In particular,
> footnote 1 will be an extended lemma in the revised manuscript. To address the sometimes fragmented readability of Section
> 2, we will use the extra page in the camera-ready version of the paper to integrate parts of the appendix in the main paper.
>
> We address your concerns regarding the assumptions in our theoretical results as follows:
>
> a) For equality line searches, the boundedness of the $0$-level set does not seem enough, but the upper bound on the step size is also needed as the step size can potentially increase at each iteration. On the other hand, the implication $z_k \to w_k\Rightarrow\nabla f(z_k)\to \nabla f(w_k)$ in the proof of Theorem 2.8 does not hold without further assumptions like the boundedness of the $0$-level set or the uniform continuity of the gradient.
>
> b) The convergence of the iterates can be proved for instance when the $0$-level set is bounded, or when the loss function is
> coercive. The latter holds true for MSE losses with ReLU activation functions (as in our setting) or for cross entropy loss
> that are either regularized or trained with weight decay.
>
> We respectfully disagree with the assessment that the technical novelty is limited. While Theorem 2.8 is admittedly classical, the same cannot be said for the other theoretical contributions. Specifically, Propositions 2.4 and 2.10 establish a relationship that – precisely because it is not required for the standard convergence analysis of GD – has been overlooked in the optimization literature. We argue this is a meaningful gap to fill: such results enrich our understanding of line search methods beyond what convergence proofs strictly demand, and their clarity makes them natural candidates for inclusion also in optimization courses. Beyond these, Propositions 2.7, Corollary 2.9, Lemma 3.1, and Lemma B.22 are also, to the best of our knowledge, novel contributions.
>
> Key Questions for Authors:
> 1. In our view, large step sizes are the opposite of classic "conservative" step sizes, i.e., step sizes that *always* ensure a
> monotone decrease of the objective function. That is, step sizes that are *always* smaller than 2 over the local smoothness
> constant. If one negates this statement (in a logical sense), one gets that large step sizes are those that are *sometimes*
> larger than 2 over the local smoothness constant. To exclude pathological cases where a step size is always very small
> but crosses the threshold only rarely, we additionally impose condition (i) in Definition 2.3. Notice that step sizes that
> are *always* larger than 2 over the local smoothness constant are also not desirable as they provably diverge on quadratic
> functions. Our numerical experiments validate that NLS, NLS-ub and PoNLS yield large step sizes according to Definition 2.3.
> For instance Figure 1 shows that $\eta_k \geq \frac{2}{l(w_k, \eta_k)}$ does not hold at every iteration, but if we consider intervals ($[0, P]$) of sufficient size it is true in every such interval.
> 2. We would like to thank the reviewer for the interesting question. The difference is due to the proof technique that we
> use. At the moment, we cannot exclude that there is a different way to achieve this, but our proof strategy nonetheless
> fails without a positive $\varepsilon_k >0$ (i.e., for monotone line searches). The reason goes back to the use of Young’s inequality in Lemma B.10, which is used to translate the Descent Lemma with exponents $\nu+1$ and $2$ (Lemma B.9) into a version of the Descent Lemma with just the exponent $2$ (Lemma B.10). To achieve this result, we need $\varepsilon_k >0$, that is, a (strictly) nonmonotone line search.
> 3. We confirm that the training loss of the globally flat points tends to be poor (see R264-R270 in the paper), corresponding indeed to a poorly trained network - and this is precisely one of the pitfalls of GD with large step sizes that our paper aims to expose. However, our contribution goes beyond simply documenting this failure: we show that when GD leaves the globally flat region, the network can be trained successfully again (e.g., NLS on the left column of Figure 1). This motivates our proposed method NLS-ub, which avoids globally flat regions and transforms unsuccessful flat training runs into successful ones (e.g., Figure 3). We also clarify that in this setting a poorly trained network corresponds to a saddle point of the loss landscape. In summary, a key contribution of our paper is exposing a phenomenon that can generically derail training (and showing how to fix it), yet has gone unnoticed by the community until now.

---

> > ### Author Rebuttal · Reviewer_b9Sb · 2026-03-31
> >
> > I would like to thank the authors for their detailed response. Several of my questions and concerns remain:
> >
> > 1. **Step size upper bound in Theorem 2.8.** As I mentioned in my original review, the value of the current upper bound on step sizes $\eta_{max}$ is not specified. As it stands, the requirements only suggest the supremum of step sizes is finite. Please clarify what you need here.
> >
> > 2. **Sequential convergence in Corollary 2.9.** I do not agree with the authors' view that bounded iterates/level sets are sufficient for iterates convergence, especially under the equality line search. Please either 1) provide the results in the reference and clarify what exactly the necessary assumptions are, or 2) formulate a precise claim and prove it.
> >
> > 3. **Necessity of part (i) in Definition 2.3.** This question was only partially addressed. For the large step sizes that you can theoretically justify, is it always the case that $P = 0$ so that the first part of the definition becomes redundant?

---

> > > ### Author Response · Authors · 2026-04-07
> > >
> > > We would like to thank the reviewer for the further questions, this further pass through the theory was very useful.
> > > 1. **Step size upper bound in Theorem 2.8**. The upper bound $\eta_{\max}$ plays the same role as the finite initial step size in classical backtracking line searches (see e.g. Theorem 4.25 of Beck [2014]): any finite value suffices. However, we thank the reviewer for asking for clarifications on this point, as the proof of Theorem 2.8 indeed needs a small adaptation in order to be applied to equality line searches. Specifically, one must distinguish two cases: if $\eta_k = \eta_{\max}$, then the step size is bounded away from zero and the conclusion follows directly from (18); if $\eta_k < \eta_{\max}$, then $\eta_k/\delta$ was a genuine rejected candidate of the line search, and the proof proceeds as in the current version. We will update the proof of Theorem 2.8 accordingly.
> > > 2. **Sequential convergence in Corollary 2.9**. We thank the reviewer for this question. The assumption that $\\{w_k\\}$ converges to $w^{\*}$ in Corollary 2.9 is indeed strong in general. However, as shown in the proof of Theorem 2.8 up to (18), under the same assumptions we have $\\lim_{k\\to\\infty} \\| w_{k+1} - w_{k}\\| = 0$, and since $\\{w_k\\} \\subset \\mathcal{L}_0$ is bounded, Bolzano–Weierstrass guarantees the existence of at least one converging subsequence  $\\{w_k\\}\_{k\\in K} \\to w^{\*}$. Corollary 2.9 can therefore always be applied along such a subsequence, whose existence is guaranteed. We will add a remark after the corollary to clarify this point.
> > > 3. **Necessity of part (i) in Definition 2.3**. The short answer is no: with $P = 0$, GD with large step sizes diverges on quadratic functions, which in our view makes such step sizes theoretically unjustifiable. For the long answer, we refer to reply 1 to reviewer ZtZM, where we prove that for nonmonotone line searches there exist $\delta\approx 1$ and $c\approx0$ such that Definition 2.3 holds with $P=0$. In this setting, condition (i) of Definition 2.3 is indeed redundant. However, $\delta\approx1$ is not practical, as the cost of the line search grows exponentially as $\delta\to1$ (see Appendix G for a further discussion). For all other practically relevant parameter choices, $P>0$ and condition (i) are necessary.
> > >
> > > **References**
> > >
> > > A. Beck. Introduction to nonlinear optimization: Theory, algorithms, and applications with MATLAB. SIAM, 2014.

---

### Official Review · Reviewer_jb2J · 2026-03-07

**Soundness:** 2
**Presentation:** 3
**Significance:** 2
**Originality:** 3
**Overall Recommendation:** 3
**Confidence:** 4

**Summary:**

The submission analyzes the training dynamics of neural networks trained on $K$ classification tasks at large learning rates. The authors quantify what a large learning rate means by introducing a path-dependent curvature measure. Furthermore, the authors propose Non-monotone Equality Line Search (NLS) to estimate critical learning rates using only first-order information. Empirically, authors show that NLS tends to minimize the curvature in regions where the top $K$ eigenvalues reach $2/K$. They also provide an explanation of this phenomenon based on the sub-Hessian of the last layer.

**Compliance With Llm Reviewing Policy:**

Affirmed.

**Final Justification:**

I have increased my score after discussing with the authors and other reviewers. However, the following issue remains:

- While it's novel to study the flattest trainable regions of the landscape, the performance is quite low in these regions, and the lessons learnt may not be practical.

**Key Questions For Authors:**

**The global minimum of the landscape is bounded by curvature K/2**

This can arise from various reasons, and I would like to request the authors to verify these claims in the following settings:

1. No biases: Does the result hold when the model has no biases?
2. Prior works (such as [2]) show that with learning rate warmup, the models can achieve an LR 10x larger than 2/K. The same work also shows that the learning rate can be ~100x larger for Maximal Update Parameterization.

In the past few years, the community has moved away from using biases in neural networks, as biases are linked to training stabilities at large scales [3]. This further motivates analyzing the result in the absence of biases.


**SGD Failures vs Divergences**

Prior works [1, 2] suggest that SGD typically achieves its best performance at the largest learning rate below divergence. By comparison, the submission shows experiments where GD fails to achieve the best performance at large learning rates but does not diverge. Does this also stem from the usage of biases?

**Definition of Large Learning Rates**

Definition 2.3 only requires that LR is greater than local segment smoothness once in a while. Operating at EoS is not a well-motivated definition of a large learning rate. Can the authors help me understand why this definition is a good one?

[1] Rethinking conventional wisdom in machine learning: From generalization to scaling, 2024
[2] Why warmup the learning rate? underlying mechanisms and improvements, NeurIPS 2024
[3] PaLM: Scaling Language Modeling with Pathways, 2022

**Limitations:**

- The advantage of the proposed method over methods such as CDAT is not apparent.
- The experiments are limited to classical image classification tasks, and their generalizability to modern LLM training is unclear. (~100x)

**Strengths And Weaknesses:**

Strengths:

- The paper analyzes the practical setting of large learning rate dynamics in neural networks. In particular, quantifying what it means for a learning rate to be 'large' is a well-motivated and important question.
- The paper is clearly written and easy to read, and the theoretical results are sound to the best of my understanding.
- The result that the sharpness value of $2/K$ quantifies the flattest region of the landscape is intriguing.

Weakness:
- As a learning rate tuner method, NLS performs similarly (with 2/K bound) to or at times worse than CDAT (without 2/K bound). This limits the practicality of the method
- The claim that the sharpness value of $2/K$ quantifies the flattest region of the landscape seems to be contingent on the last layer bias, and I am unsure if it holds universally true. This result should be validated without biases, with different parameterizations, and with a learning rate warmup. See questions.

---

> ### Author Rebuttal · Authors · 2026-03-30
>
> We thank the reviewer for the careful reading and for recognizing the importance of quantifying large learning rates and the intriguing nature of our sharpness result. We welcome the opportunity to clarify several points that we believe may have led to an overly critical assessment.
>
> We believe the reviewer may have partially misinterpreted the scope of our work. Our goal is not to propose a superior optimization method, but rather to (i) precisely characterize what it means for a step size to be large, (ii) design an algorithm
> to find such step sizes without diverging, (iii) identify other potential pitfalls, (iv) characterize them from a loss landscape
> perspective, and (v) propose ways to circumvent them. Despite that, the method NLS-ub proposed in Section 3.2 is a first-order algorithm that works as well (if not better) than the second-order method CDAT, while requiring approximately one
> third of its memory (see Appendix G for details on the computational costs of the various algorithms).
>
> We thank the reviewer for the suggestion to validate our claim in the three additional settings described below. We note that the experimental section, which already required approximately 1400 GPU-hours on a mix of A100 and H100 (see Appendix C.6), has been independently described as extensive by other reviewers. We hope this further demonstrates our commitment to thorough empirical validation. We also remark that whether the no-bias setting is truly more representative of modern practice than the with-bias setting is debatable: all batch normalization layers implemented in pytorch include biases by default and are ubiquitous in modern architectures. We describe the results of the three additional settings as follows:
> 1. When removing biases, GD with large step sizes no longer converges to a globally-flat point, but instead collapses to the
> $0$-network, i.e., a trivial stationary point that outputs zero regardless of the input. This is consistent with our previous
> results: as discussed in R275–R277, the last-layer bias is precisely what keeps the training “barely alive” in globally flat
> regions. At the $0$-network, both the gradient and the sharpness vanish (but not necessarily all its weights), confirming
> it as a degenerate but valid stationary point. We will include a discussion of the no-bias setting in the updated manuscript, as it nicely ties up one loose end of the story.
> 2. Warm-up learning rates are discussed in our paper in Appendix D. We show that pushing the step size above the value $K$ so that the sharpness reaches $2/K$ leads to worse performance than keeping the step size below $K$ and avoiding the globally flat region entirely.
> 3. Finally, we ran experiments using the Maximal Update Parameterization (MUP) and found that this parameterization
> allows for sharpness values below the $2/K$ threshold, consistent with the findings of prior work. This seems to be in line with Lemma B.20: by rescaling the outputs of the network layers, MUP induces a corresponding rescaling of the
> subhessian.
>
> Key Questions for Authors:
> 1. Please see the experimental results discussed above. Regarding the comment that “models can achieve an LR $10×$
> larger than $2/K$”: we want to clarify that in our paper it is the *sharpness* that is lower bounded by $2/K$, not the step
> size. In fact, NLS and PoNLS often select step sizes far exceeding $2/K$. As a concrete example, in Figure 1 (VGG11
> on CIFAR-100), $2/K = 0.02$ while selected step sizes exceed $100$ – more than $5000×2/K$. Similar ratios appear
> throughout our experiments.
> 2. Addressing your question about GD failing to achieve the best performance for large step sizes and whether this stems from bias usage, we point to the discussion above. When in the "with-bias" setting the method would converge to a globally-flat point, in the "no-bias" setting it does not diverge but instead converges to the trivial, collapsed $0$-network. The ability to not diverge despite the very large step sizes is ensured by the nonmonotone line search. The common wisdom pointed out by the reviewer (e.g., [1,2]) is not contradicted by our results (see for instance the numerical results of NLS-ub) but rather completed by adding a warning sign: "Keep your step sizes outside flatland."
>
> 3. Although the edge of stability (EOS) inspired our definition of large step sizes, Definition 2.3 is more general than the
> EOS phenomenon. In our view, large step sizes are the opposite of classic "conservative" step sizes, i.e., step sizes that
> *always* ensure a monotone decrease of the objective function. That is, step sizes that are *always* smaller than 2 over
> the local smoothness constant. If one negates this statement (in a logical sense), one gets that large step sizes are those
> that are *sometimes* larger than 2 over the local smoothness constant. To exclude pathological cases where a step size
> is always very small but crosses the threshold only rarely, we additionally impose condition (i) in Definition 2.3.

---

> > ### Author Rebuttal · Reviewer_jb2J · 2026-04-02
> >
> > I thank the reviewers for their rebuttal. However, the rebuttal references new experimental results but provides no figures or data to support them. It would be helpful to provide results with the claims. I mention some of them below.
> >
> > >  all batch normalization layers implemented in PyTorch include biases by default and are ubiquitous in modern architectures.
> >
> > - My understanding is that large-scale training avoids training with biases since Palm. Only old vision models, such as ResNets, VGG19, etc, keep on using it. Furthermore, the instability arguments are based on last-layer biases; would that apply to BatchNorm?
> >
> > > When removing biases, GD with large step sizes no longer converges to a globally-flat point, but instead collapses to the $0$-network, i.e., a trivial stationary point that outputs zero regardless of the input.
> >
> > - Can the author provide experiments showing it? This contradicts my understanding, and I would like to see the results (and reproduce them).
> >
> > > Finally, we ran experiments using the Maximal Update Parameterization (MUP) and found that this parameterization allows for sharpness values below the $2/K$ threshold, consistent with the findings of prior work.
> >
> > - Same issue here. Can the authors provide the experimental results?
> >
> > > On the large step size condition
> >
> > - I am still not convinced that this is a sufficient condition. Consider a toy setup with progressive sharpening with the global minimizer $\theta^*$ with sharpness $K$ and initial sharpness is k >> K. Choose a learning rate slightly larger than 2/k. Due to progressive sharpening, the loss will go down in an oscillatory fashion.
> > Would that be a large learning rate according to your definition? The definition allows a learning rate that never comes close to the 2/K threshold to still qualify as "large." Can the authors please clarify?

---

> > > ### Author Response · Authors · 2026-04-07
> > >
> > > We would like to thank the reviewer for the further questions.
> > > 1. To bias or not to bias. While we agree with the reviewer that many recent large-scale LLMs have moved away from the
> > > use of biases, biases remain widely used in the community - BERT, GPT-2, GPT-J, and notably BLOOM and OPT,
> > > both released after PaLM, all retain them. The same holds for the majority of transformers-based models designed for
> > > image classification: ViT, DeiT, BEiT, and MAE all include biases. Furthermore, LayerNorm and GroupNorm layers
> > > in Pytorch all include biases by default, and these are widely used layers in transformers-based architectures.
> > > More fundamentally, we would argue that biases can never become truly "outdated": they allow the basic modeling
> > > capacity of shifting the output of a function from the origin. Removing them is a pragmatic engineering choice at large
> > > scale, not a sign that they are no longer meaningful or worth studying.
> > > Finally, yes, Lemma 3.1 and Lemma B.22 apply to any network that employs a bias in the last layer.
> > > 2. We provide experiments showing that GD with large step sizes converges to the $0$-network when training both the CNN
> > > model on EMNIST and the resnet34 model on SVHN without last layer biases. For more details on our experiment
> > > setup, see Appendix C of our paper. In the following links the reviewer can see the sharpness ([link](https://imgur.com/2AzCj8p),[link](https://imgur.com/0PEZe3P)), the training
> > > loss ([link](https://imgur.com/1G6G78s),[link](https://imgur.com/YAEK3Xs)) and the gradient norm ([link](https://imgur.com/Jpxd620), [link](https://imgur.com/arzkFeh)) respectively for CNN on EMNIST and resnet34 on SVHN. In both settings, we see that the sharpness and the gradient norm converge to $0$. The training loss converges to the random
> > > guessing values of $1/26$ and $1/10$, respectively. To test that this is indeed a $0$-network, we checked the output of the
> > > network on each image of the dataset and this is constantly $0$.
> > > 3. Similar to the no bias experiments, we provide the results for the NLS method for training our CNN model on EMNIST
> > > and a resnet34 model on SVHN using the MUP initialization. We provide the sharpness plot for EMNIST at this [link](https://imgur.com/l5vZCli) and SVHN sharpness plot at this [link](https://imgur.com/CO07IWv). For the EMNIST plot, we see that the sharpness drops below $2/26 \approx 0.0769$, and for the SVHN plot, the sharpness drops below $2/10 = 0.2$.
> > > 4. In the toy setup suggested by the reviewer, the loss will go down in an oscillatory fashion not because of progressive
> > > sharpening but because of the use of large step sizes (as defined in Definition 2.3) that *will* periodically go beyond
> > > the value $2/l$, where $l$ is the *local* Lipschitz constant (i.e., in the approximation made by the reviewer the *local*
> > > sharpness). This local value will be initially $k$ and later on $K$. Thus, if one employs large step sizes with e.g. $P = 1$,
> > > they *will* go beyond $2/K$ when GD is very close to $\theta_{*}$. Notice that the case $P = 0$ is covered by our definition, but
> > > this would induce provable divergence of GD on many functions.

---

### Official Review · Reviewer_ZtZM · 2026-03-10

**Soundness:** 4
**Presentation:** 4
**Significance:** 3
**Originality:** 4
**Overall Recommendation:** 4
**Confidence:** 4

**Summary:**

The paper investigates gradient descent with large stepsizes using non-monotonic linesearches as a tool to enforce such large stepsizes. Namely, the authors define large stepsizes as being always higher than the optimal stepsize given a local smoothness condition with regular hops above the maximal stepsize for decrease. They show that non-monotonic linesearches can ensure such a behavior.

Equipped with an algorithm capable of such large stepsizes, they analyze the behavior of gradient descent (full batch) on deep networks. They find that sur permanent large stepsizes drive the sharpness of the network towards its minimum, a phenomenon never observed before (up to my knowledge). They also observe such minimum sharpness is actually quite bad for performance, which challenges the idea that "flat minima are good". Nevertheless they are capable of finding "successful flat minima" by ensuring that the sharpness remains slightly over its minimum. The authors provide an extensive set of measurements to tentatively explain the observed phenomenon.

**Compliance With Llm Reviewing Policy:**

Affirmed.

**Final Justification:**

After discussion with the reviewers, I agree that the theoretical parts need a thorough revision to ensure that the paper is at its best for publication. I hope the authors will find that this is an opportunity to et the possible paper out.

**Key Questions For Authors:**

- Can the authors provide a rigorous formulation and proof of the fact "by setting $\delta\approx 1$, and $c\approx 0$ then (ii) holds with $P\approx 0$"?
- Non-monotonic linesearches were investigated by [1], it would be reasonable to include this work as it seems quite related and maybe discuss the results.
- (detail slightly unrelated) On the small remarks in the related work: the authors imply that UGM from Nesterov can be run without the need of $\varepsilon$, but what would be then the convergence rate (which usually had a $\varepsilon$)?

[1] Fox, C., Galli, L., Schmidt, M. and Rauhut, H., 2024. Nonmonotone Line Searches Operate at the Edge of Stability. In OPT 2024: Optimization for Machine Learning.

**Limitations:**

The analysis is restricted to the full batch case.

**Strengths And Weaknesses:**

**Strengths**
- The overall contributions are really original to me. The idea that sharpness could be driven by learning rate schedule was brushed off by [1] but this paper shows a much deeper control driving the sharpness to a minimum that had not been observed before. In particular the authors show that monotonic or non-monotonice linesearches, with their choice of parameters, can provide full probing tools on the optimization behavior of gradient descent.
- The theory is carefully crafted. The smoothness assumption is restricted to a segment and verified in the experiments. The theoretical results are simple but correct and informative. The link between linesearches for holder smooth functions and nonmonotic linesearches is also interesting.
- Proving that the non-monotonic linesearches actually reach the minimum is also an important finding that is well shown.
- The authors analyze the phenomenon through many interesting lenses. For example, the authors show saturation of the neurons explaining the failure of the flat minima. It would be quite interesting to then revisit the questions of plasticity loss/gain with large/small learning rates.
- The paper presents extensive results (70 pages) covering many questions a reader would have.
- (Great title, and great tone throughout the paper)

**Weaknesses**
- The analysis is restricted to the full batch case.
- Some parts would benefit from a few clarifications see below.

Clarifications needed:
  - the reason why $h(2/l(w, \eta_w) \geq 0$ in the first footnote (page 2 second column) is unfortunately unclear to me. Maybe add a full lemma somewhere for clarity?
  - line 148: $\delta$ is used before being defined
  - Proposition 2.7: define $\varepsilon$
  - below Proposition 2.7 "For the reference value from ..." What reference value?
  - Corollary 2.9: say what $\mu_i$ are eigenvalues of. (Its the Hessian but a proposition should be self contained as much as possible).


[1] Roulet, V., Agarwala, A., Grill, J.B., Swirszcz, G., Blondel, M. and Pedregosa, F., 2024. Stepping on the edge: Curvature aware learning rate tuners. Advances in Neural Information Processing Systems, 37, pp.47708-47740.

---

> ### Author Rebuttal · Authors · 2026-03-30
>
> We would like to thank the reviewer for the comments and the time spent reading the paper. The reviewer fully captured the spirit and goals of our work. We also found the connection to plasticity loss and neural collapse to be worthy of further exploration, as flatland is indeed an area of reduced plasticity, while staying just above it allows to largely mitigate this issue. Below we address your comments on our work.
>
> In the final version of the paper, we will address the various clarifications needed as suggested by the reviewer. In particular,
> footnote 1 will be an extended lemma in the revised version of the paper.
>
> Key Questions for Authors:
> 1. We would like to thank the reviewer for pointing this out. A careful analysis reveals that this result holds only for
> nonmonotone line searches, bringing up another difference between these methods and their monotone counterpart. In
> the following, we provide a sketch of this proof. From the proof of Proposition B.5, by keeping $\varepsilon_k>0$ (this is lower
> bounded by zero in the paper) we obtain
> $$\eta_k \geq \frac{2(1-c)\delta}{l(w_k, \eta_{w_k})} + \varepsilon_k \frac{2\delta^2}{l(w_k,\eta_{w_k})\|g_k\|^2} > \frac{2(1-c)\delta}{l(w_k, \eta_k)} .$$
> Now, substituting $c=0, \delta=1$ yields $\eta_k > \frac{2}{l(w_k, \eta_{w_k})}$, therefore, as the right hand side is continuous in $\delta$ and $c$ there exists $c\approx 0$, $\delta \approx 1$ with $\eta_k > \frac{2}{l(w_k, \eta_k)}$, which satisfies condition (ii) of Definition 2.3 with $P=0$. Notice that this does not contradict convergence, for instance on quadratic functions, as $\varepsilon_k\to0$, and, consequently $\eta_k\to\frac{2}{l(w_k, \eta_k)}.$
> 2. We will add a citation to the work and discuss the differences with our submission.
> 3. Considering Theorem 1 of the UGM paper (Nesterov 15’), the $\varepsilon$ will disappear from the right hand side
> of the bound and appear on the left hand side, as this value (e.g., $\varepsilon:=\varepsilon_k=\frac{1}{k}\sum_i^kf(w_i)-f(w_k)$) is automatically controlled by the algorithm and goes to $0$ for $k$ that goes to $\infty$.

---

> > ### Author Rebuttal · Reviewer_ZtZM · 2026-04-02
> >
> > The authors provided required details and I trust them to integrate them.
> > I will keep my score. This paper shows that one can control the sharpness of the network with adequate stepsizes. I don't think it was shown as clearly as it is here, and even if it does not provide yet a new state of the art algorithm, it advances our understanding of the field.

---

### Official Review · Reviewer_F4JT · 2026-03-10

**Soundness:** 3
**Presentation:** 3
**Significance:** 4
**Originality:** 3
**Overall Recommendation:** 5
**Confidence:** 3

**Summary:**

This paper proposes a method for finding large step sizes using line search and analyse SGD when run with these step sizes. The authors find that the model immediately drops to the sharpness that Edge of Stability work predicts it will reach and often gets stuck in globally flat minima that are not useful for training.

**Compliance With Llm Reviewing Policy:**

Affirmed.

**Final Justification:**

Most points/questions addressed satisfactorily. Clarity concerns are hard to confirm without seeing updated manuscripts and these require significantly rethinking the structure. As such I am maintaining my score.

**Key Questions For Authors:**

1. The typical claim is 'flatness means better generalization' not 'flatness means better' (L371), what is the generalization gap between train/test for the globally flat solutions? I suspect the generalization gap is small as the train loss is so bad but if not that would be very interesting to include.
2. Is PoNLS the Polyak initial step size with NLS? It doesn't seem to be stated anywhere, just that it is 'the deterministic version of PoNoS'. If it is then a simple statement of that would be helpful.

**Limitations:**

yes

**Strengths And Weaknesses:**

Strengths
1. Proposes a clear way to find large step sizes close to the largest possible step size
2. Shows that these step sizes often result in finding the globally most flat point in parameter space but that these points are not useful for training/generalization, providing important insights into the discussion on flat minima
3. Provides a fix for the above in order to train well with this method

Weaknesses:
1. Writing can often be a bit unclear and requires multiple passes, jumping back/forth and sometimes reading appendices to follow.

Minor points:
1. $\lambda_1$ is not defined in the main text, add a note of how it differs from $\mu_1$ before the first reference.
2. Algorithm 2 seems to be written strangely with the boolean flag, two consecutive while loops would both make it more obvious what is happening and match the theory more cleanly.
3. A description of $R_k$ and how it relates to monotone/non-monotone line searches around equations 2 & 3 would be helpful

---

> ### Author Rebuttal · Authors · 2026-03-30
>
> We would like to thank the reviewer for the comments and the time spent reading the paper. We are glad that the reviewer appreciated the clarity of our approach, and found our analysis of the relationship between large step sizes and flat minima to be a meaningful contribution to the ongoing discussion on flat minima and generalization.
>
> Regarding the clarity of our writing, we will use the extra page to integrate parts of the appendix into the main paper with the
> goal of reducing back-and-forths in the reading. Additionally, in the revised submission we will fix the three minor points
> following the reviewer’s suggestions.
>
> Key Questions for Authors:
> 1. As the reviewer expected, the generalization gap is small since the training loss and test accuracy are both poor in the
> globally flat regions. The test accuracy is reported in the last row of Figures 19-26 for MSE loss and in the second to
> last row of Figure 11-18 for cross entropy loss.
> 2. The reviewer is right, PoNLS is NLS using the Polyak step size for initialization. We will clarify this in the revised
> submission of the paper.

---

> > ### Author Rebuttal · Reviewer_F4JT · 2026-04-01
> >
> > Minor points/questions addressed satisfactorily (though I would recommend the authors adjust the phrasing on L371 in light of their response to Q1, I don't think it is ever claimed that flatter is better independent of train accuracy, just that it *generalizes* better). I unfortunately cannot increase my score as I cannot confirm any improvements in clarity and structure that caused a reduced score initially.

---

### Decision · Program_Chairs · 2026-04-30

**Decision:**

Accept (regular)

**Comment:**

The reviewers appreciate the **strengths** of the paper: clear method (F4JT), important insights (F4JT, ZtZM, jb2J, b9Sb), original (ZtZM), carefully crafted theory (ZtZM, b9Sb), extensive results (ZtZM, b9Sb), clearly written (jb2J).

The reviewers also find the **weaknesses** of the paper: writing clarity concerns (F4JT, ZtZM, b9Sb), restricted to the full batch case (ZtZM), limited practicality of the method (jb2J).

The reviewers were somewhat split on this paper. After reading the paper and considering the reviews and author's responses carefully, I recommend to accept this paper and encourage the authors to incorporate the feedback as many reviewers think it requires major revision.